# SOX11 regulates SWI/SNF complex components as member of the adrenergic neuroblastoma core regulatory circuitry

Bieke Decaesteker [1,2,10] ✉, Amber Louwagie [1,2,10], Siebe Loontiens [1,2], Fanny De Vloed [1,2], Sarah-Lee Bekaert [1,2], Juliette Roels [1,2], Suzanne Vanhauwaert [1,2], Sara De Brouwer[1,2], Ellen Sanders[1,2], Alla Berezovskaya[3], Geertrui Denecker[1,2], Eva D'haene[1,2], Stéphane Van Haver[1,2,4], Wouter Van Loocke[1,2], Jo Van Dorpe [2,5], David Creytens[2,5], Nadine Van Roy[1,2], Tim Pieters[1,2], Christophe Van Neste [1,2], Matthias Fischer [6], Pieter Van Vlierberghe [1,2], Stephen S. Roberts[4], Johannes Schulte [7], Sara Ek [8], Rogier Versteeg [9], Jan Koster [9], Johan van Nes[9], Mark Zimmerman [3], Katleen De Preter [1,2,11] & Frank Speleman [1,2,11] ✉

The pediatric extra-cranial tumor neuroblastoma displays a low mutational burden while recurrent copy number alterations are present in most high-risk cases. Here, we identify SOX11 as a dependency transcription factor in adrenergic neuroblastoma based on recurrent chromosome 2p focal gains and amplifications, specific expression in the normal sympatho-adrenal lineage and adrenergic neuroblastoma, regulation by multiple adrenergic specific (super-)enhancers and strong dependency on high *SOX11* expression in adrenergic neuroblastomas. SOX11 regulated direct targets include genes implicated in epigenetic control, cytoskeleton and neurodevelopment. Most notably, SOX11 controls chromatin regulatory complexes, including 10 SWI/SNF core components among which *SMARCC1, SMARCA4/BRG1* and *ARID1A*. Additionally, the histone deacetylase *HDAC2*, PRC1 complex component *CBX2*, chromatin-modifying enzyme *KDM1A/LSD1* and pioneer factor *c-MYB* are regulated by SOX11. Finally, SOX11 is identified as a core transcription factor of the core regulatory circuitry (CRC) in adrenergic high-risk neuroblastoma with a potential role as epigenetic master regulator upstream of the CRC.

Neuroblastoma (NB) is the most common extra-cranial solid childhood cancer, originating from the developing sympatho-adrenergic nervous system[1]. The genomic landscape of NB is characterized by a low mutational burden and highly recurrent structural rearrangements. NB is considered a developmental disorder that is controlled by the complex interplay of multiple transcription factors (TFs) and reshaping of epigenetic landscapes[1]. Tumor cells can co-opt normal developmental pathways for functions that are linked to tumor progression and may become addicted to survival mechanisms controlled by developmental master transcription factors[2]. Recent studies in NB revealed two distinct super-enhancer-associated differentiation states, i.e. adrenergic (ADRN) and early neural crest/mesenchymal (MES), each programmed by a specific core regulatory circuitry (CRC) defined by multiple lineage-specific transcription factors[3,4]. Furthermore, lineage identity switching and plasticity is an emerging key factor in therapy resistance of several cancers. Therefore, further insights into the nature and contribution of master transcription factors may be important to understand frequently occurring relapses in high-risk

---

NBs[5]. Given that (1) in other tumors oncogenic master transcription factors were overexpressed through amplification[5] and that (2) in addition to frequent *MYCN* amplification also other oncogenic (co-)drivers were found to be amplified in NB[1], we aim to identify master TFs implicated in NB by delineating rare focal copy number gain and amplification events.

The SRY-related HMG-box transcription factor 11 (SOX11) belongs to the SOX family of proteins, which are critical regulators of many developmental processes, including neurogenesis[6]. These TFs bind and bend the minor groove of the DNA using their highly similar high mobility group (HMG) domains. SOX11 belongs to the SoxC subgroup, which also includes SOX4 and SOX12, and the expression of these proteins is of key importance for the survival and development of the neural crest, multipotent neural and mesenchymal progenitors, and the sympathetic nervous system[7,8]. In addition to its presumed canonical transcription factor activity, SOX11 was recently also shown to have pioneering activity and thus can be assumed to direct chromatin accessibility at loci controlling cell fates[9].

Here, we identify *SOX11* as the sole protein coding gene residing in the shortest region of overlap for amplicons at chromosome 2p distal to *MYCN* and pinpoint SOX11 as an important transcription factor implicated in adrenergic NB development. In this work, we show that SOX11 acts as a master transcription and dependency factor in adrenergic NB cells. We identify and validate SOX11 target genes which includes genes involved in epigenetic control, cytoskeleton and neurodevelopment. Notably, (1) SOX11 directly regulates 10 SWI/SNF core components and subunit encoding genes, including *SMARCC1*, *SMARCA4* and *ARID1A*, (2) is identified as an early expressed transcription factor of the adrenergic CRC in adrenergic high-risk neuroblastoma and (3) impacts on the adrenergic or mesenchymal transcriptional cell identity but does not induce full phenotypic conversion. We propose SOX11 as epigenetic master regulator upstream of the core regulatory circuitry involved in co-initiation or establishment and/or maintenance of the adrenergic neuroblastoma core regulatory circuit and cell identity.

## Results

### Rare focal amplifications and lineage-specific expression of *SOX11* in neuroblastoma

To identify lineage dependency TFs implicated in NB, we reanalysed DNA copy number profiling data of 556 high-risk primary NB tumors[10] together with those from 263 additional published and 223 unpublished NB tumors[11,12] and 39 neuroblastoma cell lines. We specifically searched for focal gains and/or amplifications of chromosomal segments encompassing TFs with a putative or known role in normal (neuronal) development. Within the 270 kb shortest region of overlap of the commonly gained large chromosome 2 p segment (31% of cases), we identified focal amplifications (three primary NB cases with log2 ratio >2) and high-level gains (two primary NB cases and one cell line with log2 ratio >0.5) encompassing the transcription factor *SOX11* as the only protein coding gene (Fig. 1a). Furthermore, all tumors showing *SOX11* focal amplification or high-level gain were also *MYCN* amplified (Supplementary Fig. 1a). FISH analysis could be analyzed in two tumors and showed that *SOX11* and *MYCN* reside in two independently amplified segments (Fig. 1b). *SOX11* mRNA expression levels were found to be elevated in primary NB tumors with higher *SOX11* copy numbers (*p*-value = 1.82e-09, *t*-test) (*n* = 276) (Fig. 1c). Next, we observed that high *SOX11* mRNA expression levels (fourth quartile) are significantly related to worse overall and progression free survival in two independent NB cohorts of 276 and 498 patients (Fig. 1d, Supplementary Fig. 1b). In addition, SOX11 immunohistochemical analysis using two independent SOX11 antibodies (Supplementary Fig. 1c) showed that high SOX11 protein expression levels were associated with worse overall survival in a cohort of 68 cases consisting of 11 *MYCN*-amplified (MNA) and 57 *MYCN* non-amplified (MNoA) cases (Fig. 1e, Supplementary Data 1). *SOX11* is significantly higher expressed in high-risk *MYCN* amplified tumors as compared to high-risk *MYCN* non-amplified tumors and low risk tumors (Supplementary Fig. 1d). Analysis of *SOX11* tissue specific expression patterns in R2 platform (http://r2amc.nl) and the Cancer Cell Line Encyclopedia (CCLE) showed the highest *SOX11* mRNA expression levels, highest copy number ratio and lowest methylation levels in NB tumors and cell lines as compared to other entities (Fig. 1f, Supplementary Fig. 1e, f). Lineage restricted expression was evident from high mRNA expression levels in human fetal neuroblasts and in sympathetic neuronal lineages during early development as compared to normal cortex from the adrenal gland (Supplementary Fig. 1g) and temporal increase of *SOX11* expression was also noted in mouse NB tumor models in early hyperplastic lesions and full-blown tumors as compared to normal adrenal gland (Supplementary Fig. 1h, i). Higher expression levels of *SOX11* both at mRNA and protein level were observed in MNA and MNoA NB tumors as compared to tumors with mesenchymal super-enhancer signature (Fig. 1g) as well as adrenergic compared to mesenchymal NB cell lines and tumors (Supplementary Fig. 1j–l). Taken together, we identified recurrent focal copy number alterations of the *SOX11* locus in *MYCN* amplified tumors and adrenergic lineage-specific *SOX11* expression levels that are associated with poor prognosis in NB patients.

### *SOX11* is flanked by multiple *cis*-interacting adrenergic specific enhancers

Master transcription factors implicated in defining cell lineage and identity are typically under the control of super-enhancers (SE)[2,13]. *SOX11* was previously identified as a super-enhancer-associated transcription factor in adrenergic NB cell lines[3,14]. Gartlgruber et al. reported super-enhancers in comprehensive published dataset of 60 NB tumors and 25 cell lines[14] and identified one consensus *SOX11* super-enhancer (present in at least 2 samples, not overlapping with H3K4me3 and 5 kb away from a transcription start site), in the adrenergic NB subtype (*MYCN* amplified, high-risk *MYCN* non-amplified and low-risk *MYCN* non-amplified group), while absent or strongly attenuated in the mesenchymal super-enhancer defined group, both in cell lines (Fig. 2a and b, Supplementary Figure 2a–c) and tumors (Fig. 2a and c). Upon more detailed analysis, we observed a large (1.1 Mb) gene desert without protein coding genes marked by multiple H3K27ac peaks, distal to the 3' end of the *SOX11* locus, indicative of the presence of (super-)enhancer activity (Supplementary Fig. 2a, b). In keeping with the presumed gene regulatory activity of this super-enhancer region, the super-enhancer signal is correlated with *SOX11* expression both in NB cell lines (r = 0.774, p = 8.39e-5, Fig. 2d) and tumors (r = 0.778, p = 1.25e-10, Fig. 2d), supporting a functional interaction between this enhancer region and *SOX11* transcriptional regulation. In concordance with the absence of *SOX11* expression in the non-malignant neural crest cell lines (P4 and P5) and the mesenchymal/neural crest like NB cell lines (SH-EP, HD-N-33 and SK-N-AS, GI-ME-N), H3K27ac enhancer peaks were absent in the gene desert distal to *SOX11* (Supplementary Fig. 2c). To provide further physical evidence for looping and contact of the cell type-specific enhancers with the promoter of *SOX11*, we performed 4C-seq analysis for the *SOX11* locus in CLB-GA, KELLY (adrenergic MNoA and MNA cell line respectively with multiple *SOX11* downstream enhancers), SH-EP and SK-N-AS (mesenchymal) NB cell lines and observed looping in this highly active region between the downstream enhancer loci with the *SOX11* promoter in the adrenergic cell lines CLB-GA and KELLY while this interaction was not detectable in the mesenchymal cell lines SH-EP and SK-N-AS (Supplementary Fig. 2d). In support of our findings, interaction of the consensus super-enhancer with the *SOX11* promoter in adrenergic NB cells KCNR was found by Banerjee et al.[15] using Hi-C analyses. Moreover, targeting of this super-enhancer using CRISPR interference caused attenuated *SOX11* expression. In summary, multiple adrenergic specific enhancers and a consensus SE are flanking the *SOX11* locus with multiple independent data supporting its role in *SOX11* regulation.

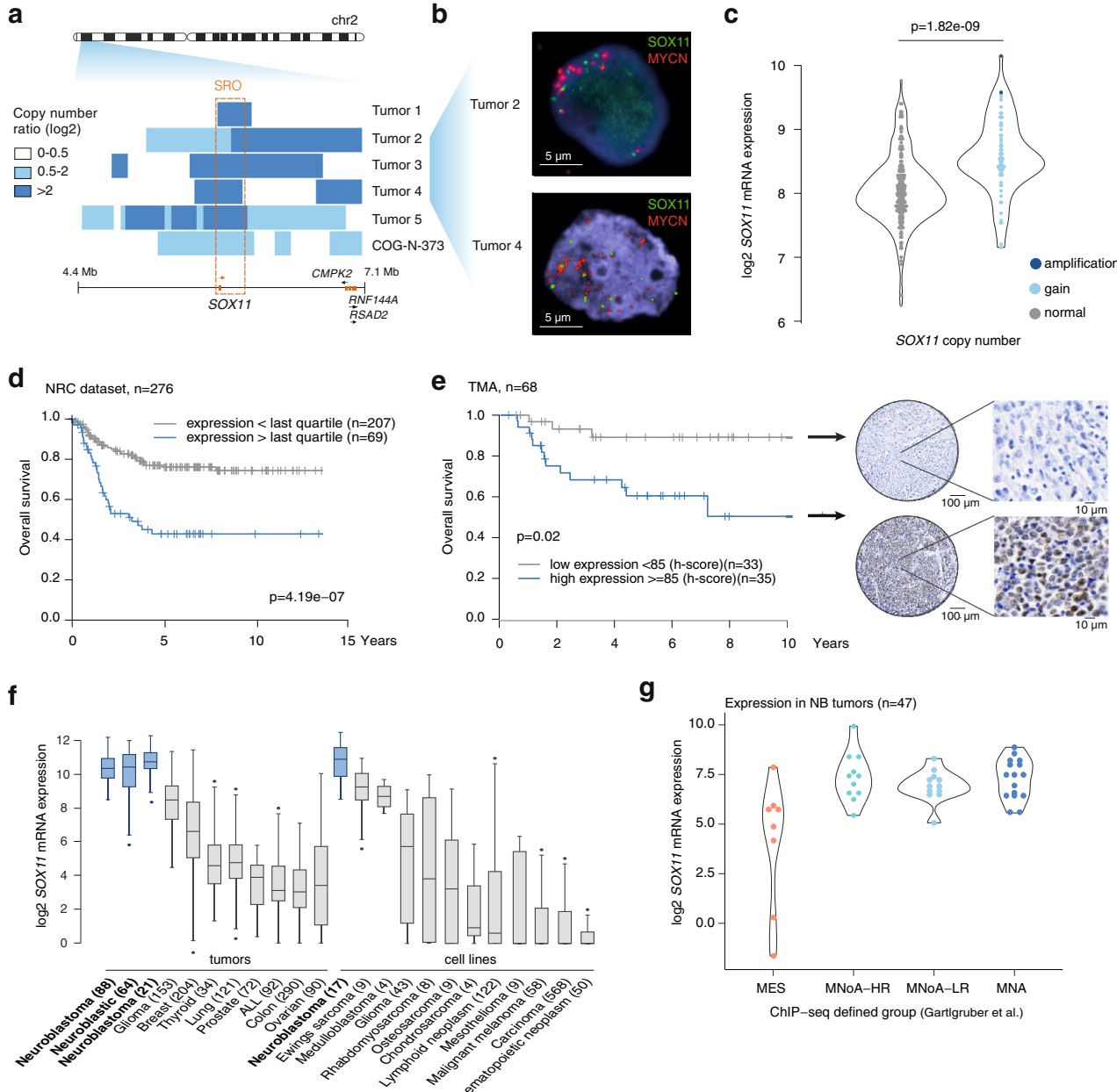

**Fig. 1 | Rare focal amplifications and lineage-specific expression of *SOX11* in NB.**
**a** Log2 copy number ratio on chromosome 2p (4.4–7.1 Mb, hg19) showing shortest region of overlap (SRO) (chr2:5,828,671-6,098,736; hg19) for high-level focal 2p gains (log2ratio > 0.5) and amplification (log2ratio > 2) in 5 NB tumors and 1 NB cell line encompassing the *SOX11* locus (GSE103123[10]). Tumor 1 is evaluated by arrayCGH, tumor 5 by whole genome sequencing, and tumor 2, 3, 4 and the cell line by shallow whole genome sequencing. **b** FISH analysis performed on two tumor cases with *SOX11* amplification from Fig. 1a (asterix) showing independent amplification of *SOX11* (green) and *MYCN* (red). Scalebar represents 5 μm. **c** *SOX11* (log2) mRNA levels according to copy number status (amplification (log2ratio > 2), gain (log2ratio > 0.3); normal) in cohort of 276 patients (NRC cohort, GSE85047)[17] (two-tailed *T*-test, *p* = 1.82e-09). The samples with *SOX11* amplification are tumor 4 and 5 from Fig. 1a. **d** Kaplan–Meier analysis (overall survival) of 276 neuroblastoma patients (NRC cohort, GSE85047)[17] with high (69) or low (207) *SOX11* (log2) expression (highest quartile cut-off) (*p* = 4.19e-07, Kaplan–Meier).
**e** Immunohistochemical staining for SOX11 on tissue micro-array (TMA) of 68 NB

tumors[52] and correlation of high (35) or low (33) SOX11 protein levels (median cut-off of H-score = 85, see Methods) with overall worse survival (*p* = 0.02, Kaplan–Meier). For each group, a representative immunohistochemical staining is depicted. Scalebar represents 10 μm. **f** *SOX11* expression (log2) in NB tumors and cell lines (blue) compared to other entities (details see DATA availability). Boxplots show 1st quartile to 3rd quartile and median. Whiskers represent outer two quartiles maximized at 1.5 times the size of the box. If values outside of the whiskers are present, this is indicated with a single dot. Indicated in brackets are number of replicates per entity. **g** *SOX11* (log2) expression in four H3K27ac profiling based groups identified in NB tumors: *MYCN*-amplified (MNA), high-risk *MYCN* non-amplified (MNoA-HR), low-risk *MYCN* non-amplified group (MNoA-LR) and mesenchymal (MES). *SOX11* is higher expressed in MNoA-HR, MNoA-LR and MNA groups as compared to MES group (ANOVA and two-tailed post-tukey test, significant comparisons: MNoA-HR vs MES *p* = 4e-04, MNoA-LR vs MES *p* = 2e-03, MNA vs MES *p* = 2e-04) (GSE136209)[14]. For Fig. 1c–g, source data are provided as Source Data file.

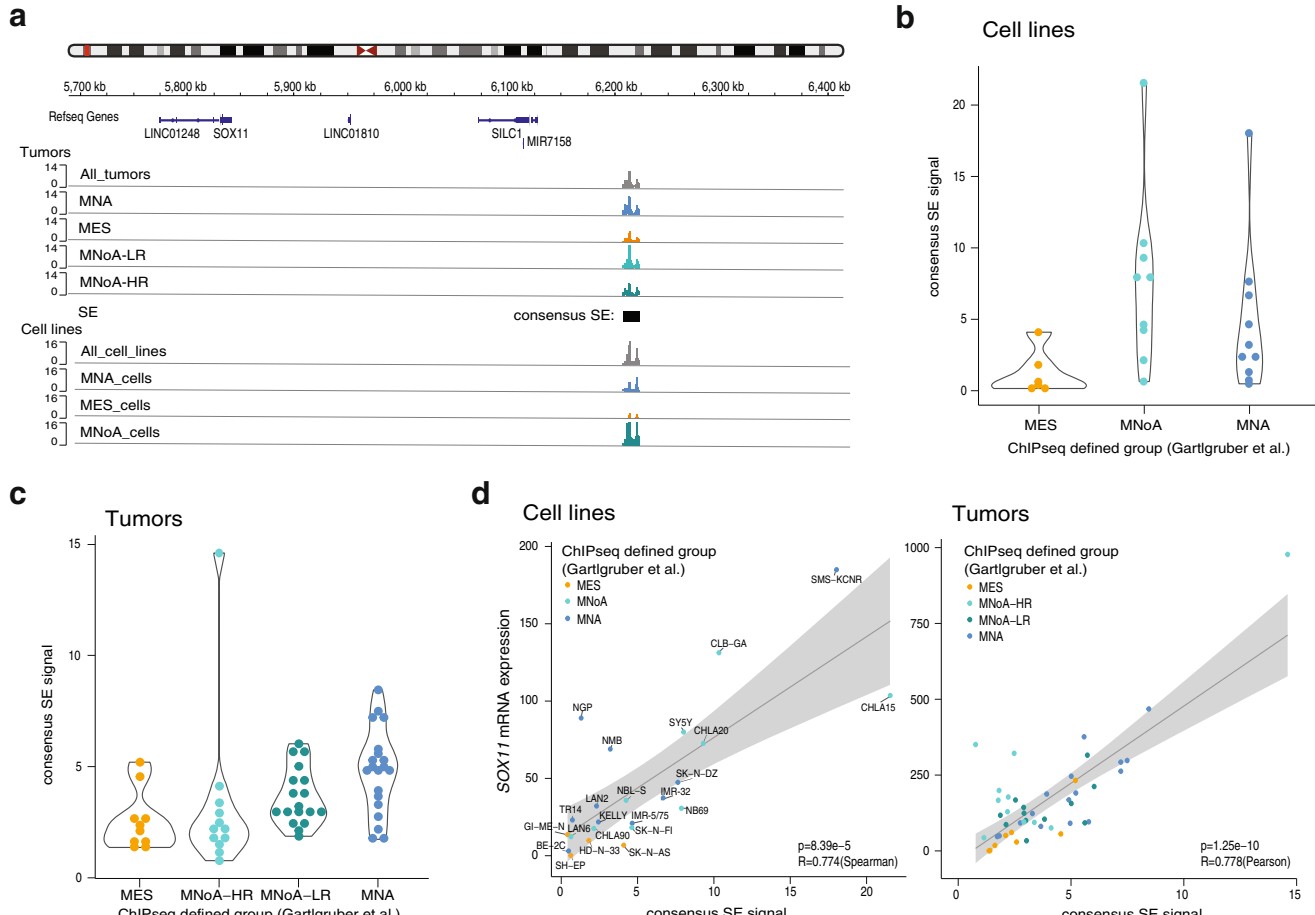

**Fig. 2 | *SOX11* is flanked by multiple *cis*-interacting adrenergic specific enhancers. a** Super-enhancer calling (present in at least 2 samples and not overlapping with H3K4me3 5 kb from transcription start site) downstream of the *SOX11* locus in NB tumors (MNA = adrenergic *MYCN* amplified, MES = mesenchymal, MNoA-LR = adrenergic *MYCN* non-amplified low-risk, MNoA-HR = adrenergic *MYCN* non-amplified high-risk) and NB cell lines (MNA = adrenergic *MYCN* amplified, MES = mesenchymal, MNoA = adrenergic *MYCN* non-amplified) (GSE136209)[14]. **b** Violin plot showing super-enhancer signal for each individual NB cell line colored by their ChIP-seq signature defined subgroup (GSE136209)[14]. **c** Violin plot showing super-enhancer signal for each individual NB tumors colored by their ChIP-seq signature

defined subgroup (GSE136209)[14]. **d** (Left) Correlation of *SOX11* expression with super-enhancer signal of the consensus super-enhancer in a dataset of 25 NB cell lines, colored by their ChIP-seq signature defined subgroup (*p*-value = 8.39e-5, *R*-value = 0.774, two-tailed Spearman correlation). (Right) Correlation of *SOX11* expression with super-enhancer signal of the consensus super-enhancer in a dataset of 47 NB tumors, colored by their ChIP-seq signature defined subgroup (*p*-value = 1.25e-10, *R*-value = 0.778, two-tailed Pearson correlation) (GSE136209)[14]. Trend line is shown with 95% confidence interval. For Fig. 2b–d, source data are provided as Source Data file.

## SOX11 is a dependency factor in adrenergic NB cells

In a next step, we investigated whether adrenergic NB cells are dependent on *SOX11* expression for growth and survival as previously noted for *MYCN* and CRC members. According to the publicly available CRISPR screen data in 1086 cell lines (CRISPR 22Q2 Chronos, available via the DepMap Portal)[16], *SOX11* is identified as a strongly selective gene with dependency in 25 NB cell lines and significantly selective for NB (*p* = 2.5e-40) and more specifically in 11 *MYCN* amplified NB cell lines (*p* = 7.4e29, Supplementary Fig. 3a). We further assessed the phenotypic effects of *SOX11* knockdown in adrenergic NB cell lines, including two *MYCN* amplified cell lines (NGP and IMR-32) and two *MYCN* non-amplified cell lines with high *MYC* activity (SK-N-AS) or *hTERT* activation (CLB-GA), using RNA interference knockdown experiments (siRNAs and/or shRNAs) (Fig. 3a, b). Transient siRNA mediated knockdown of *SOX11* for 48 h in NGP, SK-N-AS and CLB-GA cells resulted in the expected decreased number of colonies, as compared to transfected control cells (Fig. 3c). Concomitantly, long-term phenotype assessment after *SOX11* knockdown in the NGP, CLB-GA and IMR-32 cell lines, using the two most efficient shRNAs (sh3 and sh4), induced a significant G0/G1 growth arrest and reduction of proliferation (Fig. 3d, e).

Our results confirm the CRISPR screen predicted strong lineage dependency role for SOX11 in the adrenergic NB cell line models we tested, both *MYCN* amplified and non-amplified.

## The SOX11 regulated transcriptome is involved in epigenetic control, cytoskeleton and neurodevelopment

To identify the key SOX11 regulated factors and pathways contributing to the adrenergic NB cell phenotype, we performed global transcriptome analysis upon transient siRNA-mediated knockdown of *SOX11* in IMR-32, CLB-GA and NGP cells after 48 h (Supplementary Fig. 4a, Supplementary Data 2).

In support of robustness and biological relevance of our experiments, we noted a significant overlap in both commonly down (*n* = 310, adj. *p*val < 0.05) and upregulated genes (*n* = 71, adj. *p*val < 0.05) across the three NB cell lines (Supplementary Fig. 4b). To further validate these findings and to filter out transcriptional bystanders, we performed an orthogonal experiment using *SOX11* inducible SH-EP cells, which under control conditions do not express *SOX11*, and obtained transcriptome data at 9 h (SOX11 early regulated genes) and 48 h (SOX11 late regulated genes) after *SOX11* induction (Fig. 4a, Supplementary Fig. 4a and c, Supplementary

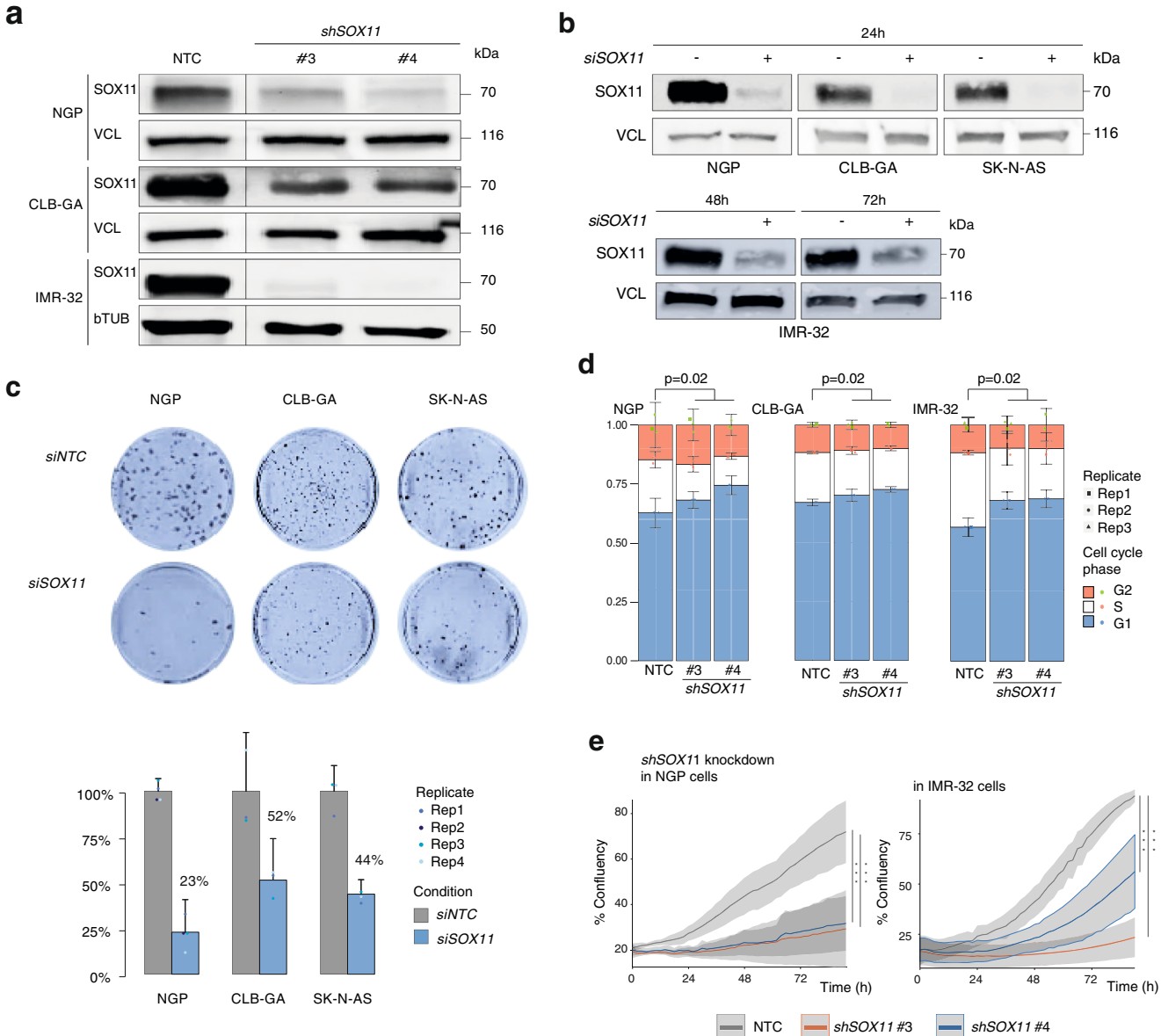

**Fig. 3 | SOX11 is a dependency factor in adrenergic NB cells. a** SOX11 protein levels 6 days upon shSOX11 treatment in NGP, CLB-GA and IMR-32 cells with 2 different shRNAs and one non-targeting control (NTC). Vinculin (VCL) and β-tubulin (bTUB) were used as loading control. NTC and shRNA samples were run on the same blot. Blots for NGP and CLB-GA have been repeated three time with similar results. Blot for IMR-32 has been repeated once. **b** SOX11 protein levels 24 h upon siRNA treatment in NGP, CLB-GA and SK-N-AS cells (dharmafect transfection). SOX11 protein levels 48 h and 72 h upon si*SOX11* treatment in IMR-32 (nucleofection). Vinculin (VCL) is used as loading control. Blot for IMR-32 for 48 h has been repeated four times with similar results, rest once. **c** Reduction in colony formation capacity for NGP, CLB-GA and SK-N-AS cells, 14 days upon siRNA *SOX11* treatment (dharmafect transfection) as compared to non-targeting control (siNTC). Data were generated in triplicate for each cell line, and quantification was done using ImageJ. Data-points were mean-centered and scaled to the *siNTC* condition. Barplot

represents the mean for each condition with error bars representing the standard deviation of the three biological replicates. **d** Cell cycle analysis 6 days upon shRNA *SOX11* treatment in the IMR-32, NGP and CLB-GA cell line. G1 cell cycle arrest upon *SOX11* knockdown as compared to the non-targeting control (NTC). The two-tailed Mann–Whitney statistical test is based on G1 phase percentage (*shSOX11* vs NTC: NGP *p*-value = 0.2, CLB-GA *p*-value = 0.2, IMR-32 *p*-value = 0.2). Data-points were mean-centered and auto scaled. Error bars represent the 95% CI of three biological replicates (rep) for every cell line. **e** Reduced proliferation (% confluence) over time upon prolonged *SOX11* knockdown with 2 different shRNAs for 4 days as compared to non-targeting control (NTC) in NGP and IMR-32 cells. Trend line represents the mean and error marks represent the 95% CI of 3 biological replicates. ANOVA test followed by Tukey post-hoc test (NGP: sh3 vs NTC *p* = 4.51e-14, sh4 vs NTC *p* = 5.06e-14; IMR-32: sh3 vs NTC *p* = 2.16e-14, sh4 vs NTC *p* = 2.74e-06). For Fig. 3a–e, source data are provided as Source Data file.

Data 2). First, in support of *SOX11* knockdown and overexpression specificity, we observed enrichment of predicted SOX11 binding sites in the promoters of differentially expressed genes (Supplementary Fig. 4d). Second, intersection analysis yields 114 common SOX11 early regulated genes and 187 common SOX11 late regulated genes upon comparison of *SOX11* knockdown versus SOX11 early (9 h) and late (48 h) regulated genes, respectively (Fig. 4b). Third, we determined which of these genes showed significant correlation

with *SOX11* expression in 2 independent NB tumor cohorts (GSE85047[17] and GSE45547[18]) and narrowed down the SOX11 early and late regulated gene list and established a SOX11 early (56 genes) and late (68 genes) gene signature (Fig. 4c, Supplementary Data 3). Genes in the early SOX11 gene signature are predominantly down-regulated upon *SOX11* knockdown and upregulated upon *SOX11* overexpression, suggesting that SOX11 acts mainly as transcriptional activator.

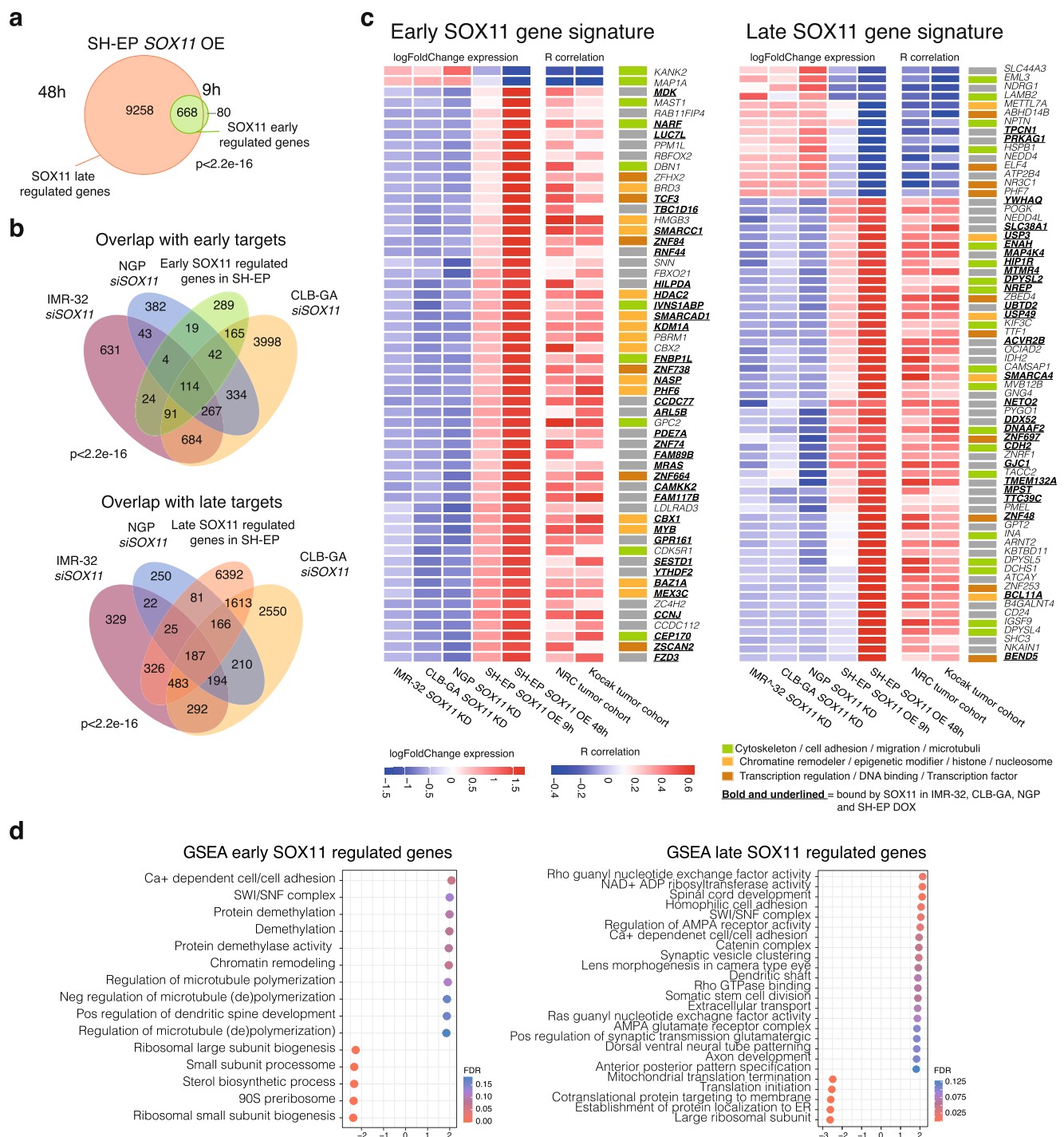

**Fig. 4 | The SOX11 regulated transcriptome is involved in epigenetic control, cytoskeleton and neurodevelopment. a** Overlap of genes perturbed in SH-EP upon *SOX11* overexpression (OE) for 48 h and 9 h (adj *p* value <0.05) with differential genes upon 9 h overexpression being the SOX11 early regulated genes and differential genes upon 48 h but not 9 h overexpression being the SOX11 late regulated genes (one-tailed fisher test *p*-value <2.2e⁻¹⁶). **b** Overlap of genes perturbed in IMR-32, CLB-GA and NGP upon *SOX11* knockdown for 48 h (adj *p* value <0.05) with respectively the SOX11 early and late regulated genes in SH-EP (one-tailed fisher test *p*-value <2.2e⁻¹⁶). **c** SOX11 early and late genes signature obtained by the overlap of differentially expressed genes upon *SOX11* knockdown in IMR-32,

CLB-GA and NGP with the early and late SOX11 regulated genes in SH-EP (adj *p* value <0.05, log FC > 0.5 or < −0.5) as well as with genes correlation with *SOX11* expression in 2 different NB tumor cohorts (NRC GSE85047[17], Kocak GSE45547[18], *p*-value <0.05). A color next to each gene represents the involved pathways. Bold and underlined represents genes that are bound by SOX11 in IMR-32, CLB-GA, NGP and SH-EP after *SOX11* overexpression for 48 h. **d** Top enriched genesets after doing GSEA analysis (http://www.gsea-msigdb.org/gsea/index.jsp, ontology gene sets C5) for SOX11 early and SOX11 late regulated genes. Depicted is the normalized enrichment score (NES, x-axis) and the false discovery rate (FDR, color). For Fig. 4c–d source data are provided as Source Data file.

Using these gene sets, we then sought for enrichment for cellular functions to gain insight into the presumed contribution of *SOX11* expression to the high-risk NB adrenergic phenotype. Gene set enrichment analysis for the SOX11 late regulated genes revealed strong enrichment for axon outgrowth, neural crest cell migration and cytoskeleton (Fig. 4c, d, S4e). In line with this, Afanasyeva et al. demonstrated decreased migration upon *SOX11* knockdown in IMR-32 spheroids, further showing that *SOX11* knockdown fosters morphological asymmetric cell divisions and causes reprogramming of nucleokinesis migration in ADRN-type NB cells[19]. To functionally validate the predicted role in control of cytoskeleton and cell migration by our *SOX11* overexpression RNA-sequencing data, we performed wound-healing assays confirming that *SOX11* overexpression enhances wound healing capacity and migration potential in SH-EP NB cells (Supplementary Fig. 3b, c).

In accordance with the putative role of SOX11 in adrenergic NB cell identity, we observed enrichment of genes of the proneuronal subtype in glioblastoma and adrenergic subtype in NB amongst the upregulated genes upon *SOX11* overexpression and downregulated genes upon *SOX11* knockdown and vice versa is true for gene sets of the mesenchymal subtype[20]. Most notably, we already observed an enrichment of the adrenergic NB gene sets in the early SOX11 regulated genes suggesting a possible direct role of SOX11 in maintenance of cell identity (Supplementary Fig. 4f).

Of further interest, one of the top regulated genes across the different data sets is midkine (MDK), which could, at least partly, explain the role for SOX11 in neurodevelopment[6]. Midkine has been implicated in the regulation of neural stem cells and embryonic central nervous system development and was proposed to act through phosphorylation and activation of the ALK receptor tyrosine kinase[21], known to be mutated and amplified in NB.

In order to better understand how SOX11 controls these phenotypes through its early regulated genes, we looked into top enriched gene sets in the early regulated genes which included "SWI/SNF complex" and "chromatin remodeling". This is supported by differential early regulation of several SWI/SNF components such as BAF core component *SMARCC1/BAF155* and pBAF specific *PBRM1/BAF180*. Further, other epigenetic regulators were found including the histone deacetylase *HDAC2*, the *PHF6* NurD component, the H3K27me3 reader and canonical PRC1 complex component *CBX2*, the chromatin-modifying enzyme lysine-specific demethylase 1 *KDM1A/LSD1* and pioneer transcription factor *c-MYB*, which were all also identified as direct SOX11 targets (see further) (Fig. 4c–d, S4e and g, Fig. 5e). Interestingly, when investigating the impact of SOX11 on all 29 known SWI/SNF components of the three known SWI/SNF complexes (c-BAF, nc-BAF, p-BAF), we observed that 20 out of 29 known SWI/SNF components are differentially upregulated after *SOX11* overexpression, 16 of which are also differentially downregulated after *SOX11* knockdown in at least one NB cell line. Moreover, 13 components are already differentially upregulated at the 9 h time point of *SOX11* overexpression, of which 10 are directly bound by SOX11 (see further) (Fig. 4f). This strongly suggests a direct and early role of SOX11 on the expression of the SWI/SNF complex.

Taken together, the SOX11 controlled transcriptome revealed multiple functions related to cell migration, cytoskeleton, neuronal differentiation and a broad regulatory effect on multiple epigenetic regulatory factors and protein complexes.

## SOX11 directly regulates multiple major modulators of the epigenome including the SWI/SNF remodeling complex

To define in more detail the SOX11 DNA bound direct target genes, we performed CUT&RUN-sequencing using the atlas SOX11 antibody (HPA000536) in the adrenergic *MYCN* amplified NB cell lines IMR-32 and NGP and *MYCN* non-amplified cell line CLB-GA. We also evaluated DNA binding sites upon forced overexpression of *SOX11* in

mesenchymal SH-EP NB cells for which parental cells do not express *SOX11* (Supplementary Data 4). A comparative analysis of identified targets revealed a significant overlap for common direct binding sites for these three adrenergic NB cell lines as well a significant overlap with common targets for SOX11 DNA binding sites in SH-EP cells after overexpression of *SOX11* (Fig. 5a and Supplementary Fig. 5a). Taken together, these data provide support for the validity of the identified targets as SOX11 regulated genes in NB cells. Moreover, the forced overexpression in SH-EP cells indicates that SOX11 can bind similar genes in NB cells with mesenchymal cell identity which otherwise exhibit no *SOX11* expression. To further support the specificity of SOX11 binding, we performed DNA binding motif analysis and observed enrichment for a de novo SOX motif in the SOX11 bound sites (Supplementary Data 5). Furthermore, overlap was observed for SOX11 binding sites identified by CUT&RUN using the atlas SOX11 antibody (HPA000536) and ChIP-sequencing using the in-house SOX11-PAb antibody as additional validation (Supplementary Fig. 5a, Fisher test *p*-value<2.2e$^{-16}$). Of further note, SOX11 peaks were significantly enriched for H3K27ac marks for active chromatin, H3K4me3 promoter marks and open chromatin (ATAC-seq), consistent with binding of SOX11 to both proximal and distal active transcriptional regulatory regions of both protein coding and non-coding genes (Supplementary Fig. 5b, c). We further validated SOX11 DNA bound targets through correlation analysis for the established early and late SOX11 gene signatures (Fig. 4c). Using transcriptome datasets for a panel of 29 NB cell lines and a primary NB tumor dataset (NRC, GSE85047)[17], we furthermore showed strong overlap between SOX11 CUT&RUN activity score, the above transcriptome derived SOX11 signatures, *SOX11* expression levels and NB patient survival outcome (Supplementary Fig. 5d–f).

To identify putative functional SOX11 regulated direct targets, we filtered for genes marked by SOX11 promotor binding in combination with significant altered expression levels after *SOX11* overexpression in the neuroblastoma cell line SH-EP, both after 9 h (early targets) and 48 h (late targets). This yielded 304 early direct targets (239 upregulated and 65 downregulated) and 2165 late direct targets (939 upregulated and 1226 downregulated) from the total of 3984 commonly identified SOX11 binding sites identified across the different cell lines (Fig. 5c). Strong enrichment of SOX11 binding was preferentially found in genes correlated with *SOX11* expression in NB tumors (GSE85047[17], Fisher test *p*-value<2.2e$^{-16}$), further supporting that SOX11 acts mainly as transcriptional activator (Fig. 5c). Using this SOX11 DNA binding analysis and selection of early direct targets, we further verified putative SOX11 regulated cellular functions and confirmed chromatin remodeling and transcriptional regulation as the most significantly enriched pathways as determined by EnrichR (Fig. 5d, S5h). As indicated above, a total of 10 SWI/SNF components are directly regulated by SOX11 including core components and subunit specific encoding genes, including *SMARCC1*, *SMARCA4* and *ARID1A* (Fig. 4f, Fig. 5e, Supplementary Fig. 5d). In addition, other constituents or actors in epigenetic regulatory processes were also SOX11 bound and regulated including; (1) *CBX2*, a H3K27me3 reader of the canonical PRC1 complex; (2) *KDM1A* (*LSD1*), a lysine-specific demethylase 1; (3) the histone deacetylase *HDAC2* and (4) demethylase and chromatin regulator *TET1*. Of further interest, *c-MYB*, a known oncogene and pioneering factor, is marked by strong regulation upon forced *SOX11* induction in SH-EP cells and knockdown in adrenergic NB cells (Fig. 4c, Fig. 5e, Supplementary Fig. 5d). In addition to the transcriptional regulation, SOX11 controlled regulation at protein levels was also confirmed for c-MYB, SMARCC1 and SMARCA4 (Supplementary Fig. 5h–i).

In order to gain initial insights into the impact of the broad upregulation by SOX11 of most SWI/SNF complex components, we furthermore analysed SMARCA4 binding sites in SOX11 regulated genes and observed enrichment for SMARCA4 DNA binding only in

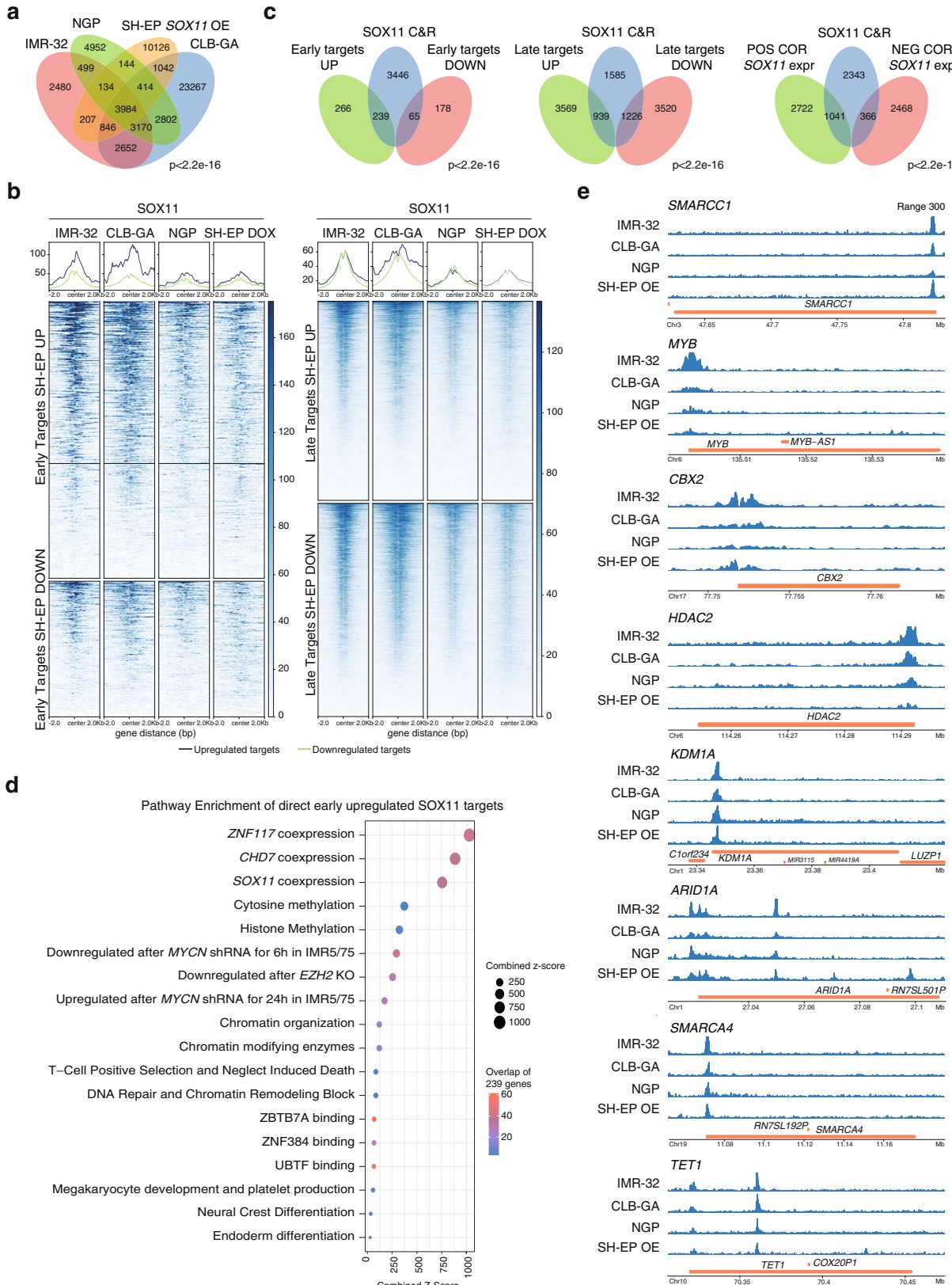

late SOX11 regulated genes (Supplementary Fig. 5j). Although further studies are warranted, this could suggest that enhanced SWI/SNF activity associated with the observed SMARCA4 binding acts downstream of SOX11 to contribute to its induction of a gene regulatory program.

In summary, integration of SOX11 DNA binding data with SOX11-associated expression gene signatures reveals SOX11 targeting of multiple core and subunit specific SWI/SNF components as well as several other proteins implicated in diverse epigenetic regulatory processes.

**Fig. 5 | SOX11 directly regulates multiple major modulators of the epigenome including the SWI/SNF remodeling complex. a** Overlap (min. overlap = 20 bp) of the SOX11 CUT&RUN peaks (MACS2 peakcalling $q$val < 0.05) in IMR-32, CLB-GA, NGP cells and SH-EP cells after *SOX11* overexpression (OE) for 48 h (one-tailed fisher test $p$-value<2.2e-16). **b** Heatmap profiles −2 kb and +2 kb around the transcription start site of early and late SOX11 targets in SH-EP, subdivided in upregulated and downregulated genes. On these regions the SOX11 CUT&RUN data in IMR-32, CLB-GA, NGP and SH-EP cells after *SOX11* overexpression (SH-EP DOX) for 48 h are mapped and ranked according to the sums of the peak scores across all datasets in the heatmap. **c** Overlap of common SOX11 CUN&RUN peaks (common peaks in IMR-32, CLB-GA, NGP cells and SH-EP cells after *SOX11* overexpression for 48 h) and early and late up- and downregulated genes in SH-EP after *SOX11* overexpression for 48 h (adj $p$ value <0.05, log FC > 0.5 or < −0.5) and overlap of common SOX11 CUT&RUN peaks with genes positively (POS COR) and negatively (NEG COR)

correlated with *SOX11* expression in NB tumor cohort ($n$ = 283, GSE85047[17], $p$-value<0.05, one-tailed fisher test $p$-value<2.2e-16). **d** EnrichR analysis[53] for overlap of common SOX11 CUT&RUN peaks (common peaks in IMR-32, CLB-GA, NGP cells and SH-EP cells after *SOX11* overexpression for 48 h) and early upregulated genes in SH-EP after *SOX11* overexpression for 48 h. Depicted is the combined Z-score which is computed by taking the log of the p-value from the Fisher exact test and multiplying that with the z-score of the deviation from the expected rank (size), as well as the number of genes that overlap with the enriched genesets (color). **e**. Binding of SOX11 at c-*MYB*, *SMARCC1*, *CBX2*, *KDM1A*, *SMARCA4*, *HDAC2*, *ARID1A* and *TET1* in IMR-32, CLB-GA, NGP and SH-EP cells after *SOX11* overexpression for 48 h. Signal represents log likelihood ratio for the ChIP signal compared to input signal (RPM normalised). All peaks are called by MACS2 ($q$ < 0.05). For Fig. 5d source data are provided as Source Data file.

## SOX11 is a core regulatory circuitry transcription factor in adrenergic NB

Our SOX11 DNA binding analysis also provided further insight into proposed role of SOX11 as core regulatory circuitry (CRC) master transcription factor. CRCs are a group of interconnected auto-regulating transcription factors that form loops and can be identified by super-enhancers[2,13]. Our data suggest that SOX11 is a adrenergic NB CRC member. First, SOX11 binds its own promotor and binds the *SOX11* 3' downstream enhancer landscape, including the above-mentioned consensus *SOX11* super-enhancer (Fig. 6a). Second, binding of major CRC members including HAND2, PHOX2B and GATA3 was observed at the *SOX11* promoter. Also, HAND2, GATA3, MYCN, ASCL1 and TWIST1 bind the downstream enhancer landscape including the consensus *SOX11* super-enhancer (Fig. 6a). Third, SOX11 binding is observed at the promotors of *HAND2, PHOX2B, GATA3, ASCL1, TWIST1,* and *TCF3* (Fig. 6b). Finally, as reported for other CRC transcription factor members, *SOX11* knockdown causes partial CRC collapse notified by attenuated expression of several CRC members such as *TCF3, ISL1, PHOX2B, TFAP2B, MYCN* and *KLF7*. However full adrenergic-to-mesenchymal transitions and establishment of the mesenchymal CRC is not observed after *SOX11* knockdown (Fig. 6c, Supplementary Fig. 6a, Supplementary Data 2), suggesting knockdown of *SOX11* on its own is not sufficient to induce a lineage switch.

To investigate further the functional connection between SOX11 and the adrenergic CRC, we compared the binding sites of the major adrenergic CRC members to our own SOX11 multi-omics data. Co-binding of MYCN, ASCL1 and TWIST1 at SOX11 bound enhancers and promotors can be observed as well as HAND2, GATA3 and PHOX2B co-binding at SOX11 bound enhancers (Fig. 6d). More specifically, we observe strong co-binding of MYCN, ASCL1 and TWIST1 at transcription start sites of early and late regulated SOX11 targets (Supplementary Fig. 6b). Having established that SOX11 is a member of the adrenergic CRC, we next looked into the dynamic regulation of SOX11 and other CRC members from RNA-sequencing data obtained in a human pluripotent stem cell based differentiation model for developing human sympathoblasts (Van Haver et al., in preparation). *SOX11* was found to be expressed in earlier developmental stages prior to emergence of the adrenergic master regulator PHOX2B and the other CRC members including *HAND2* and *GATA3* (Fig. 6e, Supplementary Fig. 6c).

Taken together, our findings support the notion that SOX11 is a canonical CRC member and plays a distinct role, during early sympathoblast development prior to emergence of the adrenergic master regulator PHOX2B and the other CRC members including HAND2 and GATA3. In conclusion, we postulate that SOX11 mediates establishment and maintenance of the adrenergic core regulatory circuitry by modulating the expression of chromatin remodeling complexes and acting as an epigenetic master regulator upstream of the core regulatory circuitry.

## Discussion

Cellular mechanisms that govern lineage-specific proliferation and survival during development may be co-opted by tumor cells. Consequently, these tumors will be selectively dependent on such lineage factors offering interesting options for targeting these tumor cells whilst sparing normal tissues. We identified *SOX11* as a dependency gene in adrenergic neuroblastoma (NB) which is recurrently affected by large segmental 2 p gains in high-risk NBs as well as recurrent focal gains and amplifications. *SOX11* was identified as the sole protein coding gene residing in the shortest region of overlap at 2 p distal to *MYCN*, suggesting a role as driver for selection of the respective amplicons during tumor formation. *SOX11* is known to exhibit specific expression during normal development, predominantly in the neuronal lineage. In line with the data from the CRISPR screen available through the DepMap portal, our *SOX11* knockdown data support that SOX11 is a dependency factor in NB. Furthermore, higher *SOX11* expression levels were found to be correlated with poor prognosis for NB patients.

To gain insight into the functional contribution to the tumor phenotype, we identified functional SOX11 target genes through genome wide DNA binding analysis combined with transcriptome profiling after *SOX11* knockdown in multiple adrenergic NB cell lines and forced *SOX11* overexpression in mesenchymal SH-EP NB cells. Analysis of the SOX11 regulated transcriptome identified genes implicated in epigenetic control, cytoskeleton and migration, and neurodevelopment. Subsequent identification of early transcriptionally regulated genes and direct DNA bound gene promotors revealed a remarkable enrichment for genes encoding the SWI/SNF chromatin remodeler multiprotein complex, including *SMARCC1, SMARCA4* and *ARID1A*. In addition, several other important epigenetic regulators were noted including chromatin silencing PRC1 complex components and pioneering transcription factor *c-MYB*. While these targets require further individual functional validation, the finding of multiple functional targets implicated in a broad range of essential epigenetic regulatory processes is intriguing. Structural disruption of the BAF complex, achieved by silencing of its essential subunits and direct SOX11 targets *ARID1A* and *ARID1B*, exerted an epigenomic reprogramming in neuroblastoma cells, resulting in reduced neuroblastoma invasiveness and metastasis[22]. In line with this, overexpression of *SMARCA4/BRG1*, the catalytic component of the SWI/SNF complex and early direct target of SOX11, is essential for NB cell viability[23] and therefore, at least partially could explain dependency of adrenergic NB cells for survival. A further independent link between SOX11 and SWI/SNF functionality comes from the observation of disrupted normal (neuronal) development in the intellectual disability Coffin-Siris syndrome[24] which is mostly caused by mutations in SWI/SNF components including *SMARCE1* but has also been reported in patients with *SOX11* germline loss of function.

Additionally, based on integrated analysis of SOX11 occupancy as determined by CUT&RUN and transcriptome analysis upon

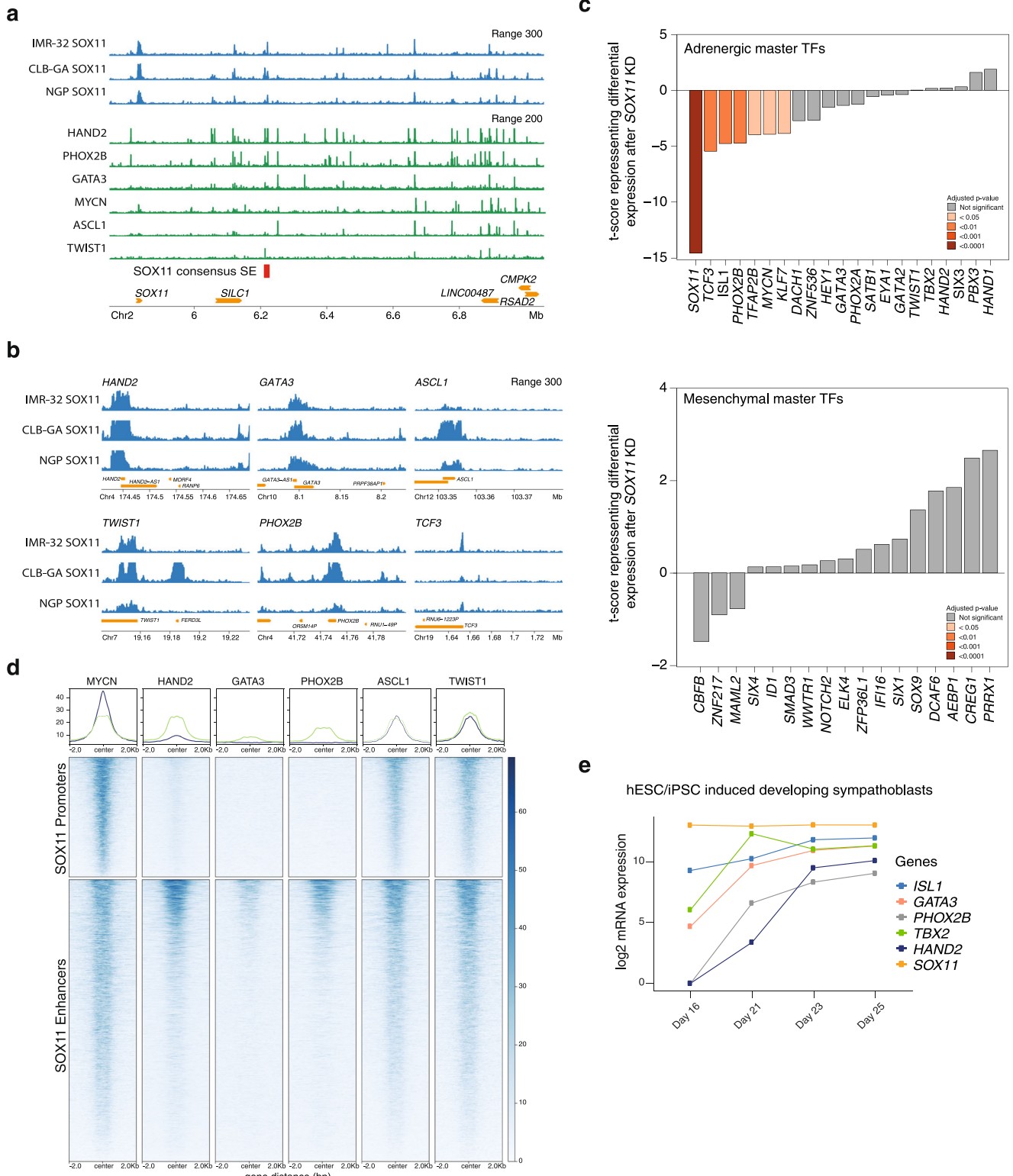

knockdown, we propose SOX11 as a master transcription factor of the recently established adrenergic core regulatory circuitry (CRC) in neuroblastoma[3,4]. In addition to its role as CRC member, SOX11 may act also in an upstream hierarchical function based on gene expression analysis of *SOX10* positive neural crest derived maturing sympathetic adrenergic neuroblasts. While the adrenergic master regulator *PHOX2B* is strongly induced at day 23 of differentiation together with several CRC TFs including *HAND2* and *GATA3*, *SOX11* is clearly expressed much earlier from day 16 on of the differentiation track.

Taken together, these observations, together with the previously established role of SWI/SNF chromatin remodeling in maintenance of lineage-specific[25], we hypothesize that SOX11 allows NB cells to benefit from enhanced SWI/SNF activity and chromatin remodeling to sustain the establishment and maintenance of the adrenergic core regulatory circuitry of these arrested immature transforming sympathoblasts during tumor initiation. We also propose that several SOX11 driven functions are co-opted by the transformed neuroblasts contributing to the aggressive phenotype of high-risk adrenergic NBs. Further support

**Fig. 6 | SOX11 is a core regulatory circuitry transcription factor in adrenergic NB. a** Binding of SOX11 (CUT&RUN), HAND2, PHOX2B, GATA3 (GSE90683)[4], MYCN, TWIST1 (GSE94822)[46] and ASCL1 (GSE159613)[47] and at the *SOX11* locus and downstream enhancer landscape. Signal represents log likelihood ratio for the ChIP and CUT&RUN signal compared to input signal (RPM normalised). Super-enhancers of CLB-GA are annotated using ROSE (red bar) showing the *SOX11* consensus super-enhancer (SE). **b** Binding of SOX11 to the *PHOX2B, HAND2, GATA3, ASCL1, TCF3* and *TWIST1* locus. Signal represents log likelihood ratio for the ChIP and CUT&RUN signal compared to input signal (RPM normalised). **c** Tscore representing differential expression of MES and ADRN core regulatory circuitry transcription factors (TFs) in RNA-sequencing data after *SOX11* knockdown (KD) in IMR-32, CLB-GA and NGP. Significant adjusted *p*-values are indicated with coloured bars. Statistical testing was done using the empirical Bayes quasi-likelihood *F*-test. **d** Heatmap profiles −2 kb and +2 kb around the summit of SOX11 CUT&RUN peaks in IMR-32, grouped for promoters or enhancers (homer annotation). On these regions HAND2, PHOX2B, GATA3 (GSE90683)[4], MYCN, TWIST1 (GSE94822)[46] and ASCL1 (GSE159613)[47] ChIP data is mapped and ranked according to the sums of the peak scores across all datasets in the heatmap. **e** *ISL1, GATA3, PHOX2B, TBX2, HAND2* and *SOX11* (log2) expression during induced differentiation of hPSC cells along the sympatho-adrenal lineage. Expression levels depicted starting from day 16, upon sorting the cells for *SOX10* expression indicating cells committed to truncal neural crest cells, and followed-up during sympatho-adrenal development until day 25. For Fig. 6c and e source data are provided as Source Data file.

for this view comes from the recent description of a unique aggressive transitional cell state important for the inter-transition between adrenergic and mesenchymal cells through single-cell RNA-sequencing analysis of peripheral neuroblastic tumors[26]. Interestingly, *SOX11* is described as a marker gene of the transitional state, amongst *MYCN* and others, while *GATA3* and *HAND2* mark the adrenergic state. This is in line with the fact that *SOX11* was reported to be activated in a so-called proliferative active bridging population (transient cellular state) connecting a progenitor cell type coined Schwann cell precursors and their differentiated counterpart, chromaffin cells[27], while the transitional signature in the above described tumors is enriched in the bridging population. In this context it is intriguing that depletion of direct SOX11 target *ARID1A* promotes partial adrenergic-to-mesenchymal CRC conversion in adrenergic NB cells by regulating enhancer mediated expression through alteration of the binding sites of the SWI/SNF chromatin remodeling complex[28]. Additionally, a recent paper also attributed a crucial role for SOX11 as one of the required factors for fibroblasts-to-neuron conversion in the presence of exogenous reprogramming factors[29]. However, while we observed an enrichment of adrenergic and mesenchymal gene signatures upon *SOX11* overexpression and knockdown respectively, *SOX11* overexpression or knockdown in itself was not sufficient to induce a full transition in cell lineage, at least not at 48 h after induction or knockdown of *SOX11*.

In conclusion, we identify SOX11 as a dependency factor and member of the CRC in adrenergic high-risk NB with a putative function as epigenetic master regulator upstream of the core regulatory circuitry. Further DNA occupancy data for SWI/SNF complex components and CRC members during normal differentiation and upon controlled perturbation in NB cells will shed more light on how this CRC is formed and how it is hijacked during tumor development. Additionally, studies in in vitro cellular models and targeted overexpression to the sympathetic adrenergic lineage in mice or zebrafish as well as developmental studies in animals or hPSC differentiation models are needed to further explore the complex interplay of the broad range of transcription factors in adrenergic neuroblasts.

## Methods
### Samples and cell lines
Copy number analysis was performed on primary untreated NB tumors, representative of all genomic subtypes including 263 and 223 samples[11,12] of the NRC cohort (GSE85047[17]), 556 samples of the NB high-risk cohort (GSE103123[10]), and one unpublished in-house sample. In addition, copy number data of 33 NB cell lines[11] and the cell line COG-N-373 (Fig. 1a) were used. *SOX11* expression analysis was performed on 283 NB tumors for which copy number (*n* = 218), mRNA expression (*n* = 283) and patient survival (*n* = 276) data were available from the Neuroblastoma Research Consortium (NRC, GSE85047[17]), which is a collaboration between several European NB research groups. Additionally, the NB dataset from Su et al. (*n* = 489, GSE45547[18]) was used as validation cohort[30]. For super-enhancer

analysis, the published dataset of 60 NB tumors and 25 cell lines of Westermann et al.[14] was used.

All NB cell lines used in this manuscript (genotype, mutation status and source provided in Supplementary Data 6), were grown in RPMI1640 medium supplemented with 10% foetal bovine serum (FBS), 2 mM L-Glutamine and 100 IU/ml penicillin/streptavidin (referred further to as complete medium) at 37 °C in a 5% CO2 humid atmosphere. Short Tandem Repeat (STR) genotyping was used to validate cell line authenticity prior to performing the described experiments and Mycoplasma testing was done every two months and no mycoplasma was detected.

### High-resolution DNA copy number analysis
DNA was obtained using the QiaAmp DNA Mini kit (Qiagen #51304) according to the manufacturer's instructions and concentration was determined by Nanodrop (Thermo Scientific) measurement. Array comparative genomic hybridisation (arrayCGH) was performed using 105 K (amadid#019015) or 180 K (amadid#023363) Human Genome CGH Microarray slides from Agilent Technologies (Santa Clara, CA, USA) following the manufacturer's protocols. For shallow whole genome sequencing, DNA extraction was performed on 10 μm-thick FFPE tissue sections, using the QIAamp DNA FFPE tissue kit (Qiagen, Hilden, Germany) and protocol and deparaffinization solution. Covaris' adaptive focused acoustics technology and M220 focused ultrasonicator (Covaris, Woburn, MA) were used to prepare fragmented DNA with fragment sizes of 200 bp. DNA libraries were constructed with the NEXTflex rapid DNA-Seq kit and protocol and NEXTflex DNA barcodes (Bioo Scientific, Austin, TX) using 200 ng of fragmented DNA as starting material, size selection, and eight PCR cycles. Cluster generation and sequencing were accomplished by, respectively, a cBot 2 and HiSeq 3000 system (Illumina, Essex, UK). The minimal number of reads (single-end; 50-cycle mode) per sample was intended to be at least 10 million (mean coverage of ~ 0.15 ×).

Copy number data were processed, analysed and visualised using VIVAR[31]. For fluorescent in situ hybridization (FISH), four-micron-thick tissue sections were cut onto positively charged slides. The unstained slides were deparaffinized in xylene and dehydrated in graded alcohols. Cell conditioning was performed in a 1 M sodium thiocyanate water bath at 80 °C for 30 min, followed by a washing step in 2 × saline-sodium citrate (SSC) buffer and an incubation step using proteinase K (Roche, Indianapolis, IN, USA) for 20 min at 37 °C. Probes for for the SOX11 locus (CTD-2037E22) was applied, following heat block denaturation at 80 °C for 5 min, and hybridization at 37 °C for 14 to 18 h. The coverslip was removed by washing in 2 × SSC buffer. Excess probe was eliminated with 0.5 × SSC buffer stringency washes, followed by similar graded stringency washes. The digoxigenin-labeled probes were visualized using fluorescein isothiocyanate-antidigoxigenin (Roche). Using 4′,6-diamidino-2-phenylindole counterstain, nucleated cells were highlighted. A microscope, equipped with a dual-pass filter (Green/Orange; Vysis) and two single-pass filters (Green; Vysis, and Orange, Vysis), was employed to ultimately observe FISH signals. *SOX11* amplifications

and high-level focal gains were identified as copy number segments overlapping with the *SOX11* locus with log2 ratio > = 2 and > = 0.3 respectively and a maximal size of 5 Mb.

## Tissue micro-array

For NB tissue micro-array, 73 formalin-fixed and paraffin-embedded primary untreated neuroblastoma tumors were used (Supplementary Data 1) from which 5-μm sections were made. Antigen retrieval was done in citrate buffer and endogenous peroxidases were blocked with $H_2O_2$ (DAKO). The sections were incubated with primary antibodies (1:300 SOX11-C1 antibody, from Prof. Sara Ek, Lund University[32] = Antibody 1, 1:50 SOX11 antibody from Klinipath (cat#ILM3823-C01) = Antibody 2), followed by incubation with the Dako REALTM EnvisionTM-HRP Rabbit/ Mouse system and substrate development was done with DAB (DAKO). Scanning of the slides was done using the Zeiss Axio Scan.Z1 (Zeiss) and counting of SOX11 positive NB cells was done by H-scoring. In brief, the percentage of SOX11-positive cells is each time multiplied by the intensity (0, 1, 2 of 3) : [1 × (% cells 1 + ) + 2 × (% cells 2 + ) + 3 × (% cells 3 + )]. Blind scoring was done by two independent persons. Each sample was present in triplicate and scores are presented as the average of the three replicates. 15 samples were omitted due to lack of survival data.

## Statistical and transcriptomic analysis of NB cohorts and other entities

Neuroblastoma transcriptomic analysis was performed on a dataset of 283 NB tumors for which copy number ($n = 275$), mRNA expression ($n = 283$) and survival ($n = 276$) data were available from the Neuroblastoma Research Consortium (NRC, GSE85047[17]), which is a collaboration between several European NB research groups. Additionally, the NB dataset from Su et al. ($n = 489$, GSE45547[18]) was used as validation cohort[30]. H3K27ac ChIP-seq data and super-enhancer annotation were public available from Gartlgruber et al. [14]. The Depmap array[16] and R2 platform (http://r2amc.nl) were used as repositories for gene expression and dependency data of different tumor entities.

All statistical analyses (two-sided t-test, Wilcoxon test, kruskal-wallis, ANOVA, post-hoc dun-test or tukey test, Kaplan–Meier, correlation spearman and pearson) were done using R (version 3.6.1) and RStudio (v.1.1.463). For correlation analysis, genes were ranked according to Pearson correlation coefficient.

## 4C-sequencing

The 4 C templates were prepared according to the protocol by Van de Werken et al. [33]. For 4C-sequencing, $10^7$ separated single cells that were cross-linked with 2% formaldehyde for 10 min at room temperature, cells were lysed, and cross-linked DNA was digested was digested using 200U of NlaIII (NEB #R0125S) as primary restriction enzyme. This was followed by proximity ligation using 50U of T4 DNA ligase (Roche #10799009001) and ligated DNA circles were de-cross-links overnight using proteinase K and purified with NucleoMag P-Beads (Macherey-Nagel, Düren, Germany) to obtain an intermediate 3 C template. To generate the 4 C template, DNA was redigested using 50 U DpnII (NEB #0543 S) as secondary antibody followed again by proximity ligation. Digestion and ligation efficiency were evaluated via agarose gel electrophoresis. Adaptor-containing reading and non-reading primers, specific to the viewpoints of interest, were designed to amplify all captured, interacting DNA fragments (*SOX11* viewpoint reading = CCACCAAAATTTTCATCATG and non-reading = TCTTCTATGCATCC GATTCT; without adapter sequence). 4C-sequencing fragments were amplified using the Expand Long Template PCR System (Roche, 11681834001) and using High Pure PCR Product Purification kit (Roche #11732676001) and QIAquick PCR Purification kit (Qiagen #28106).

Sequencing was performed on the Nextseq 500 platform (single-end, 75 nt, loading concentration 1.6 pM). Sequencing reads from the 4 C library were demultiplexed using the demultiplex.py script (https://gist.github.com/meren/7632184) and aligned using bowtie2

(v.2.3.1). For the cis chromosome, mapped read counts were summarized per DpnII restriction fragment. Finally, 4 C coverage profiles were obtained by normalizing the per fragment coverage to reads per million (RPM) on the cis chromosome and smoothing the normalized coverage using the rollmean function from the R 'zoo' package with a window size of 21 fragments. For visualization purposes, the viewpoint was removed (chr2:58210000-4835000) and the plot was generated using R package Sushi (v1.32.0) with normalized bedgraph files. Interaction peaks were called on the raw fragment count data using the peakC R package[34]. Settings and parameters were: "single analysis", window = 2e6, vp.pos = 5834393 (SOX11), minDist = 15e3, wSize = 21, qWd = 2, qWr = 2.

## Transfection and nucleofection of cell cultures

Cells were seeded in 6-well tissue culture plates 24 h prior to transfection. 100 nM of siRNA non-targeting control (siRNA NTC, D-001810-10-05) or siRNA *SOX11* (L-017377-01-0005, Dharmacon) were transiently transfected using DharmaFect 2 (Thermo Fisher Scientific) according to the manufacturer's guidelines. For nucleofection, cells were nucleofected with 100 nM of the above described siRNA NTC and siRNA SOX11 using the Neon Transfection System (Thermo Fisher Scientific) and subsequently seeded in 6-well or T25 tissue culture plates.

## Generation of stable cell lines

Four different mission shRNAs from the TRC1 library (Sigma-Aldrich, TRCN0000019174, TRCN0000019176, TRCN0000019177, TRCN00 00019178 referred in the manuscript as sh1, sh2, sh3, sh4 respectively) targeting *SOX11* and one non-targeting shRNA control (SHC002, NTC) were used to generate neuroblastoma cell lines with *SOX11* knockdown.

Virus was produced by seeding $3 × 10^6$ HEK-293TN cells in a 15 cm² dish 24 h prior to transfection. Transfection of the cells was done with trans-lentiviral packaging mix and lentiviral transfection vector DNA according to the Thermo Scientific Trans-Lentiviral Packaging Kit (TLP5913) using $CaCl_2$ and 2x HBSS. 16 h after transfection, cells were refreshed with reduced serum medium and lentivirus-containing medium was harvested 48 h later. Virus was concentrated by adding 2500 μl ice-cold PEG-IT (System Biosciences) to 10 ml harvested supernatants and incubating overnight at 4 °C, after which complete medium was added to the remaining pellet upon centrifugation. NGP, CLB-GA and IMR-32 cells were transduced by adding 250 μl concentrated virus to 1750 μl complete medium. 24 h after transduction cells were refreshed with medium and 48 h after transduction, cells were selected using 1 μg/ml puromycin.

For *SOX11* inducible overexpression, the OriGene vector SC303275 containing the cDNA of *SOX11* was amplified by PCR and the obtained fragment was gel purified and ligated into the opened NdeI site of response vector pLVX-TRE2G-Zsgreen1 (Takara, cat#631353) producing pLVX-TRE3G-Zsgreen1-IRES-hSOX11. The constructed plasmid was verified by restriction digest and sequenced by Sanger DNA sequencing (GATC). Lenti-X 293 T Cells (Takara, cat#632180) were transfected with the regulator vector pLVX-pEF1a-Tet3G (cat#631353) and Lenti-X Packaging Single Shots (VSV-G) (cat#631275) according to the manufacturer's instructions. The supernatant containing the lentivirus was collected, filtered through a 0.45 μm filter and concentrated using PEG-IT. SH-EP cells were infected with the concentrated virus and 48 h of incubation thereafter, the transduced cells were selected using 500 μg/ml G418. Three individual clones were obtained by limiting dilution. After clonal expansion, the TET protein expression in each clone was checked by immunoblotting using TetR monoclonal antibody (Clone 9G9) (Clontech, cat#631131). In addition, induction of each expressing clone was tested after transduction with the pLVX-TRE3G-Luc control vector. Selected clones were transduced with lentivirus produced as described above from vector pLVX-TRE3G-Zsgreen1-IRES-hSOX11 and subsequently selected with

1 μg/ml puromycin. The SH-EP *SOX11* clones were grown in completed medium supplemented with 10% tetracyclin-free FBS to avoid leakage.

## Phenotypic assessment of cells

For the colony formation assay, 2000 viable NGP, CLB-GA and SK-N-AS cells with or without *SOX11* knockdown were seeded in a 6 cm dish in a total volume of 5 ml complete medium and were then left unaffected for 10–14 days at 37 °C. After an initial evaluation under the microscope, the colonies were stained with 0.005% crystal violet and digitally counted using ImageJ (v1.53). The IncuCyte® Live Cell imaging system (Essen BioScience) was used for assessment of proliferation after *SOX11* knockdown or overexpression. Briefly, $15 \times 10^3$ viable NGP or $10 \times 10^3$ viable SH-EP cells, with or without *SOX11* knockdown or overexpression, were seeded in 5 replicates in a 96-well plate (Corning costar 3596) containing complete medium. Cell viability was measured in real-time using the IncuCyte by taking photos every 3 h of the whole well (4x). Masking was done using the IncuCyte® ZOOM Software (v2016B).

For the scratch wound migration assay, cells were seeded as described above in an Imagelock 96-well plate (4379, Essen Bioscience) and a scratch wound was made in a confluent cell monolayer using a 96-well Incucyte® Wound Maker. Phase contrast imaging took place every 3 h. Images were analysed using IncuCyte® S3TM 2018B-2019A software. For statistical testing, a Levine's test was performed in SPSS (v27) at the 1% significance level upon which an independent paired *t*-test was performed at the 5% significance level.

For cell cycle analysis, $7 \times 10^5$ cells were seeded in a T25 in complete medium and nucleofected with *SOX11* siRNA or transduced with *SOX11* shRNAs and respectively controls and selected with puromycin, as described above. Cells were trypsinized and washed with PBS. The cells were resuspended in 300 μl cold PBS and while vortexing, 700 μl of 70% ice-cold ethanol was added dropwise to fix the cells. Following incubation of the sample for minimum 1 h at −20 °C, cells were washed in PBS and resuspended in 500 μl PBS with RNase A to a final concentration of 0.25 mg/ml. Upon 1 h incubation at 37 °C, 20 μl Propidium Iodide solution was added to a final concentration of 40 μg/ml. Samples were loaded on a BioRad S3™ Cell sorter and analysed with the Dean-Jett-Fox algorithm for cell-cycle analysis using the ModFit LT™ software package (v6.0).

## Culture and RNA-sequencing of hPSC differentiation track

Utilizing a modified dual-SMAD inhibition differentiation protocol developed by the Studer laboratory at the Memorial Sloan Kettering Cancer Center, we performed in vitro differentiations of hPSCs into SAPs. Over the course of a 40-day differentiation, cells were cultured and sorted on day 16 for the CD49d maker (*SOX10* positive cells), when cells are committed to trunc neural crest cells. Cells were harvested at the neural crest and hSAP stages. RNA was isolated from the collected cell pellets by lysing the cells in TRIzol Reagent (ThermoFisher catalog #15596018) and inducing phase separation with chloroform. Subsequently, RNA was precipitated with isopropanol and linear acrylamide and washed with 75% ethanol. The samples were resuspended in RNase-free water. After RiboGreen quantification and quality control by Agilent BioAnalyzer, 534–850 ng of total RNA with DV200% varying from 38–74% was used for ribosomal depletion and library preparation using the TruSeq Stranded Total RNA LT Kit (Illumina catalog #RS-122-1202) according to manufacturer's instructions with 8 cycles of PCR. Samples were barcoded and run on a HiSeq 4000 in a 50 bp/50 bp paired end run, using the HiSeq 3000 / 4000 SBS Kit (Illumina). On average, 48 million paired reads were generated per sample and 35% of the data mapped to the transcriptome.

## RNA-sequencing of perturbated NB cells

For IMR-32 and CLB-GA after *SOX11* knockdown for 48 h and SH-EP after *SOX11* overexpression for 48 h poly-adenylated stranded mRNA

sequencing was performed using the TruSeq Stranded mRNA Sample Prep Kit from Illumina according to the manufacturer's instructions (#20020594). For NGP after *SOX11* knockdown for 48 h and SH-EP after *SOX11* overexpression for 9 h, QuantSeq 3'UTR mRNA sequencing was performed according to the manufacturer's instructions (Lexogen, 015.24). In brief, the samples were respectively prepared using the TruSeq Stranded mRNA Sample Prep Kit from Illumina or QuantSeq 3' mRNA-Seq Library Prep Kit from Illumina and subsequently sequenced on the Nextseq 500 platform. Sample and read quality were checked with FastQC (v0.11.3). The QuantSeq generated reads were trimmed using cutadapt version 1.11 to remove the "QuantSEQ FWD" adaptor sequence. Reads were aligned to the human genome GRCh38 with the STAR aligner (v2.5.3a) and gene count values were obtained with RSEM (v1.2.31). Genes were only retained if they were expressed at counts per million (cpm) above 1 in at least four samples. Counts were normalized with the TMM method (R-package edgeR, v3.36.0), followed by voom transformation and differential expression analysis using limma (R-package limma, v3.503.3). A general linear model was built with the treatment groups (knockdown or overexpression) and the replicates as a batch effect. Statistical testing was done using the empirical Bayes quasi-likelihood *F*-test. Gene Set Enrichment Analysis was performed on the genes ordered according to differential expression statistic value (*t*). Signature scores were conducted using a rank-scoring algorithm[35]. A custom-made ReplotGSEA function was used to generate gene set enrichment plots (https://github.com/PeeperLab/Rtoolbox/blob/master/R/ReplotGSEA.R). For the data generated on the foetal adrenal glands and differentiation along the sympatho-adrenal lineage, normalisation was done using DESeq2 and rlog transformation, which is more robust in the case when the size factors vary widely.

## Western blot analysis and antibodies

Proteins were isolated using a RIPA lysis buffer (5 mg/ml sodium deoxycholate, 150 mM NaCl, 50 mM Tris-HCl pH 7.5, 0,01% SDS solution, 0,1% NP-40) supplemented with protease inhibitors. In total, 40 μg of protein lysate was loaded onto an SDS-PAGE gel (10% Pre-cast, Bio-Rad), run for 1 h at 150 V and subsequently blotted onto a nitrocellulose membrane. Antibodies were selected based on validation described on the manufacturer's website and existing citations. Next, antibodies were validated in the lab by western blot with total protein lysates collected from different neuroblastoma cell lines. The membranes were probed with the following primary antibodies: anti-SOX11 antibody (SOX11-PAb, 1:1000 dilution), anti-c-MYB antibody (12319 S, Cell Signaling, 1:1000 dilution), anti-MYCN antibody (SC-53993, Santa Cruz 1:1000 dilution), anti-SMARCC1 antibody (11956 S, Cell Signaling 1:1000), and anti-SMARCA4 antibody (3508 S, Cell Signaling, 1:500). As secondary antibody, we used HRP-labeled anti-rabbit (7074 S, Cell Signalling, 1:10,000 dilution) and anti-mouse (7076P2, Cell Signalling, 1:10,000 dilution) antibodies. Antibodies against Vinculin (V9131; Sigma-Aldrich, 1:10,000 dilution), alpha-Tubulin (T5168, Sigma-Aldrich, 1:10,000 dilution) or beta-actin (A2228, Sigma-Aldrich, 1:10,000 dilution) were used as loading control. The rabbit polyclonal antibody, SOX11-PAb, was custom made (Absea biotechnology, China) against the immunogenic peptide p-SOX11C-term DDDDDDDDDELQLQIKQEPDEEDEEPPHQQLLQPPGQQ PSQLLRRYNVAKVPASPTLSSSAESPEGASLYDEVRAGATSGAGGGSRLY YSFKNITKQHPPPLAQPALSPASSRSVSTSSS and used for western blot and chromatin immunoprecipitation for SOX11. All antibodies were diluted in milk/TBST (5% non-fat dry milk in TBS with 0.1% Tween-20). Binding of the antibodies with the membrane was evaluated using the SuperSignal West Dura or Femto Extended Duration Substrate (Thermo Scientific #34075 and #34096). Pictures were taken with the ChemiDoc-It Imaging System (UVP) using the VisionWorks analysis software (UVP, v2.0.0), quantification of the blots were performed using ImageJ (v1.53). Uncropped scans of the blots used can be found in the Source Data files.

## Chromatin immunoprecipitation (ChIP) assay and ATAC

Chromatin immunoprecipitation (ChIP) and Assay for Transposase-Accessible Chromatin (ATAC) sequencing was performed as previously described[36]. For ChIP-seq a total of $10 \times 10^7$ cells were cross-linked with 1% formaldehyde while shaking for 7 min at room temperature, quenched with 125 mM glycine, lysed and sonicated with the S2 Covaris for 30 min to obtain 200–300 bp long fragments. Chromatin fragments were immunoprecipitated overnight using 1 μg antibody of SOX11-PAb antibody (custom made by Absea biotechnology, China) and 20 μl Protein A UltraLink® Resin (Thermo Scientific, #53139) beads per $10 \times 10^7$ cells. Reverse crosslinking was done at 65 °C for 15 h and chromatin was resuspended in TE-buffer, incubated for 2 h at 37 °C with 0.2 mg/ml RNase and followed by an incubation of 2 h at 55 °C with 0.2 mg/ml proteinase K. DNA was isolated using 400 μl phenol:chloroform:isoamylalcohol (P:C:IA) in phase lock gel tubes (5Prime). Upon centrifugation, the aqueous layer was transferred to a new tube with 200 mM NaCl, 30 μg glycogen and 800 μl 100% ethanol, and incubated for 30 min at −20 °C. Upon centrifugation, the pellet was washed with 80% Ethanol and resuspended in RNase/DNase free water. DNA concentration was measured using the Qubit® dsDNA HS Assay Kit. Library prep was done using the NEBNExt Ultra DNA library Prep Kit for Illumina (E7370S) with 500 ng starting material and using 8 PCR cycles according to the manufacturer's instructions. For ATAC-seq, 50,000 cells were lysed and fragmented using digitonin and Tn5 transposase. The transposed DNA fragments were amplified and purified using Agencourt AMPure XP beads (Beckman Coulter). ChIP and ATAC library concentrations were measured with the Illumina Kapa Library quantification kit (Roche #07960140001) and libraries were sequenced on the NextSeq 500 (Illumina) using the Nextseq 500 High Output kit V2 75 cycles single-end (Illumina).

## CUT&RUN assay

CUT&RUN coupled with high-throughput DNA sequencing was performed using Cutana pA/G-MNase (Epicypher, 15-1016) according to the manufacturer's protocol. Briefly, cells (0.5 M cells/sample) were washed and incubated with activated Concanavalin A beads for 10 min at room temperature. Cells were then resuspended in antibody buffer containing 0.01% digitonin, 1:100 dilution of each antibody (anti-SOX11, HPA000536; IgG goat, sc-2028; anti-CTC, Merck 07-729) was added to individual cell aliquots and tubes were rotated at 4 °C overnight. The following day, targeted chromatin digestion and release was performed with 2.5 mL Cutana pA/G-MNase and 100 mM CaCl2. Retrieved genomic DNA was purified with the MinElute PCR purification kit and eluted in 10 mL of buffer EB. Sequencing libraries were prepared with the automated Swift 2 S system, followed by 100 bp-PE sequencing with Novaseq 6000.

## CUT&RUN, ChIP-seq and ATAC-seq data-processing and analysis

Prior to mapping to the human reference genome (GRCh37/hg19) with bowtie2 (v.2.3.1), quality of the raw sequencing data of CUT&RUN, ChIP-seq and ATAC-seq was evaluated using FastQC and adapter trimming was done using TrimGalore (v0.6.5). Quality of aligned reads were filtered using min MAPQ 30 and reads with known low sequencing confidence were removed using Encode Blacklist regions. For CUT&RUN, because of oversequencing, reads were subsampled, and mapping was done with 10 M reads (recommended read depth), for ChIP-seq and ATAC-seq all sequenced reads were mapped. Peak calling was performed using MACS2 (v2.1.0) taking a q value of 0.05 as threshold and peaks were filtered for chr2p amplified regions in the case of IMR-32 cells. Homer[37] (v4.10.3) was used to perform motif enrichment analysis, with 200 bp around the peak summit as input. Overlap of peaks, annotation, heatmaps and pathway enrichment was analysed using DeepTools (v3.5.1), the R package ChIPpeakAnno (v3.28.1), and the web tool enrichR. Sushi (v1.32.0) was used for visualization of the

data upon RPKM normalization or log likelihood ratio calculation with MACS2 (v2.1.0).

## Reporting summary

Further information on research design is available in the Nature Portfolio Reporting Summary linked to this article.

## Data availability

The raw RNA-sequencing, CUT&RUN, ChIP-sequencing and ATAC-sequencing datasets generated during this study were deposited in the ArrayExpress database at EMBL-EBI (www.ebi.ac.uk/arrayexpress) and in Gene Expression Omnibus (GEO) with the following accession numbers: E-MTAB-9340 (RNA-seq IMR-32 *siSOX11*), E-MTAB-11883 (RNA-seq CLB-GA and NGP *siSOX11*), E-MTAB-11892 (RNA-seq SHEP 9 h *SOX11* OE), E-MTAB-9338 (RNA-seq SHEP 48 h *SOX11* OE), E-MTAB-9464 (ChIP-seq SOX11 IMR-32), E-MTAB-11905 (CUT&RUN SOX11 IMR-32, CLB-GA, NGP and SH-EP 48 h *SOX11* OE), GSE224245 (CUT&RUN CTCF CLB-GA), GSE224245 (4C-Sequencing SOX11 viewpoint CLB-GA, KELLY, SK-N-AS and SH-EP) and GSE224245 (ATAC-sequencing IMR-32, CLB-GA and SH-EP). The publicly available neuroblastoma tumor data that supports the findings of this study are available from the Neuroblastoma Research Consortium (NRC, GSE85047)[17], Su et al. (Kocak, GSE45547 and GSE62564)[18,30], Depuydt et al. (GSE103123)[10], Versteeg et al. (GSE16476)[38] and Janoueix-Lerosey et al. (GSE12460)[39]. Additional tumor data sets were used for glioma (GSE4290)[40], breast cancer (GSE12276)[41], thyroid cancer, lung cancer, prostate cancer (GSE2109), ALL (GSE10609)[42], colon cancer (GSE14333)[43] and ovarian cancer (GSE12172)[44]. Cancer cell line information (expression, copy number, methylation and dependency data) is available via the CCLE database (https://depmap.org/portal/). The publicly available RNA expression data for isogenic NB cell lines used in Supplementary Figure 1L is provided by van Groningen et al. (GSE90803)[45]. ChIP-seq data are available from: MYCN, TWIST1 (GSE94822)[46], ASCL1 (GSE159613)[47], HAND2, GATA3 and PHOX2B (GSE90683)[4], SMARCA4 (GSE134626)[28]. The public H3K27ac ChIP-seq and RNA-seq expression data from NB cell lines and tumor used in Fig. 2 and Supplementary Figure 2 is provided by Gartlgruber et al. (GSE136209)[14]. The publicly available expression data in the *TH-MYCN*+/+ NB tumor mice model is provided by De Preter et al. (E-MTAB-3247)[48]. Additional publicly available data used are H3K27ac (GSE136209)[14], H3K4me1 (E-MTAB-6570)[36] in CLB-GA, ATAC, H3K4me3 and H3K27ac (E-MTAB-6570)[36] in IMR-32, ATAC (GSE80151)[49], H3K4me3 and H3K27ac (GSE138314)[50] in NGP, ATAC (GSE138293)[50] and H3K27ac (GSE136209)[14] in KELLY, H3K27ac (GSE189174)[51] in SH-EP and ATAC (GSE138293)[50] and H3K27ac (GSE136209)[14] in SK-N-AS. The remaining data are available within the Article, Supplementary Information or Source Data file. Source data are provided with this paper.

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

## Acknowledgements

The authors would like to thank C. Nunes, L. Mus, K. Verboom, S. Claeys, J. Van Laere and E. De Smet for technical assistance, P. Reynolds and M. Hogarty for providing the COG-N-373 cell line. We acknowledge the use of the Integrated Genomics Operation Core, funded by the NCI Cancer Center Support Grant (CCSG, P30 CA08748), Cycle for Survival, and the Marie-Josée and Henry R. Kravis Center for Molecular Oncology. This research was supported by the following funding agencies: the Belgian Foundation against Cancer (project 2015-146 and F/2018/1246) to F.S., Ghent University (BOF10/GOA/019, BOF16/GOA/23 and BOF/24 J/2021/150) to F.S., the Belgian Program of Interuniversity Poles of Attraction (IUAP Phase VII - P7/03) to F.S., the Fund for Scientific Research Flanders (Research projects G053012N, G050712N and G051516N to F.S., G021415N to K.D and F.S.), 'Kom op tegen Kanker' (Stand up to Cancer) the Flemish cancer society (Research grant to F.S.), the European Union H2020 (OPTIMIZE-NB GOD9415N and TRANSCAN-ON THE TRAC GOD8815N to F.S.) and FP7 (ENCCA 261474 and ASSET 259348 to F.S.), 'Kinderkankerfonds' (Research grant to F.S.), Olivia Fund to F.S. and Villa Joep to F.S. The following authors B.D., A.L., and S.V. were supported by an FWO grant (FWO3F02013001701/02, FWO.3F0.2019.0033.01, and FWO.3F0.2019.0041.01 respectively). B.D. was supported by CRIG (E/02578/01). A.L. was supported by Kom op tegen Kanker (STI.VLK.2018.0016.01). M.W.Z. was supported by grants from the Alex's Lemonade Stand Foundation, Charles A. King Trust, and Claudia Adams Barr Foundation.

## Author contributions

B.D., A.L., S.L., S.VHau., S.D., W.V., G.D., C.V., E.S., and N.R. contributed to the development and design of methodology; B.D., A.L., S.B., C.V., E.D., S.V., J.R., J.V., D.C. and J.K. performed computational and statistical analysis; B.D., A.L., S.L., S.D., F.D., S.VHav., E.S., A.B. and M.W.Z. performed experiments; R.V., J.N., J.K., N.R., M.F., J.S. and S.E. provided material, data and analysis tools, B.D. managed the maintenance of data, B.D., A.L., F.S. and K.D. wrote the original draft, S.L., S.VHau., C.V., J.N., J.K., W.V., S.S.R., T.P. and P.V. contributed to manuscript review and editing, B.D., A.L., S.B and G.D. contributed to data representation and visualization; F.S. and K.D. directed the project and were responsible for funding.

## Competing interests

The authors declare no competing interests.

## Ethics

Patient sample collection of NB tumor used for tissue micro-array (TMA)[52] was approved by the ethical committee (Ethical Committee Ghent University Hospital, EC/208-2006/Svdm) and written informed consent was obtained from the patients and/or their parents. All other datasets involving patient data are publicly available and information regarding ethical consent can be found in the linked accession codes and publications.

## Additional information

[1]Department of Biomolecular medicine, Ghent University, Ghent 9000, Belgium. [2]Cancer Research Institute Ghent (CRIG), Ghent 9000, Belgium. [3]Department of Pediatric Oncology, Dana-Farber Cancer Institute, Boston, MA 02215, USA. [4]Department of Pediatrics, Memorial Sloan Kettering Cancer Center, New York, NY 10065, USA. [5]Department of Pathology, Ghent University Hospital, Ghent, Belgium. [6]Department for Experimental Pediatric Oncology, and Center for Molecular Medicine Cologne (CMMC), Medical Faculty, University of Cologne, Cologne, Germany. [7]Department of Pediatric Oncology and Hematology, Charité-Universitätsmedizin Berlin, Berlin 13353, Germany. [8]Department of Immunotechnology, Lund University, Lund, Sweden. [9]Department of Oncogenomics, Academic Medical Center, Amsterdam 1105 AZ, The Netherlands. [10]These authors contributed equally: Bieke Decaesteker, Amber Louwagie. [11]These authors jointly supervised this work: Katleen De Preter, Frank Speleman. ✉e-mail: Bieke.Decaesteker@ugent.be; Franki.Speleman@ugent.be

