## [Peer Review File · Nature Communications]

SOX11 regulates SWI/SNF complex components as member of the adrenergic neuroblastoma core regulatory circuitryReviewers' Comments:

Reviewer #1:

Remarks to the Author:

Decaesteker and Louwagie et al. provide an incisive and intriguing set of observations to suggest that SOX11 is a lineage-specific factor in high risk neuroblastoma. The dominant findings of the paper are that SOX11 was identified as being important in neuroblastoma through the observation of rare copy number amplifications in neuroblastoma tumors. Further, in tumors without copy number gain, it appears that SOX11 is increased in expression through local enhancers that are bound by members of the core-regulatory circuitry of adrenergic-subtype neuroblastoma. They go on to demonstrate that SOX11 is involved in control of neuronal cell lineage, perhaps by epigenetic regulation of transcription. By a combination of over expression and knockdown methods, they demonstrate that SOX11 co-binds with MYCN, enriched at promoter elements, and that this activity is associated with up regulation of nucleosome remodeling complex members. Finally, in an intriguing set of experiments, they demonstrate that SOX11 regulates chromatin accessibility, implicating it as a potential pioneer factor in neuroblastoma. Their findings are intriguing and original, and will align well with unpublished work from other groups describing SOX11 as a lineage-specific regulator. Their observations are unique, and therefore of high relevance to the community. I have two major concerns with this manuscript: 1) The assertion of SOX11 binding is limited to ChIP studies in one cell line and 2) the authors demonstrate transcriptional changes without identifying whether these have functional outcomes. These should be addressed by the authors. I have otherwise structured my comments in terms of both major and minor suggestions, as I believe they are intended to strengthen this manuscript, and they are as follows:

Major Suggestions:

1. It is not clear whether the TMA staining was done on exclusively MYCN-amplified cases. This should be specifically stated in the text, given the association between SOX11 and MYCN expression. If cases are not MYCN-amplified, but are MYC enhancer hijacked, or high MYCN expressors, then they should be stated as such. This is important to determine if high SOX11 expression is at all related to MYCN, or, as is likely to be the case, is found associated with any neuroblastoma tumor with high level MYC family protein expression.
2. In figure 2, the authors present multiple enhancers at the SOX11 locus. Are these truly individual enhancers, or should they be bioinformatically "stitched" to represent one common enhancer? It is confusing to refer to multiple small enhancers as multiple small "super-enhancers," when it is likely that these represent a single super-enhancer. This will likely change their observation that a small number of adrenergic neuroblastoma cell lines are characterized by super-enhancers at the SOX11 locus. The authors should bolster their data with publicly available H3K27-acetyl data from John Maris/Jo-Lynne Rokita's group, in addition to available data from Boeva et al., van Groningen et al. and Durbin et al. to analyze a larger set of cell lines, both mesenchymal and adrenergic. If it is true that a super-enhancer is only found in a minority of noradrenergic cell lines, then the authors should attempt to identify the mechanism by which SOX11 is highly expressed, without an epigenetic basis, in adrenergic neuroblastoma, or at least comment on this.
3. The indication to discuss the SILC1 ncRNA in figure 2 is unclear – the authors demonstrate that it contains similar expression as SOX11, which would be expected if it was co-regulated by similar enhancers. Is there functional significance to SILC1 expression? Do the authors contend that, as in development, SILC1 expression is required for SOX11 expression? If so, then the authors should perform genetic disruption of SILC1 and examine SOX11 expression levels, in addition to expression of other CRC members – If disrupted, then SILC1 could be functioning as a ncRNA CRC regulator, as previously demonstrated for other ncRNA species by Suzuki, Sharp and Young (Cell 2017). In the absence of mechanistic data such as this to implicate SILC1 in the regulation of SOX11 expression, this observation is somewhat confusing, and I would suggest the authors remove this distracting information.
4. The authors should present the Depmap data they refer, at least in a supplementary figure.
5. Major concern: In Figure 3, the authors show that SOX11 knockdown by either shRNA or siRNA

results in a growth phenotype, with an induction of a G1 cell cycle arrest in Figure 3. In Figure 4, however, they demonstrate effects on RNA expression associated with changes in cell adhesion, migration and motility. Are these transcriptional changes related? The authors should examine whether SOX11 loss causes changes in migratory or metastatic behaviors of their cell lines, as they undergo change in fate. Similarly, are the effects on mRNA processing and translation functional? While loss of a lineage-specifying transcription factor is likely to result in a variety of changes, it should be made clear which changes have functional outcomes, and which are “transcriptional bystanders” and do not reflect the altered biology of the cell.

6. Major concern: It is important that the authors provide more evidence for SOX11 binding. They have conducted ChIP-seq to SOX11 in one cell line (IMR-32), and in order to make formal claims about locations of SOX11 binding, it is important to see this data in more cell lines. To this end, an analysis of where SOX11 binds, distinguishing promoters, typical enhancers, super-enhancers and gene bodies would be helpful. The overlap between H3K4me3 and SOX11 binding would indicate that SOX11 is primarily a promoter binding factor – is this true? The authors also speculate that SOX11 may co-bind with other master TFs in neuroblastoma, such as HAND2, TWIST1, TCF3, ASCL1. Much of this data is publicly available, and the authors should perform further ChIP studies in cell lines where publicly available data exists, in order to correlate.

7. The authors identify that SOX11 is required for opening of sites associated with adrenergic neuroblastoma in figure 6. This data is striking and well performed. Do the authors content that this effect is direct (i.e. SOX11 interacts physically with nucleosome remodeling complex members, to recruit them to maintain these sites?). If so, then the authors should provide evidence that SOX11 physically interacts with members of the nucleosome remodeling complex. If not, then how do the authors suggest that this occurs? If SOX11 itself has nucleosome remodeling activity in neuroblastoma, this should be identified by nucleosome assays.

8. Two other recent papers on BioRxiv have suggested that loss of SOX11 is associated with increased SOX4 expression - if SOX11 is required to maintain open chromatin sites, then is SOX4 or some other SOX member upregulated to permit accessibility in mesenchymal subtype neuroblastoma? This comparison may be easy to make, by simply comparing RNAseq expression of SOX family members in different cell lines (adrenergic vs. mesenchymal) and then relating this to effects with SOX11 knockdown and/or overexpression. One might expect that since SOX11 knockdown cells do not become “fully mesenchymal” that a SOX transcription factor required to open chromatin in mesenchymal neuroblastoma is not induced.

Minor Concerns:

1. The authors demonstrate that SOX11 is coamplified with MYCN, and that higher expression goes with poorer survival. It would be helpful to perform similar correlational analyses with age and stage.

2. The statement: Higher expression levels of SOX11 both at mRNA and protein level were observed in adrenergic NB cell lines compared to mesenchymal NB cell lines and tumors (Fig. 1g, Fig. S1j-k) – should be softened – in Fig 1g, the expression of SOX11 is high regardless of MES or ADRN status (10-14 log2)

3. The title of figure 3 should be revised, the data does not reflect that SOX11 is a lineage factor.

4. The extensive heatmap analysis in Figure 4a/b is of limited utility and should be supplemented. In contrast, Supplemental figure 4a and b, d and f are extremely helpful to understanding the primary goal of the figure, and should be brought to the main figure.

5. The authors provide evidence for broad dysregulation of BAF complex transcription but do not describe it in the text. They should postulate on how SOX11 regulates BAF complexes, especially in the context of the recent evidence for BAF complex members in the regulation of an adrenergic-mesenchymal switch (Shi H et al Science Advances 2020).

6. There appears to be no supplementary figure 3.

7. The authors comment that SOX11 is not a member of the extended regulatory network (ERN), however, do not test this. An expressed transcription factor, whether it is super-enhancer regulated or not, is, by definition, a member of the ERN. It is without doubt that the CRC factors bind to the SOX11 promoter and enhancer, resulting in regulation of it – this makes SOX11 at least a member of the ERN.

8. Was the RNAseq analysis performed using external RNA controls? Given the likely interaction with

MYCN, and the effect of MYCN on transcriptional amplification (Nie et al. Cell 2012; Lin et al, Cell 2012), this is necessary for interpretation of the results. The authors should comment on this.

Reviewer #3:

Remarks to the Author:

The authors present an intriguing study on the potential role of the SOX11 transcription factor in neuroblastoma cells. Unfortunately, reaching sound conclusions on the influence of SOX11 isn't possible since much of the data is preliminary and the authors do not present a thorough comparison of the impact of SOX11 in any single cell background. Rather, they use different NB cells lines throughout the course of the manuscript, which prevents a rigorous comparison of the various datasets. In considering this work for publication the following points should be weighed.

Which one is the pioneering factor SOX11 or MYCN or is the combination required? Given the correlation between SOX11 gain and MYCN amplification (Fig. S1a), it would seem the later but that should be validated.

As potential super-enhancers were only detected in the minority of NB cell lines (Fig. 2 and S2a), what is primary phenotype of the majority?

Statistical analysis is required for the data in Fig. 2d, otherwise it is not possible to distinguish the relative differences between the CLB-GA and SH-EP cells. If there are statistically notable differences, is it only with CLB-GA or it common to other NB cell lines (aka, is CLB-GA an anomaly)? Much of the presented functional work is accomplished in IMR-32 cells, is this why? Oddly, they switch back to CLB-GA cells for the chromatin accessibility test (Fig. 6) without justification. Are the CLB-GA and IMR-32 cells reacting differently to changes in SOX11? The lack of chromatin accessibility data for IMR-32 is particularly disappointing since inclusion would have allowed a very useful comparison of the ChIP determined SOX11 localization and SOX11-dependent changes in gene expression.

The high variability in knocking down SOX11 yet comparable functional impact on the cells suggests there is either an indirect effect that is driving the differences or the slight reduction in SOX11 observed in CLB-GA cells impacts the key activity (Fig. 3). The authors should identify and report what that key function is. In addition, the colony formation data in Fig. 3c needs quantification otherwise it is not possible to reach a conclusion.

The authors wait 48 h after overexpressing SOX11 (Fig. 4b). As this provide ample time for downstream players to impact the system, do the authors observe comparable gene expression changes after 2-4 hours? Does SOX11 occupy the regulatory elements controlling all the various gene loci as detected by ChIP? Relying on "predicted SOX11 binding sites" (Fig. S4b) is not sufficient. The reported low (12%) use of predicted SOX11 binding sites (Fig. 5) substantiate this concern. Importantly, any ChIP results should be validated by checking that the signal is lost/reduced when SOX11 is knocked down including for the results presented in Fig. 5.

The authors make an intriguing suggestion that the potential SOX11-dependent increased expression of CDKN1a (Fig. S4c) is the causative factor for the cell cycle arrest. Does knocking down CDKN1a in these cells restore the cell cycle?

The authors point out that SOX11 repressed target genes fall within the same gene categories as those previously reported to be controlled by MYCN, but are these the same genes as would be expected if SOX11 and MYCN are working together, or do the loci just happen to fall with common gene categories?

The authors state that SOX11 DNA motifs correlate with MYCN and MAX sites but do not describe whether the correlation varies with impact on gene expression—is there a correlation of MYCN or MAX motifs with SOX11 repressed or activated genes?

Below, we have answered each questions of the reviewers, adding relevant sections of the manuscript in blue with new added text underlined and removed text marked by 'strike through'. Taken together all the suggestions of both reviewers we have furthermore adapted our abstract, introduction and discussion for which the changes can be found in the attached updated manuscript file. In the attached updated manuscript file, new text is indicated in blue and removed text is also marked by 'strike through'. References for the rebuttal are listed at the end and if appropriate also included in the adapted manuscript.

Array Express access codes (<https://www.ebi.ac.uk/arrayexpress/>) for all included omics datasets are given below. Reviewer access codes will be send when data is accepted on Array Express:

Technique	Cell line	Conditions	Array Express accession code
RNA-seq	IMR-32	siSOX11 48h	E-MTAB-9340
	CLB-GA	siSOX11 48h	E-MTAB-11883
	NGP		
	SH-EP	SOX11 OE 9h	E-MTAB-11892
	SH-EP	SOX11 OE 48h	E-MTAB-9338
ChIP-seq	IMR-32	SOX11 IP	E-MTAB-9464
CUT&RUN	IMR-32	SOX11 IP	E-MTAB-11905
	CLB-GA		
	NGP		
	SH-EP	SOX11 OE 48h, SOX11 IP	
ATAC-seq	SH-EP	SOX11 OE 48h	E-MTAB-11898

Reviewer #1 (Remarks to the Author):

Author's response: We would like to thank Reviewer #1 for the helpful comments. We have addressed the two major proposed concerns as well as the other points raised by this reviewer. We have provided extra data or information/clarifications and changed the manuscript where appropriate. A detailed point-by-point response is also included below indicating which changes were made in the manuscript (highlighted in blue), as well as all necessary modifications to the figures or new figures.

Major concerns:

Reviewer 1, Major Question 1: *It is not clear whether the TMA staining was done on exclusively MYCN-amplified cases. This should be specifically stated in the text, given the association between SOX11 and MYCN expression. If cases are not MYCN-amplified, but are MYC enhancer hijacked, or high MYCN expressors, then they should be stated as such. This is important to determine if high SOX11 expression is at all related to MYCN, or, as is likely to be the case, is found associated with any neuroblastoma tumor with high level MYC family protein expression.*

1. Author's response: We thank the reviewer for this remark and agree this additional information can be useful to the reader.

1.1. To address this remark, we first added more information on the clinico-genetic parameters of the tumors tested for SOX11 protein expression levels in the tissue micro-array as indicated in the new Supplementary table 1. TMA staining was done on 83 primary NB tumor cases and samples were scored according to the histopathological classification. We provide age at diagnosis, overall survival status, days of follow up or disease, stage and MYCN status for the included tumors.

For four tumors SOX11 staining failed. In total, the analysis was performed on 69 tumors, of which 11 MYCN amplified and 58 MYCN non-amplified cases. For the Figure included in our original manuscript,

we have collected missing survival status for all 69 patients except one (for which this information could not be retrieved) and performed a re-analysis of our data. We updated the survival plot (see below) (n=68) and replaced this with the original plot in the manuscript (Fig. 1e).

Adjusted Figure 1e:

Adjusted legend Figure 1e: Immunohistochemical staining for SOX11 on a tissue micro-array (TMA) of 68 NB tumors and correlation of SOX11 protein levels (median cut-off of H-score 85, for details see Material and Methods) with overall survival (p=0.02). For each group, a representative immunohistochemical staining of one tumor is depicted on the right.

To inform more about the consistency of the TMA, we added following information in the text (underlined): In addition, SOX11 immunohistochemical analysis using two independent SOX11 antibodies (Fig.S1c) showed that high SOX11 protein expression levels were associated with worse overall survival in a cohort of 68 cases consisting of 11 MYCN-amplified (MNA) and 57 MYCN non-amplified (MNoA) cases (Fig. 1e, Supplementary Table 1).

1.2. Information on structural variants driving MYC expression is not available for the tumor panel. We therefore performed staining for MYC on the TMA, to determine the correlation of SOX11 protein levels with MYC protein levels. MYC protein expression is very low or absent in all tumors of our cohort for the TMA. Based on the H-score ((number of cells in percent) * (intensity score: 0,1,2,3)), the maximum score to be reached is 300. For all tumors on the TMA, we reach a maximum score of 37 with an average score of 3 for MYC expression levels. In comparison, for Burkitt-lymphoma the scores for MYC with this antibody are always higher than 240. The low MYC expression in our tumor panel indicates that SOX11 expression is associated with MYCN, independent of MYC. Similar to the protein expression levels of MYC in the TMA, we evaluated MYC mRNA expression in the NRC cohort used for survival analysis in Fig. 1d. While *SOX11* expression is significantly positively correlated with *MYCN* expression in this tumor dataset and significantly higher expressed in the MNA tumor group, there is no clear association observed between *MYC* and *SOX11* mRNA expression levels (Rebuttal figure 1). Of further interest, according to Zimmerman et al.¹, a structural variant causing enhanced MYC expression is a rare event (5% of the investigated tumors identified with a MYC enhancer focal amplification or enhancer hijacking by translocation into the MYC locus). Taken together, in view of our expanded staining data and in keeping with MYC activation being a rare event in NB, we trust our TMA data as presented are reliable.

Rebuttal Figure 1: A-B: *SOX11 and MYCN expression are significantly positively correlated, while SOX11 and MYC expression are significantly negatively correlated in the NRC cohort (n=283). C. SOX11 expression is significantly higher in tumors with MYCN amplification in the NRC cohort (n=277, 6 samples omitted because of unknown MYCN status).*

Reviewer 1, Major Question 2: *In Figure 2, the authors present multiple enhancers at the SOX11 locus. Are these truly individual enhancers, or should they be bioinformatically “stitched” to represent one common enhancer? It is confusing to refer to multiple small enhancers as multiple small “super-enhancers,” when it is likely that these represent a single super-enhancer. This will likely change their observation that a small number of adrenergic neuroblastoma cell lines are characterized by super-enhancers at the SOX11 locus. The authors should bolster their data with publicly available H3K27-acetyl data from John Maris/Jo-Lynne Rokita’s group, in addition to available data from Boeva et al., van Groningen et al. and Durbin et al. to analyze a larger set of cell lines, both mesenchymal and adrenergic. If it is true that a super-enhancer is only found in a minority of noradrenergic cell lines, then the authors should attempt to identify the mechanism by which SOX11 is highly expressed, without an epigenetic basis, in adrenergic neuroblastoma, or at least comment on this.*

2. Author’s response: We acknowledge the concern of this reviewer (and reviewer 2) in relation to the super-enhancer (SE) calling and data representation, with overall only a subset of adrenergic NB cell lines with a SE being called. To address this in more depth we looked into two important data sets which became available after submission of our manuscript and which, in our view, provides strong support for proposing *SOX11* as a super-enhancer marked core regulatory circuitry master transcription factor in adrenergic NB.

First, we took advantage of the recent paper by the Westermann team² who performed an extensive mapping of super-enhancers in 60 NBs (including 49 primary cases), 25 NB cell lines and two neural crest-derived cell lines thus currently representing the most comprehensive available data set. SE calling was performed using the ROSE algorithm, filtering out SE’s with H3K27ac peaks closer than 5 kb to a set of 40,512 consensus H3K4me3 peaks and SE’s present in less than 2 samples. Based on H3K27ac profiling and performed SE- calling, four major SE-driven epigenetic subtypes and their underlying master regulatory networks were described, namely *MYCN*-amplified, *MYCN* non-amplified high-risk, *MYCN* non-amplified low-risk and mesenchymal NBs. Using this approach, one consensus SE was annotated downstream of *SOX11*. This SE was called predominantly in the group of primary, adrenergic *MYCN* amplified tumors and *MYCN* non-amplified tumors and *SOX11* was, together with *MYCN*, *TWIST1* and *TBX2* considered as the four major constituents of a "highly subtype-specific CRC-TF module for *MYCN* amplified tumors". We noted that this *SOX11* SE was not called in mesenchymal NB cell lines as expected and called in four adrenergic NB cell lines. Furthermore, we tested the consensus SE signal data from the Gartlgruber et al. paper versus *SOX11*

expression levels and observed a strong correlation in both the primary tumor data set as well as cell lines, thus supporting a functional connection between super-enhancer activity of the *SOX11* consensus SE and the level of *SOX11* gene expression (Fig. 2a-d, Fig. S2a-c).

Second, additional functional support for a regulatory role of the called enhancers (and the SE in the subset of cell lines/tumors) is evident from the study of Banerjee et al., showing that silencing of this SE in KCNR neuroblastoma cells using dCas9-KRAB targeted guides resulted in a 40% decrease in *SOX11* mRNA levels and a reduced cell growth³.

Finally, depending on which algorithm (ROSE or LILLY), significantly less SE are called by ROSE which is most notable when comparing the called SE from Extended Data Fig. 6. from Gartlgruber et al.² versus Fig. 1.g from Boeva et al.⁴. Both algorithms apply given thresholds for H3K27ac in a particular genomic region and do not take into account functional information but lack support from functional data. Although this type of calling has been widely used, also in the context of identifying SE marked CRC TFs, this observation marks the limitation of this approach and may be improved in future algorithms when supplemented with functional enhancer activity data.

Of further note, it is clear that based on the H3K27ac profiles multiple common predicted enhancers are present within the large gene desert distal to the 3' coding end of *SOX11*, both in cell lines and tumors with or without a called SE in the Westerman data set. In our view, extended enhancer regions not called as super-enhancers but functionally active may act in control of highly expressed lineage-dependent TFs that are part of CRCs, although this will require further extensive investigations.

Taken together, both the novel SE mapping data on primary NBs and NB cell lines from the Westermann team and the functional analysis of the *SOX11* SE in KCNR cells strongly support our initial description of *SOX11* as SE-marked lineage dependency TF.

In line with the above, the manuscript has been adapted as follows:

Title: *SOX11* is flanked by multiple cis-interacting adrenergic specific super-enhancers

Master transcription factors implicated in defining cell lineage and identity are typically under the control of super-enhancers (SE). *SOX11* was previously identified as a super-enhancer-associated transcription factor in adrenergic NB cell lines. Gartlgruber et al. reported super-enhancers in comprehensive published dataset of 60 NB tumors and 25 cell lines and identified one consensus *SOX11* super-enhancer (present in at least 2 samples, not overlapping with H3K4me3 and 5kb away from a transcription start site), in the adrenergic NB subtype (*MYCN* amplified, high-risk *MYCN* non-amplified and low-risk *MYCN* non-amplified group), while absent or strongly attenuated in the mesenchymal super-enhancer defined group, both in cell lines (Fig. 2a and b, Fig. S2a-c) and tumors (Fig. 2a and c). Upon more detailed analysis, we observed a large (1.1 Mb) gene desert without protein coding genes marked by multiple H3K27ac peaks, distal to the 3' end of the *SOX11* locus, indicative of the presence of multiple active enhancers-(super-)enhancer activity (Fig. S2a-b). In keeping with the presumed gene regulatory activity of this super-enhancer region, the super-enhancer signal is correlated with *SOX11* expression both in NB cell lines ($r=0.774$, $p=8.39e-5$, Fig. 2d) and tumors ($r=0.778$, $p=1.25e-10$, Fig. 2d), supporting a functional interaction between this enhancer region and *SOX11* transcriptional regulation. In concordance with the absence of *SOX11* expression in the breast cancer cell line MCF-7 cell line, the non-malignant neural crest cell lines (P4 and P5) and the mesenchymal/neural crest like NB cell lines (SH-EP, HD-N-33 and SK-N-AS, GI-ME-N), H3K27ac enhancer peaks were absent in the gene desert distal to *SOX11* (Fig. S2c).

In support of our findings, interaction of the consensus super-enhancer with the *SOX11* promoter in KCNR NB cells was found by Banerjee et al. using HiC analyses. Moreover, targeting of this super-enhancer using CRISPR interference caused attenuated *SOX11* expression. In summary, multiple

adrenergic specific enhancers and a consensus SE are flanking the SOX11 locus with multiple independent data supporting its role in SOX11 regulation.

Adjusted Figure 2a-d:

Adjusted legend Figure 2: A. Super-enhancer calling (present in at least 2 samples and not overlapping with H3K4me3 5kb from the transcription start site) downstream of the SOX11 locus in NB tumors (MNA = adrenergic MYCN amplified, MES = mesenchymal, MNoA-LR = adrenergic MYCN non-amplified low-risk, MNoA-HR = adrenergic MYCN non-amplified high-risk) and NB cell lines (MNA = adrenergic MYCN amplified, MES = mesenchymal, MNoA = adrenergic MYCN non-amplified). B. Violin plot showing super-enhancer signal for each individual NB cell line colored by their ChIP-seq signature defined subgroup. C. Violin plot showing super-enhancer signal for each individual NB tumors colored by their ChIP-seq signature defined subgroup. D. (Left) Correlation of SOX11 expression with super-enhancer signal of the consensus super-enhancer in a dataset of 25 NB cell lines, colored by their ChIP-seq signature defined subgroup (p -value= $8.39e-5$, R -value= 0.774 , Spearman correlation). (Right) Correlation of SOX11 expression with super-enhancer signal of the consensus super-enhancer in a dataset of 47 NB tumors, colored by their ChIP-seq signature defined subgroup (p -value= $1.25e-10$, R -value= 0.778 , Pearson correlation).

Adjusted Supplemental Figure 2a-c:

Adjusted legend Supplemental Figure 2: **A.** H3K27ac activity for the region immediately downstream of *SOX11* (chr2, 5.7–7.0Mb, hg19) in 10 adrenergic MYCN amplified cell lines (SMS-KCNR, BE-2C, IMR-32, IMR-5/75, KELLY, LAN2, NGP, NMB, SK-N-DZ and TR14). **B.** H3K27ac activity for the region immediately downstream of *SOX11* (chr2, 5.7–7.0Mb, hg19) in 9 adrenergic MYCN non-amplified cell lines (CHLA15, CHLA20, CLB-GA, CHLA90, LAN6, NB69, NBL-S, SK-N-FI and SY5Y). **C.** H3K27ac activity for the region immediately downstream of *SOX11* (chr2, 5.7–7.0Mb, hg19) in 4 mesenchymal NB cell lines (GI-ME-N, HD-N-33, SH-EP and SK-N-AS) and 2 non-malignant neural crest cells (P4 and P5). For figure A-C, signal represents RPKM normalised ChIP signal, super-enhancers are annotated using ROSE (orange bar).

Reviewer 1, Major Question 3: The indication to discuss the *SILC1* ncRNA in figure 2 is unclear – the authors demonstrate that it contains similar expression as *SOX11*, which would be expected if it was co-regulated by similar enhancers. Is there functional significance to *SILC1* expression? Do the authors contend that, as in development, *SILC1* expression is required for *SOX11* expression? If so, then the authors should perform genetic disruption of *SILC1* and examine *SOX11* expression levels, in addition to expression of other CRC members – If disrupted, then *SILC1* could be functioning as a ncRNA CRC regulator, as previously demonstrated for other ncRNA species by Suzuki, Sharp and Young (Cell 2017). In the absence of mechanistic data such as this to implicate *SILC1* in the regulation of *SOX11* expression, this observation is somewhat confusing, and I would suggest the authors remove this distracting information.

3. Author's response: Given that *SILC1* does not overlap with the consensus SE for *SOX11* reported by Gartlgruber et al. and current lack of functional data to support enhancer activity of *SILC1*, we agree to follow the suggestion to remove this section from the manuscript. For the 4C-seq analysis figure

2e, we retain the *SOX11* interaction profile but remove the generated reciprocal 4C-seq data using the *SILC1* locus as viewpoint as shown below.

Adjusted Figure 2e:

Adjusted legend Figure 2E: 4C-seq analysis of the promoter site and (super-)enhancer region downstream of *SOX11* (chr2, 5.6-7.1Mb, hg19) in the NB cell lines CLB-GA (blue) and SH-EP (orange) with inclusion of published and unpublished ChIP tracks for H3K27ac, H3K4me1, PHOX2B, HAND2 and GATA3 in CLB-GA, and ATAC and H3K4me3 in SH-EP. Signal represents log likelihood ratio for the ChIP signal as compared to the input signal (RPM normalised). The viewpoint is located at the *SOX11* transcription start site and *SILC1* TSS site (cut out 100 kb). Interaction peaks called by PeakC are shown underneath 4C-seq data, CLB-GA (blue) and SH-EP (orange). Super-enhancers of CLB-GA are annotated using ROSE (orange bar).

Reviewer 1, Major Question 4: The authors should present the Depmap data they refer, at least in a supplementary figure.

We added a Supplementary Figure 3A presenting the DEPMAP data to the manuscript.

Adjusted Figure 3a:

Adjusted legend Supplemental Figure 3A: *SOX11* is characterized as a strong selective gene according to a public available CRISPR screen in 1086 cell lines (CRISPR 22Q2 Chronos) with dependency in 25/34 NB cell lines (CERES < -0.1), and significantly selective for NB ($p=2.5e-40$) and more specifically in 11/15 MYCN amplified NB cell lines ($p=7.4e-29$). Figure and citation from <https://depmap.org/portal>: "Two-group comparisons were performed in parallel across genes/compounds using the Limma R package, which uses parametric empirical Bayes methods to pool information across genes when assessing the significance of observed group differences. P-values for each gene are computed from empirical Bayes moderated t-statistics. Enriched lineages are those with p-value < 0.0005."

As the CRISPR screen was updated since manuscript submission, with more cell lines, we adjusted the text in the manuscript accordingly:

According to the publicly available CRISPR screen data in 1086 cell lines (CRISPR 22Q2 Chronos, available via the DepMap Portal), *SOX11* is identified as a strongly selective gene with dependency in 25 NB cell lines and significantly selective for NB (\$p=2.5e-40\$ ) and more specifically in 11 MYCN amplified NB (\$p=7.4e-29\$ ) cell lines (Fig. S3a).

Reviewer 1, Major Question 5: *In Figure 3, the authors show that SOX11 knockdown by either shRNA or siRNA results in a growth phenotype, with an induction of a G1 cell cycle arrest in Figure 3. In Figure 4, however, they demonstrate effects on RNA expression associated with changes in cell adhesion, migration and motility. Are these transcriptional changes related? The authors should examine whether SOX11 loss causes changes in migratory or metastatic behaviors of their cell lines, as they undergo change in fate. Similarly, are the effects on mRNA processing and translation functional? While loss of a lineage-specifying transcription factor is likely to result in a variety of changes, it should be made clear which changes have functional outcomes, and which are “transcriptional bystanders” and do not reflect the altered biology of the cell.*

5. Authors’ response: In order to further support the predicted *SOX11* functions, we generated additional transcriptome data after *SOX11* knockdown in CLB-GA and NGP NB cells and re-analyzed our data and performed further functional experiments and in depth analysis towards identifying *SOX11* driven functions. In addition, as an orthogonal experiment, we also performed additional transcriptome profiling for doxycycline-induced *SOX11* expression at an earlier time point (9h after induction in addition to the 48h timepoint) in mesenchymal SH-EP NB cells which normally do not express *SOX11*. This allows to enrich for direct or early *SOX11* regulated targets (see further question 6 of reviewer 2) for further exploration of putative *SOX11* driven cellular functions. Our new data unequivocally confirmed the initially predicted functions of "adhesion/motility/cytoskeleton" and "epigenetic regulation" while "translational initiation" was not observed in CLB-GA and NGP cells suggesting this was caused through transcriptional bystander effects or through context specific features of IMR-32 cells (see Fig.4 and Fig.S4). To test the predicted effect on cell migration, we performed scratch wound migration assays upon *SOX11* overexpression in SH-EP cells 18h after *SOX11* overexpression and evaluated the impact of *SOX11* on migration. We observed increased migration based on the scratch width measurements (Fig. 3b-c). Importantly, we also included the reference of a recent study⁵ showing that ADRN-type NB cells exhibit RAC1- and kalirin-dependent nucleokinesis (NUC) migration, which relies on several integral components of neuronal migration, including DCX and *SOX11*. The authors demonstrate decreased migration upon *SOX11* knockdown in IMR-32 derived 3D-spheroids as well as decreased closure of nuclear center-cell center distance along with DCX downmodulation, suggesting that *SOX11* knockdown causes a slower, nucleokinesis-independent migration mode. The data further indicate that *SOX11* knockdown fosters morphological asymmetric cell divisions and causes reprogramming of nucleokinesis migration in ADRN-type NB cells. Taken together, these findings support our additional observations on the effect of *SOX11* on NB cell migration.

For integration of the additional transcriptome data after *SOX11* knockdown in CLB-GA and NGP NB cells and transcriptome profiling for doxycycline-induced *SOX11* expression at an earlier time point (9h after induction) in mesenchymal SH-EP NB cells, see chapter "The SOX11 regulated transcriptome is involved in epigenetic control, cytoskeleton and neurodevelopment" and Figure 4 and S4.

In line with the new findings on the effect of *SOX11* on cell migration, the manuscript was adjusted as shown below:

Using these gene sets, we then sought for enrichment for cellular functions to gain insight into the presumed contribution of SOX11 expression to the high-risk NB adrenergic phenotype. Gene set enrichment analysis for the SOX11 late regulated genes revealed strong enrichment for axon

outgrowth, neural crest cell migration, cytoskeleton (Fig. 4c-d, S4e). In line with this, Afanasyeva et al. demonstrated decreased migration upon SOX11 knockdown in IMR-32 spheroids, further showing that SOX11 knockdown fosters morphological asymmetric cell divisions and causes reprogramming of nucleokinesis migration in ADRN-type NB cells. To functionally validate the predicted role in control of cytoskeleton and cell migration by our SOX11 overexpression RNA-sequencing data, we performed wound-healing assays confirming that SOX11 overexpression enhances wound healing capacity and migration potential SH-EP NB cells (Fig. S3b-c).

Adjusted Material and Methods section: Cell viability was measured in real-time using the IncuCyte by taking photos every 3 hours of the entire well (4x). Masking was done using the IncuCyte® ZOOM Software. For the scratch wound migration assay, cells were seeded as described above in an Imagelock 96-well plate (4379, Essen Bioscience) and a scratch wound was made in a confluent cell monolayer using a 96-well Incucyte® Wound Maker. Phase contrast imaging took place every 3h. Images were analysed using IncuCyte® S3TM 2018B-2019A software. For statistical testing, a Levine’s test was performed in SPSS at the 1% significance level upon which an independent paired t-test was performed at the 5% significance level.

Adjusted Figure 3b-c:

Adjusted Figure Legend: Fig.3 B. Induced wound healing capacity 18h upon SOX11 overexpression in SH-EP cells. The fluorescence marker ZsGreen is visible when SOX11 expression is induced. The purple color indicates the migrated cells in the scratch over time, computed with the IncuCyte® ZOOM Software. C. Confluency (%) (up, \$p=0.016\$, \$p=0.036\$ and \$p=0.038\$ respectively) and scratch width (down, \$p=0.024\$, \$p=0.093\$ and \$p=0.16\$ respectively) upon SOX11 overexpression in SH-EP cells using 3 different monoclonal expansions (3 independent biological replicates). The error bars represent the 95% confidence interval and the average of three independent technical replicates.

Reviewer 1, Major Question 6: *It is important that the authors provide more evidence for SOX11 binding. They have conducted ChIP-seq to SOX11 in one cell line (IMR-32), and in order to make formal claims about locations of SOX11 binding, it is important to see this data in more cell lines. To this end, an analysis of where SOX11 binds, distinguishing promoters, typical enhancers, super-enhancers and gene bodies would be helpful. The overlap between H3K4me3 and SOX11 binding would indicate that SOX11 is primarily a promoter binding factor – is this true? The authors also speculate that SOX11 may co-bind with other master TFs in neuroblastoma, such as HAND2, TWIST1, TCF3, ASCL1. Much of this data is publicly available, and the authors should perform further ChIP studies in cell lines where publicly available data exists, in order to correlate.*

6. Authors' response: We agree with the reviewer that further data on SOX11 binding in different NB cell lines would be valuable as well as further mining versus regulatory regions (promoters, enhancers) and other TF binding patterns (in particular in view of the reported CRCs).

6.1. To this end, we explored an optimized CUT&RUN protocol and generated additional SOX11 DNA binding data for adrenergic *MYCN* amplified (IMR-32, NGP) and *MYCN* non-amplified cell lines (CLB-GA) as well as a mesenchymal SOX11-negative cell line after *SOX11* overexpression (SH-EP SOX11 overexpression 48h). Based on these new data, we were able to confirm the initial SOX11 ChIP-seq data in IMR-32 with significant overlap of SOX11 binding sites in all cell lines tested, with 3984 common SOX11 binding sites detected by SOX11 CUT&RUN (Fig. 5a, S5a). Importantly, to provide further confidence, we also further validated the ChIP-seq targets by correlation with the newly generated RNA-seq datasets showing enrichment of SOX11 binding at the transcription start site of differentially regulated targets (Fig. 5b-c). Note that the density plots for early regulated SOX11 targets genes in SH-EP cells are more noisy due to the low number (669) of early induced genes while at 48h after SOX11 induction density plots are more robust due to higher number (9258) of differential genes.

6.2. To look further into SOX11 DNA binding positions in relation to coding and regulatory sequences, genome-wide peak and gene annotation distribution was performed for the overlapping SOX11 CUT&RUN peaks called in IMR-32, CLB-GA, NGP cells and SH-EP cells after *SOX11* overexpression for 48h (Fig. S5c). Indeed, the majority of SOX11 binding sites is located on protein-coding loci both at promoter-TSS as well as intronic regions, confirming that SOX11 is primarily a promoter binding factor.

6.3. Finally, we scrutinized for binding for co-binding of SOX11 with other master TFs in NB. For this we used publically available datasets for *MYCN* (E-MTAB-6570), *HAND2* (GSE90683), *GATA3* (GSE90683), *PHOX2B*(GSE90683), *ASCL1* (GSE159613), and *TWIST1* (GSE80151). For *TCF3*, no dataset was available for NB. Co-binding of *MYCN*, *ASCL1* and *TWIST1* at SOX11 bound enhancers and promoters can be observed as well as *HAND2*, *GATA3* and *PHOX2B* co-binding at SOX11 bound enhancers (Fig. 7d). Using our new RNA-seq and CUT&RUN datasets we further dissected the role of SOX11 in adrenergic CRC. This further discussed in minor question 7 of reviewer 1.

Taken together, we have been able to provide the requested additional SOX11 binding data and have re-analyzed our data sets and provided new figures to present all these data as indicated here below. For integration of the new CUT&RUN datasets, see chapter "SOX11 directly regulates regulates multiple major modulators of the epigenome including the SWI/SNF remodeling complex" and Figure 5 and S5.

Adjusted Figure 5a-c:

Adjusted Figure Legend Figure 5: **A.** Overlap (min. overlap = 20 bp) of the SOX11 CUT&RUN peaks (MACS2 peakcalling $qval < 0.05$) in IMR-32, CLB-GA, NGP cells and SH-EP cells after SOX11 overexpression for 48h (Fisher test p -value $< 2.2e-16$). **B.** Heatmap profiles -2 kb and $+2$ kb around the transcription start site of early and late SOX11 targets in SH-EP, subdivided in upregulated and downregulated genes. On these regions the SOX11 CUT&RUN data in IMR-32, CLB-GA, NGP and SH-EP cells after SOX11 overexpression for 48h are mapped and ranked according to the sums of the peak scores across all datasets in the heatmap. **C.** Overlap of common SOX11 CUT&RUN peaks (common peaks in IMR-32, CLB-GA, NGP cells and SH-EP cells after SOX11 overexpression for 48h) and early and late up- and downregulated genes in SH-EP after SOX11 overexpression for 48h and overlap of common SOX11 CUT&RUN peaks with genes positively and negatively correlated with SOX11 expression in NB tumor cohort ($n = 649$, GSE45547, p -value < 0.05).

Adjusted Figure 5a,d:

Adjusted Figure Legend Figure S5: A. Heatmap profiles -2 kb and $+2$ kb around the summit of SOX11 CUT&RUN peaks in IMR-32, grouped for promoters or enhancers (homer annotation). On these regions the SOX11 CUT&RUN data in IMR-32, CLB-GA, NGP and SH-EP cells after SOX11 overexpression for 48h as well as SOX11 ChIP-seq data in IMR-32 and data is ranked according to the sums of the peak scores across all datasets in the heatmap. **D.** Genome-wide peak and gene annotation distribution (%) (Homer annotation) for the overlapping SOX11 CUT&RUN peaks called in IMR-32, CLB-GA, NGP cells and SH-EP cells after SOX11 overexpression for 48h (MACS2, $q < 0.05$).

Adjusted Figure 7d:

Adjusted Legend Figure 7: D. Heatmap profiles -2 kb and $+2$ kb around the summit of SOX11 CUT&RUN peaks in IMR-32, grouped for promoters or enhancers (homer annotation). On these regions MYCN, HAND2, GATA3, PHOX2B, ASCL1 and TWIST ChIP data is mapped and ranked according to the sums of the peak scores across all datasets in the heatmap.

Adjusted Material and Methods section:

CUT&RUN assay

CUT&RUN coupled with high-throughput DNA sequencing was performed using Cutana pA/G-MNase (Epiccypher, 15-1016) according to the manufacturer's protocol. Briefly, cells (0.5M cells/sample) were washed and incubated with activated Concanavalin A beads for 10 min at room temperature. Cells were then resuspended in antibody buffer containing 0.01% digitonin, 1:100 dilution of each antibody (anti-SOX11, HPA000536; IgG goat, sc-2028) was added to individual cell aliquots and tubes were rotated at 4°C overnight. The following day, targeted chromatin digestion and release was performed with 2.5 mL Cutana pA/G-MNase and 100mM CaCl₂. Retrieved genomic DNA was purified with the MinElute PCR purification kit and eluted in 10 mL of buffer EB. Sequencing libraries were prepared with the automated Swift 2S system, followed by 100bp-PE sequencing with Novaseq 6000.

Reviewer 1, Major Question 7: *The authors identify that SOX11 is required for opening of sites associated with adrenergic neuroblastoma in figure 6. This data is striking and well performed. Do the authors content that this effect is direct (i.e. SOX11 interacts physically with nucleosome remodeling complex members, to recruit them to maintain these sites?). If so, then the authors should provide evidence that SOX11 physically interacts with members of the nucleosome remodeling complex. If not, then how do the authors suggest that this occurs? If SOX11 itself has nucleosome remodeling activity in neuroblastoma, this should be identified by nucleosome assays.*

7. Authors' response: To identify nucleosome remodeling complex members (or other factors that could act with SOX11 on chromatin accessibility, we performed IP-MS in two SOX11 expressing NB cell lines, MYCN amplified NGP cells and MYCN non-amplified CLB-GA cells. This revealed 29 putative interacting proteins (FDR < 0.01 and |log₂FC| between SOX11 and IgG IP > 1) present in two cell lines in either one or more replicate experiments (Rebuttal Figure 2). No members of the four major remodeling complexes are present. Interestingly, we did observe the presence of WDHD1 (AND-1) as commonly bound protein. WDHD1 (AND-1) is an acidic nucleoplasmic DNA-binding protein and a high mobility group domain-containing protein with remarkable capability to regulate the stability of histone H3 acetylase KAT2A (GCN5) which is a known MYC transcriptional co-activation factor and also a critical regulator of chromatin remodeling⁶. Further follow up experiments are needed to understand how SOX11-WDHD1 interaction can impact on KAT2A (GCN5) activity but we speculate that along with its proposed pioneering activities, SOX11 could stabilizes KAT2A (GCN5) through WHDH1 (AND-1) binding thus facilitating MYCN driven transcriptional activity. In a follow-up paper, we will further functionally explore the SOX11 interactome.

A.

Rebuttal Figure 2: A. Venn diagram presenting the overlap of the IP-MS results for the 2 cell lines NGP and CLBGA, with 2 biological replicated experiments for each cell line (existing each out of three technical replicates per condition). Only when $FDR < 0.01$ and $|\log_2FC|$ between SOX11 and IgG IP > 1 , the proteins are taken into account as interactors. Nine consensus interactors are overlapping between these two cell lines. **B.** Volcanoplot for each IP-MS experiment (2 times CLB-GA, 2 times NGP). For each experiment 3 biological replicates of SOX11 IP and IgG IP are taken along.

Reviewer 1, Major Question 8: Two other recent papers on BioRxiv have suggested that loss of SOX11 is associated with increased SOX4 expression - if SOX11 is required to maintain open chromatin sites, then is SOX4 or some other SOX member upregulated to permit accessibility in mesenchymal subtype neuroblastoma? This comparison may be easy to make, by simply comparing RNAseq expression of SOX family members in different cell lines (adrenergic vs. mesenchymal) and then relating this to effects with SOX11 knockdown and/or overexpression. One might expect that since SOX11 knockdown cells do not become “fully mesenchymal” that a SOX transcription factor required to open chromatin in mesenchymal neuroblastoma is not induced.

8. Authors' response:

We thank the reviewer for suggesting this analysis. First, we looked into the expression of SOX TF family members in the adrenergic and mesenchymal isogenic cell line pairs defined by van Groningen et al.⁷, in order to identify differentially expressed SOX genes. SOX13 was significantly higher expressed in MES cells while SOX9 (reported in the van Groningen paper⁷ as MES marker in Fig.3 of their paper and strongly upregulated in ARID1A deficient NGP cells exhibiting ADNRN to MES transition⁸) showed a trend towards increased expression, albeit not significant (Rebuttal Figure 3a). Next, we evaluated gene expression levels for all SOX TFs in ADNRN NB cell lines following SOX11 knockdown and observed upregulation for SOX4 and SOX2 (although only significant in IMR-32, adjusted pvalue = 0.0025 for SOX4 and 0.034 for SOX2, Rebuttal Figure 3b-c, Supplementary Table 2). This is in keeping with the compensatory upregulation of SOX4 reported by the papers of the Thiele³ and Look team⁹. SOX2 is a known stem cell marker and known downstream target of the NPM-ALK fusion protein in T-cell lymphoma¹⁰. SOX9 and SOX13 are not differentially expressed (Rebuttal Figure 3b-c, Supplementary Table 2) which may reflect that adrenergic NB cells are refractory to full MES transition following SOX11 knockdown.

Of further interest, SOX9 and SOX13 are both upregulated upon SOX11 overexpression for 48h in SH-EP (adjusted p-value = 0.0015 for SOX9 and 6.16e-05 for SOX13, Rebuttal Figure 3d), suggesting this could act as a compensatory mechanism for SOX11 overexpression towards maintaining the MES cell identity. On the other hand, additional analyses at protein level are warranted to explore this further

as e.g. we found midkine (MDK) as one of the strongly induced SOX11 target genes (Fig. 4c) which has been shown to stimulate degradation of SOX9¹¹. Consequently, SOX9 mRNA upregulation may be compensatory to MDK mediated SOX9 protein degradation.

Taken together, we do find opposite regulation for SOX11 and SOX4, in keeping with the data of Banerjee et al.³ and Zimmerman et al.⁹, but further investigations are required to unravel putative roles of SOX9, SOX13 or other TF co-factors implicated in ADRN-to-MES transition in NB, which does not fall within the scope of this manuscript.

Rebuttal Figure 3: A. SOX13 and SOX9 expression (log₂) in adrenergic and mesenchymal isogenic cell line pairs defined by Van Groningen et al.⁷. **B.** Tscore representing differential expression of SOX TFs in RNA-sequencing data after SOX11 knockdown in IMR-32, CLB-GA and NGP. Significant adjusted p-values are indicated with coloured bars. **C.** Log₂ mRNA expression of SOX transcription factors upon SOX11 knockdown in IMR-32, CLB-GA and NGP represented as a row wise z-score in a heatmap. Heatmap color reflects row-wise z-score. Genes that are underlined are differentially regulated. **D.** SOX9 and SOX13 log₂ mRNA expression levels upon SOX11 overexpression for 48h in SH-EP cells. Error bars and circles represent respectively the standard deviation and mean of the three biological replicates. Statistical analysis with moderated t-test of Limma.

Minor concerns:

Reviewer 1, Minor Question 1: The authors demonstrate that SOX11 is coamplified with MYCN, and that higher expression goes with poorer survival. It would be helpful to perform similar correlational analyses with age and stage.

1. Authors' response: As requested by the reviewer, we investigated the expression of SOX11 according to MYCN status, age group (< or >= 1 year) and INSS stage in both the NRC cohort (Rebuttal Fig. 4) and the Fischer cohort (Rebuttal Fig. 5). In the NRC cohort (GSE85047, n=283), we find a significant higher expression of SOX11 in patients with MYCN amplification, in patients with age higher

or equal to 1 year, and we find *SOX11* higher expressed in patients from the stage 4 group as compared to stage 2 group but other comparisons were not significant (ANOVA and post-hoc Tuckey test).

Rebuttal figure 4: **A.** *SOX11* is higher expressed in *MYCN* amplified tumors as compared to *MYCN* non amplified tumors ($pval=4.716e-10$, t-test, 6 samples omitted due to unknown *MYCN* status). **B.** *SOX11* is higher expressed in tumors from patients older than 1 year ($pval=0.01224$, t-test, 5 samples omitted due to unknown age status). **C.** *SOX11* is higher expressed in tumors from stage 4 as compared to tumors from stage 2 ($pval=0.044$, anova and post-hoc tukey test. No other comparisons were significantly different).

In the Fischer cohort (GSE62546, $n=498$), we find a significant higher expression of *SOX11* in patients with *MYCN* amplification, in patients with age higher or equal to 1 year, and we find *SOX11* to be higher expressed in patients from the stage 4 group as compared to all other stages (ANOVA and post-hoc Tuckey test).

Rebuttal figure 5: **A.** *SOX11* is higher expressed in *MYCN* amplified tumors as compared to *MYCN* non amplified tumors ($pval < 2.2e-16$, t-test, 5 samples omitted due to unknown *MYCN* status). **B.** *SOX11* is higher expressed in tumors from patients more than or equal to 1 year as compared to patients less than 1 year ($pval=0.0073$, t-test). **C.** *SOX11* is higher expressed in tumors from stage 4 as compared to tumors from all other stages (Anova and post-hoc tukey test. Pval: St4-St1: 0.0, St4-St2a: $3.16e-04$, St4-St2b: $1.86e-04$, St4-St3: $6.41e-03$, St4-St4s: $5.42e-05$. No other comparisons were significantly different).

Reviewer 1, Minor Question 2: Higher expression levels of *SOX11* both at mRNA and protein level were observed in adrenergic NB cell lines compared to mesenchymal NB cell lines and tumors (Fig. 1g, Fig. S1j-k) – should be softened – in Fig 1g, the expression of *SOX11* is high regardless of MES or ADRN status (10-14 log₂)

It has to be noted that Fig. 1g only contains one true mesenchymal cell line (SH-EP) for which there is no expression ($\log_2 SOX11 = 1.3$). When removing the SH-EP cell line from this plot, there is indeed no significant correlation between MES/ADR status and SOX11 expression anymore. The expression ranges from 9.5-15.4 \log_2 , which means that the cell line with highest SOX11 expression has 64 times higher SOX11 expression than the cell line with lowest SOX11 expression. But only adrenergic cell lines are left, so MES/ADR status does not have a real meaning anymore.

2. Authors' response: We thank the reviewer for this comment and agree to modify this section. We changed Figure 2 including H3K27ac data of 60 NB tumors and 27 cell lines from Gartlgruber et al.². We also evaluated SOX11 expression in 47 NB tumors subdivided in groups based on H3K27ac super-enhancer profiling (MYCN amplification, high-risk MYCN non-amplified and low-risk MYCN non-amplified group), and found SOX11 to be highly expressed in MYCN amplified and MYCN non-amplified NB tumors as compared to mesenchymal tumors. To address the reviewer concerns, we removed Fig. 1G and replaced it by the figure below:

Adjusted Figure 1g:

Adjusted legend Figure 1: G. SOX11 (\log_2) expression in four H3K27ac profiling based groups identified in NB tumors: MYCN-amplified (MNA), high-risk MYCN non-amplified (MNoA-HR), low-risk MYCN non-amplified group (MNoA-LR) and mesenchymal (MES). SOX11 is higher expressed in MNoA-HR, MNoA-LR and MNA groups as compared to MES group (Anova and post-tuckey test, significant comparisons: MNoA-HR vs MES $p=4e-04$, MNoA-LR vs MES $p=2e-03$, MNA vs MES $p=2e-04$).

In addition, we found SOX11 to be higher expressed in adrenergic subtype derived from isogenic NB cell lines (4 pairs) as compared to mesenchymal subtype. We added the Figure below to Supplementary Figure 1L:

Adjusted Figure S1l:

Adjusted legend Supplementary Figure 1: L. SOX11 \log_2 expression in the adrenergic subtypes (blue) derived from the isogenic NB cell lines compared to the mesenchymal subtypes (orange) (n=4 pairs, connected by dotted lines).

We have adjusted the manuscript accordingly:

Higher expression levels of SOX11 both at mRNA and protein level were observed in MNA and MNoA NB tumors as compared to tumors with mesenchymal super enhancer signature (Fig. 1g) as well as adrenergic compared to mesenchymal NB cell lines and tumors (Fig. S1j-l).

Reviewer 1, Minor Question 3: *The title of Figure 3 should be revised, the data does not reflect that SOX11 is a lineage factor.*

3. Authors' response: We agree with the reviewer that Figure 3 does not reflect SOX11 as a lineage factor and change the title of Figure 3 and chapter 3 to: "SOX11 is a dependency factor in adrenergic NB cells".

Reviewer 1, Minor Question 4: *The extensive heatmap analysis in Figure 4a/b is of limited utility and should be supplemented. In contrast, Supplemental figure 4a and b, d and f are extremely helpful to understanding the primary goal of the figure, and should be brought to the main figure.*

4. Authors' response: We adapted Fig.4 as suggested by the reviewer and following inclusion of our new RNA-seq data (see also question 5).

Reviewer 1, Minor Question 5: *The authors provide evidence for broad dysregulation of BAF complex transcription but do not describe it in the text. They should postulate on how SOX11 regulates BAF complexes, especially in the context of the recent evidence for BAF complex members in the regulation of an adrenergic-mesenchymal switch (Shi H et al Science Advances 2020).*

5. Authors' response: This is a very important comment, especially in the light of the additional SOX11 DNA binding data and additional transcriptome data that were generated for the rebuttal. Remarkably, based on the initial IMR-32 ChIP-seq data and new CUT&RUN data we found that SOX11 binds to 21 SWI/SNF components of the three known SWI/SNF complexes (c-BAF, nc-BAF, p-BAF). These were found in at least two out of four investigated cell lines with 13 components found as targets in all four cell lines investigated (Fig. S4g). Next, transcriptional regulation upon perturbation of SOX11 levels revealed 20 out of 29 known SWI/SNF components were differentially upregulated after SOX11 overexpression, 16 of which were also differentially downregulated after SOX11 knockdown in at least one NB cell line. Interestingly, 12 components are already differentially upregulated at the 9h time point after SOX11 overexpression, 10 of which are also bound by SOX11 in all cell lines (Fig. S4g). This strongly suggests a direct and early role of SOX11 on the expression of the SWI/SNF complex which is also indirectly supported by the occurrence of SOX11 and SWI/SNF component germline loss-of-function mutations as cause for the neurological Coffin-Siris syndrome^{12,13}. Of further interest, SMARCC1 and SMARCA4, both regulated by SOX11, have been shown to play a critical role in maintenance of distal lineage specific enhancers¹⁴.

Next, as suggested by the reviewer and to gain insight into how SOX11 impacts on SWI/SNF activity, we looked into the public available binding data of SMARCA4 in the NB cell line NGP⁸. We observed enrichment of SMARCA4 binding sites in late SOX11 targets but not in early SOX11 targets (Fig. 4g). This observation could be in keeping with impact on epigenetic regulation of gene activity at SOX11 targets following upregulation of the SWI/SNF remodeling machinery. Of further interest, our ATAC-sequencing data after SOX11 overexpression in SH-EP cells (see question 4 of reviewer 2), yield enrichment for SMARCC1 binding motifs in differential ATAC peaks (Fig. 6c).

Finally, Shi H et al. showed that ARID1A knockout in NGP cells causes downregulation of SOX11 expression and, reversibly, we observe downregulation of ARID1A after SOX11 knockdown (Fig. S4g). As ARID1A knockout in NGP cells also caused an ADRN-to-MES transition, we furthermore investigated the impact of SOX11 knockdown on the expression levels of ADRN and MES CRC components. In Fig.

7c and S7a, we show the overall effects on these CRC members across the different ADRN cell lines upon SOX11 knockdown revealing significant downregulation of several key ADRN TFs (*TCF3*, *ISL1*, *PHOX2B*, *TFAP2B*, *MYCN* and *KLF7*) but no differential expression of MES TFs (Fig. 7c, Fig. S7a, Supplementary Table 2, and see also minor question 7 of reviewer 1). Hence full adrenergic-to-mesenchymal transitions and establishment of the mesenchymal CRC is not observed after *SOX11* knockdown, suggesting knockdown of *SOX11* on its own is not sufficient to induce lineage switch.

In conclusion, we postulate that SOX11 controls chromatin accessibility by modulating the expression of the SWI/SNF chromatin remodeling complex. Further experiments *e.g.* by performing knockdown/out of *SMARCC1*, *SMARCA4* and *ARID1A* in combination with additional CUT&RUN experiments are needed to shed more light onto their role of the SWI/SNF complex as SOX11 regulated targets.

Given these new findings we have adapted the manuscript and figures accordingly:

The SOX11 regulated transcriptome is involved in epigenetic control, cytoskeleton and neurodevelopment

Interestingly, when investigating the impact of SOX11 on all 29 known SWI/SNF components of the three known SWI/SNF complexes (c-BAF, nc-BAF, p-BAF), we observed that 20 out of 29 known SWI/SNF components are differentially upregulated after SOX11 overexpression, 16 of which are also differentially downregulated after SOX11 knockdown in at least one NB cell line. Moreover, 13 components are already differentially upregulated at the 9h time point of SOX11 overexpression, of which 10 are directly bound by SOX11 (see further) (Fig. 4f). This strongly suggests a direct and early role of SOX11 on the expression of the SWI/SNF complex.

SOX11 directly regulates multiple major modulators of the epigenome including the SWI/SNF remodeling complex

In order to gain initial insights into the impact of the broad upregulation by SOX11 of most SWI/SNF complex components, we furthermore analysed SMARCA4 binding sites in SOX11 regulated genes and observed enrichment for SMARCA4 DNA binding only in late SOX11 regulated genes (Fig. S5j). Although further studies are warranted, this could suggest that enhanced SWI/SNF activity associated with the observed SMARCA4 binding acts downstream of SOX11 to contribute to its induction of a gene regulatory program.

Forced SOX11 overexpression in SH-EP NB cells impacts genome wide chromatin accessibility

We furthermore see high enrichment in the differential ATAC peaks for a TGA(G/C)TCA motif known to be bound by several transcription factors of the bZIP family including JUN, ATF3, FOSL1, FOSL2 and SWI/SNF component SMARCC1 (Fig. 6c). As mentioned previously, SMARCC1 is a direct SOX11 target with induced expression upon SOX11 overexpression and therefore a possible SOX11 controlled mediator of chromatin accessibility at these sites.

Adjusted Supplemental Figure 4:

g

Expression SWI/SNF components after SOX11 perturbation

Bold and underlined = bound by SOX11 in IMR-32, CLB-GA, NGP and SH-EP DOX

Adjusted legend Supplemental Figure 4: G. *Log(FoldChange) of SWI/SNF components upon SOX11 knockdown in IMR-32, CLB-GA and NGP as well as SOX11 overexpression after 9h and 48h in SH-EP represented in a heatmap. Heatmap color reflects row-wise z-score. Bold and underlined represents genes that are bound by SOX11 in IMR-32, CLB-GA, NGP and SH-EP after SOX11 overexpression for 48h.*

Adjusted Supplemental Figure 5:

j

Adjusted legend Supplemental Figure 5: J. *Enrichment of top 200 SMARCA4 target genes as defined by ChIP-sequencing in SOX11 early and late regulated genes respectively.*

Adjusted Figure 6:

C.

Adjusted legend Figure 6: C. *Motif enrichment for open and closed regions determined by ATAC-seq upon SOX11 overexpression for 48h in SH-EP cells.*

Reviewer 1, Minor Question 6: *There appears to be no supplementary figure 3.*

6. Authors' response: This is corrected in the manuscript, we added the missing supplementary Figure 3.

Reviewer 1, Minor Question 7: *The authors comment that SOX11 is not a member of the extended regulatory network (ERN), however, do not test this. An expressed transcription factor, whether it is super-enhancer regulated or not, is, by definition, a member of the ERN. It is without doubt that the CRC factors bind to the SOX11 promoter and enhancer, resulting in regulation of it – this makes SOX11 at least a member of the ERN.*

7. Authors' response: In view of the newly obtained SOX11 CUT&RUN (see major question 6 of reviewer 1) and RNA-seq data (see major question 5 of reviewer 1), we were now able to conduct more in depth analyses concerning the role of SOX11 as putative master transcription factor in the adrenergic core regulatory circuitry.

First we observed binding of SOX11 to its own promoter and downstream enhancer landscape, including the called consensus SOX11 super-enhancer (see major question 2 of reviewer 1) (Fig. 7a). **Second**, binding of major CRC members including HAND2, PHOX2B and GATA3 was observed at the SOX11 promoter, and HAND2, GATA3, MYCN, ASCL1 and TWIST1 bind the downstream enhancer landscape including the consensus SOX11 super-enhancer (Fig. 7a). **Third**, SOX11 binding is observed at the promoters of HAND2, PHOX2B, GATA3, ASCL1, TWIST1, and TCF3 (Fig. 7b). **Fourth**, as reported for other CRC members, SOX11 knockdown causes partial CRC collapse notified by attenuated expression of several CRC members such as TCF3, ISL1, PHOX2B, TFAP2B, MYCN and KLF7 (Fig. 7c, S7a, Supplementary Table 2). **Fifth**, co-binding of MYCN, ASCL1 and TWIST1 at SOX11 bound enhancers and promoters can be observed as well as HAND2, GATA3 and PHOX2B co-binding at SOX11 bound enhancers (Fig. 7d). More specifically, we observe strong co-binding of MYCN, ASCL1 and TWIST1 at the transcription start site of early and late regulated SOX11 targets (Fig. S7b). According to the proposed criteria for master TFs contributing to CRCs, these data establish SOX11 as an adrenergic neuroblastoma CRC master transcription factor¹⁵.

Importantly, in the original manuscript, we specifically focused on the relationship between SOX11 and MYCN in the section "SOX11 acts in concert with MYCN to regulate a subset of downstream targets". While this statement is still true, given the findings reported above, using our new data we were now able to show convincingly that SOX11 acts in concert with the broader adrenergic core regulatory circuitry and not only specifically with MYCN. For this reason we decided to take out the section on the relationship with MYCN, rewrite it and integrate it with the chapter "SOX11 is a core regulatory circuitry master transcription factor in adrenergic NB".

Given these new findings, we adapted the manuscript and figures as indicated below:

SOX11 is a core regulatory circuitry master transcription factor in adrenergic NB

Our SOX11 DNA binding analysis also provided further insight into proposed role of SOX11 as core regulatory circuitry (CRC) master transcription factor. CRCs are a group of interconnected auto-regulating transcription factors that form loops and can be identified by super-enhancers. Our data suggest that SOX11 is a bona fide adrenergic NB CRC member. First, SOX11 binds its own promoter and also binds the SOX11 3' downstream enhancer landscape, including the above mentioned consensus SOX11 super-enhancer (Figure 7a). Second, binding of major CRC members including HAND2, PHOX2B and GATA3 was observed at the SOX11 promoter. Also, HAND2, GATA3, MYCN, ASCL1 and TWIST1 bind the downstream enhancer landscape including the consensus SOX11 super-enhancer (Fig. 7a). Third, SOX11 binding is observed at the promoters of HAND2, PHOX2B, GATA3, ASCL1, TWIST1, and TCF3 (Fig. 7b). Finally, as reported for other CRC transcription factor members,

SOX11 knockdown causes partial CRC collapse notified by attenuated expression of several CRC members such as *TCF3*, *ISL1*, *PHOX2B*, *TFAP2B*, *MYCN* and *KLF7* (Fig. 7c, Fig. S7a, Supplementary Table 2). However full adrenergic-to-mesenchymal transitions and establishment of the mesenchymal CRC is not observed after *SOX11* knockdown (Fig. 7c, Fig. S7a, Supplementary Table 2), suggesting knockdown of *SOX11* on its own is not sufficient to induce a lineage switch.

To investigate further the functional connection between *SOX11* and the adrenergic CRC, we compared the binding sites of the major adrenergic CRC members to our own *SOX11* multi-omics data. Co-binding of *MYCN*, *ASCL1* and *TWIST1* at *SOX11* bound enhancers and promoters can be observed as well as *HAND2*, *GATA3* and *PHOX2B* co-binding at *SOX11* bound enhancers (Fig. 7d). More specifically, we observe strong co-binding of *MYCN*, *ASCL1* and *TWIST1* at transcription start sites of early and late regulated *SOX11* targets (Fig. S7b). Having established that *SOX11* is a member of the adrenergic CRC, we next looked into the dynamic regulation of *SOX11* and other CRC members from RNA-sequencing data obtained in a novel human pluripotent stem cell based differentiation model for developing human sympathoblasts (Van Haver et al., in preparation). *SOX11* was found to be expressed in earlier developmental stages prior to emergence of the adrenergic master regulator *PHOX2B* and the other CRC members including *HAND2* and *GATA3* (Fig. 7e, Fig. S7c).

Taken together, our findings support the notion that *SOX11* is a canonical CRC member and plays a distinct role, during early sympathoblast development prior to emergence of the adrenergic master regulator *PHOX2B* and the other CRC members including *HAND2* and *GATA3*. In conclusion, we postulate that *SOX11* mediates chromatin accessibility by modulating the expression of chromatin remodeling complexes, allowing the establishment and co-binding of the adrenergic core regulatory circuitry.

Adjusted Figure 7:

d
Adjusted legend Figure 7: A. Binding of *SOX11*, *HAND2*, *PHOX2B*, *GATA3*, *MYCN*, *ASCL1* and *TWIST1* at the *SOX11* locus and downstream enhancer landscape. Signal represents log likelihood ratio for the ChIP and CUT&RUN signal compared to input signal (RPM normalised). Super-enhancers of CLB-GA are annotated using ROSE (red bar) showing the *SOX11* consensus SE. **B.** Binding of *SOX11* to the *PHOX2B*, *HAND2*, *GATA3*, *ASCL1*, *TCF3* and *TWIST1* locus. Signal represents log likelihood ratio for the ChIP and CUT&RUN signal compared to input signal (RPM normalised). **C.** Tscore representing differential expression of *MES* and *ADRN* core regulatory circuitry members in RNA-sequencing data after *SOX11* knockdown in IMR-32, CLB-GA and NGP. Significant adjusted *p*-values are indicated with coloured bars. **D.** Heatmap profiles -2 kb and $+2$ kb around the summit of *SOX11* CUT&RUN peaks in IMR-32, grouped for promoters or enhancers (homer annotation). On these regions *MYCN*, *HAND2*, *GATA3*, *PHOX2B*, *ASCL1* and *TWIST1* ChIP data is mapped and ranked according to the sums of the peak scores across all datasets in the heatmap.

Adjusted Supplemental Figure 7:

a

Adjusted legend Supplemental Figure 7: A. Log₂ mRNA expression of adrenergic and mesenchymal CRC members upon *SOX11* knockdown in IMR-32, CLB-GA and NGP as well as *SOX11* overexpression after 9h and 48h represented in a heatmap. Heatmap color reflects row-wise z-score. Genes that are underlined are differentially expressed. **B.** Heatmap profiles -2 kb and $+2$ kb around the transcription start site of early and late *SOX11* targets in SH-EP, subdivided in upregulated and downregulated genes. On these regions MYCN, HAND2, GATA3, PHOX2B, ASCL1 and TWIST1 ChIP data is mapped and ranked according to the sums of the peak scores across all datasets in the heatmap.

Reviewer 1, Minor Question 8: Was the RNAseq analysis performed using external RNA controls? Given the likely interaction with MYCN, and the effect of MYCN on transcriptional amplification (Nie et al. Cell 2012; Lin et al, Cell 2012), this is necessary for interpretation of the results. The authors should comment on this.

8. Authors' response: For the RNA-seq analysis, we did not use external RNA controls. We understand the reviewers concern given the *SOX11*-MYCN association and the effect of MYCN on transcriptional amplification. However, based on the arguments below we consider the effect of MYCN on transcriptional amplification to be minor.

First, while *MYCN* is differentially downregulated in IMR-32, CLB-GA and NGP after *SOX11* knockdown, *SOX11* overexpression in SH-EP does not activate expression of *MYCN* in the *MYCN*-nonexpressing NB cell line SH-EP (Rebuttal figure 6a). Transcriptional changes observed in SH-EP after *SOX11* overexpression are thus considered independent of MYCN. Given the large overlap between transcriptional changes after *SOX11* knockdown and *SOX11* overexpression, we also consider transcriptional changes after *SOX11* knockdown to be largely independent of MYCN-driven transcriptional amplification. On a further note, *MYC* is not differentially expressed in any of the RNA-sequencing datasets (not shown).

Second, if *SOX11* overexpression resulted in global transcriptional activation, we would expect to see differential upregulated gene expression of the majority of active chromosome regions. However, we see both upregulation and downregulation of active genomic regions in IMR-32, as indicated by the presents of H3K27ac histone marks, with the majority of active regions being unaffected (Rebuttal Fig. 6b). Same is true after *SOX11* knockdown, where the majority of active regions are unaffected, so no global reduction in transcriptional expression is observed (Rebuttal Fig. 6b).

Rebuttal figure 6: A. MYCN log₂ mRNA expression levels upon SOX11 knockdown in IMR-32, CLB-GA and NGP cells and upon SOX11 overexpression in SH-EP cells for 48h. Error bars and dots represent respectively the standard deviation and mean of the four biological replicates. **B.** Overlap of H3K27ac peaks in untreated IMR-32 with differentially upregulated and downregulated genes upon SOX11 knockdown in IMR-32, CLB-GA and NGP cells and upon SOX11 overexpression in SH-EP cells for 48h.

Reviewer #2 (Remarks to the Author):

Author's response: We would like to thank Reviewer #2 for careful assessment of our manuscript and drawing our attention to a number of important questions. We have addressed each of the points raised and provide extra data and added this to the manuscript (highlighted in blue in the manuscript). The new information and clarifications have significantly improved the manuscript. A detailed point-by-point response is also included below indicating which changes were made in the manuscript (highlighted in blue and underlined), as well as all necessary modifications to the figures or new figures.

Questions:

Reviewer 2, Question 1: Which one is the pioneering factor SOX11 or MYCN or is the combination required? Given the correlation between SOX11 gain and MYCN amplification (Fig. S1a), it would seem the latter but that should be validated.

1. Author's response: The activity of MYCN on genome wide chromatin and transcriptional landscape has been extensively studied in NB. At enhanced levels, MYCN was found to associate with E-box binding motifs in an affinity-dependent manner, binding to strong canonical E-boxes at promoters and invading abundant weaker non-canonical E-boxes clustered at enhancers. While MYCN appears to increasingly invade promotor and enhancer regions, to the best of our knowledge so far no data have indicated a pioneering activity for MYCN. Work from the Zaret lab has shown that during cellular reprogramming, pioneering activity is exerted by Oct4, Sox2 and Klf4 while Myc (a close family member of Mycn) cannot bind nucleosomes on its own, but associates with these factors to target degenerate E-boxes on nucleosomes¹⁶. In contrast, the Cramer team recently executed a series of elegant experiments which strongly support a role for SOX11 as pioneering factor¹⁷. However, further

investigation (beyond the scope of this paper) is required to elucidate the role of SOX11 as pioneering factor in developing sympathoblasts and NB cells.

Of note, pioneering factors can recruit remodeling factors to open chromatin upon nucleosome binding. To unravel direct interaction partners of SOX11, we performed IP-MS in two SOX11 expressing NB cell lines, *MYCN* amplified NGP cells and *MYCN* non-amplified CLB-GA cells (see major question 7 of reviewer 1). While no members of the four major remodeling complexes are present, we observed the presence of WDHD1 (alias AND-1) (acidic nucleoplasmic DNA-binding protein, a high mobility group domain-containing protein) as commonly bound protein. Recent work showed that WDHD1/AND-1 has remarkable capability to regulate the stability of GCN5 (KAT2A) protein, the first histone H3 acetylation MYC transcriptional co-activation factor that was identified¹⁸. Histone acetyltransferases (HATs) have a central role in the modification of chromatin and are implicated in cancers. WDHD1/AND-1 forms a complex with both histone H3 and GCN5 (KAT2A). Downregulation of WDHD1/AND-1 results in GCN5 (KAT2A) degradation, leading to the reduction of H3K9 and H3K56 acetylation. WDHD1/AND-1 overexpression stabilizes GCN5 (KAT2A) through protein-protein interactions *in vivo*. Furthermore, WDHD1/AND-1 expression is increased in cancer cells in a manner correlating with increased GCN5 (KAT2A) and H3K9Ac and H3K56Ac, the latter which is known to be a genome-wide activator of transcription¹⁹. We speculate that along with its proposed pioneering activities, SOX11 also recruits WHDH1/AND-1 which stabilizes GCN5 (KAT2A) and as such facilitates MYCN driven transcriptional activity at the MYCN invaded enhancers. We propose not to include the IP-MS data and planned to explore the putative recruitment of WDHD1 through SOX11 to enhance MYCN stability in a separate follow up paper.

Reviewer 2, Question 2: *As potential super-enhancers were only detected in the minority of NB cell lines (Fig. 2 and S2a), what is primary phenotype of the majority?*

2. Author's response:

We acknowledge the concern of this reviewer (and reviewer 1) in relation to the super-enhancer (SE) calling and data representation, with overall only a subset of adrenergic NB cell lines with a SE being called. To address this in more depth we looked into two important data sets which became available after submission of our manuscript and which, in our view, provides strong support for proposing *SOX11* as a SE-marked core regulatory circuitry transcription factor in adrenergic NB. For further explanation on this, see the answer on major question 2 of reviewer 1.

Reviewer 2, Question 3: *Statistical analysis is required for the data in Fig. 2d, otherwise it is not possible to distinguish the relative differences between the CLB-GA and SH-EP cells. If there are statistically notable differences, is it only with CLB-GA or it common to other NB cell lines (aka, is CLB-GA an anomaly)? Much of the presented functional work is accomplished in IMR-32 cells, is this why?*

3. Author's response: We acknowledge this concern of the reviewer. We have one biological replicate for each cell line, which limits a quantitative and statistical analysis. However, to define objective differences between the adrenergic and mesenchymal cell line, we used the tool peakC from the group of Wouter de Laat²⁰, which enables non-parametric peak calling for one-versus-all 'C' methods. We selected the 'single analysis' option to detect peaks when there are no replicates available and used a stringent parameter qWr=2 which filters the results based on effect size and qWd=5 defining the absolute difference between the peak and the background.

Peaks were called in the adrenergic CLB-GA cell line at downstream enhancers which was not the case in the mesenchymal SH-EP cells, further strengthening our statement. Additionally, a strong peak is visible in CLB-GA for the consensus *SOX11* super-enhancers, which is absent in SH-EP. But due to our stringent peak-calling parameters this peak is not called by PeakC.

We added the following text to the Material & Methods section:

Interaction peaks were called on the raw fragment count data using the peakC R package. Settings and parameters were: “single analysis”, window=2e6, vp.pos= 5834393 (SOX11), minDist=15e3, wSize=21, qWd=2, qWr=2.

We adapted Figure 2 accordingly with the called interaction peaks underneath the 4C-seq tracks.

Adjusted Figure 2e:

Adjusted legend Figure 2E: 4C-seq analysis of the promoter site and (super-)enhancer region downstream of *SOX11* (chr2, 5.6-7.1Mb, hg19) in the NB cell lines CLB-GA (blue) and SH-EP (orange) with inclusion of published and unpublished ChIP tracks for H3K27ac, H3K4me1, PHOX2B, HAND2 and GATA3 in CLB-GA, and ATAC and H3K4me3 in SH-EP. Signal represents log likelihood ratio for the ChIP signal as compared to the input signal (RPM normalised). The viewpoint is located at the *SOX11* transcription start site and *SILC1* TSS site (cut out 100 kb). Interaction peaks called by PeakC are shown underneath 4C-seq data, CLB-GA (blue) and SH-EP (orange). Super-enhancers of CLB-GA are annotated using ROSE (orange bar).

Of further note, the Thiele team³ performed HiC analysis in the adrenergic neuroblastoma KCNR cell line and found interaction between the *SOX11* promoter and two lost super-enhancers upon ATRA treatment. For one of these super-enhancers we also find interaction with the *SOX11* promoter in the neuroblastoma CLB-GA cell line (rebuttal figure 7). Importantly, based on the Hi-C contact-map, other contact points support our defined peak interactions in the CLB-GA cell line. The similarity between the contact points in the Hi-C contact-map and our 4C-seq data strongly support the validity and specificity for adrenergic cells, thus indicating that the CLB-GA cells are most likely representative for other adrenergic NB cell lines as well rather than possibly being "an anomaly" as mentioned by this reviewer.

Fig. 5A Banerjee et al., 2020, BioRxiv

Rebuttal Figure 7: Mapping of our 4C-seq data onto the data from Banerjee et al. showing similar contact points downstream of *SOX11* for both datasets.

In line with the above, we added the text here below to the manuscript:

To provide further physical evidence for looping and contact of the cell type-specific enhancers with the promoter of *SOX11* and *SILC1*, we performed 4C-seq analysis for the *SOX11* and *SILC1* locus in CLB-GA (adrenergic MNOA cell line with multiple *SOX11* downstream enhancers) and SH-EP (mesenchymal) NB cell lines and observed looping in this highly active region between the consensus super-enhancer and other downstream enhancer loci, with the *SOX11* and *SILC1* promoter in the adrenergic cell line CLB-GA while this interaction was not detectable in the mesenchymal cell line SH-EP (Fig. 2e, Fig. S2d). In support of our findings, interaction of the consensus super-enhancer with the *SOX11* promoter in KCNR NB cells was found by Banerjee et al. using HiC analyses. Moreover, targeting of this super-enhancer using CRISPR interference caused attenuated *SOX11* expression.

Reviewer 2, Question 4: *Oddly, they switch back to CLB-GA cells for the chromatin accessibility test (Fig. 6) without justification. Are the CLB-GA and IMR-32 cells reacting differently to changes in SOX11? The lack of chromatin accessibility data for IMR-32 is particularly disappointing since inclusion would have allowed a very useful comparison of the ChIP determined SOX11 localization and SOX11-dependent changes in gene expression.*

4. Author's response: We understand the concern of the reviewer and did further efforts to confirm our initial data in CLB-GA cells and to expand further in other cell lines after *SOX11* knockdown. However, we noticed that in the context of these experiments it was difficult to achieve unequivocal interpretation of the ATAC-seq data obtained, although our initial and additional RNA-sequencing data after knockdown in different cell lines (added in this revised version) yielded very robust sets of differentially expressed genes. We looked further into the bioinformatic background for the apparent lack of concordance between RNA-sequencing and ATAC-seq data after *SOX11* knockdown. The answer can most likely be found in the particular challenges for differential analysis of ATAC-sequencing data. Currently, no widely used methods have been developed for ATAC-seq data analysis. Methods developed for differentially ATAC analysis such as HOMER, DBChIP and DiffBind rely on packages developed for differential RNA-sequencing like edgeR and DESeq2. However, when performing RNA-sequencing, each RNA transcript can have thousands of copies per cell, while ATAC-seq signals of a given genomic region can be obtained from only two allelic DNA copies. Hence much higher sensitivity is required to map subtle changes in differential chromatin accessibility, especially for lower-signal open chromatin region, which represent a large number of distal regulatory elements, such as enhancers and insulators^{21,22}. Indeed, while we do get a significant knockdown of *SOX11* when treating IMR-32, CLB-GA and NGP with *siSOX11*, the log fold change (logFC) of *SOX11* expression levels is only in the range of -1.27 to -1.85 (rebuttal figure 8).

While in contrast to the modest fold changes observed for *SOX11* upon knockdown, we noted that in the SH-EP *SOX11* inducible cell line, strong logFC of *SOX11* expression (7.88) were observed, more than 5 times greater compared to the logFC after *SOX11* knockdown (Rebuttal Figure 8).

Rebuttal Figure 8: *Log2 SOX11 mRNA expression levels upon siSOX11 treatment in IMR-32, CLB-GA and NGP and SOX11 overexpression for 48h in SH-EP cells. Error bars and gray point represent respectively the 95% confidence interval and mean of the four biological replicates. NOTE: for RNA-sequencing after SOX11 overexpression for 9h, the quantSeq 3' mRNA-sequencing method was used. Given that the SOX11 overexpression construct only contains the coding DNA sequence and not the 3'-UTR, SOX11 expression is not picked up using 3' sequencing.*

We then hypothesized that SH-EP *SOX11* overexpression would perform more robustly as a cellular model to assess the effects of *SOX11* activity on chromatin accessibility in NB cells. Subsequent analysis of our ATAC-seq on the *SOX11* inducible SH-EP cells yielded reproducible ATAC profiles in multiple biological replicates (different biological clones), solidifying our confidence in these data. Using this data set, we now show that *SOX11* predominantly affects chromatin accessibility of enhancers regions and overlap of *SOX11* binding sites in SHEP after *SOX11* overexpression with differential ATAC peaks indicates a direct role of *SOX11* in chromatin accessibility (Fig. 6a-b). Motif enrichment predicted a role for the *SOX11* direct target and SWI/SNF component SMARCC1, both in

open and closed regions after *SOX11* overexpression. Further, motif enrichment is also noted for a TEAD motif in open regions, which is in line with TEAD2 previously described as a direct regulated target of *SOX11* in neuronal development²³ and enrichment of a C/EBP motif in closed regions, with C/EBPB and C/EBPD being master regulators of the mesenchymal subtype in glioblastoma^{24,25} (Fig. 6c, S6a-b).

Interestingly, comparing our differential ATAC data to *SOX11*-dependent changes in gene expression did not reveal strong correlation between differential ATAC signal and gene expression changes (data not shown). Of note in this context, Kiani et al. previously showed that for single-factor perturbations, changes in chromatin accessibility are not as concordant with transcriptional changes as compared to for example similar data for differentiation trajectories that are the results of multifactorial biological changes²⁶. Additionally, time- and locus-dependent differences in changes in gene expression and chromatin accessibility can make the correlation of both parameters challenging. Of further note, downregulated genes are not completely shut down, the majority (91%) still having a log₂ mRNA expression above three after *SOX11* knockdown. Similarly, 77% of upregulated genes already have a log₂ mRNA expression above three in untreated SH-EP cells, suggesting upregulation of already active genes rather than induction of previously inactive genes. Taken together, given the fact that *SOX11* mostly affects chromatin accessibility of enhancer regions rather than transcription start sites (Fig. 6b), we hypothesize *SOX11* modulates the expression of its directly regulated targets through additional mechanisms (*e.g.* co-factor recruitment) rather than altered chromatin accessibility alone. All of the above further explains the lack of concordance between our observed chromatin accessibility and gene expression changes.

Finally, we looked into overlap between differential ATAC-seq peaks and known mesenchymal and adrenergic super-enhancers⁷, but observe no strong changes in chromatin accessibility at said super-enhancer regions (data not shown) nor do we see differential chromatin accessibility in common regions bound by the adrenergic CRC (Fig. S6c), suggesting *SOX11* overexpression in itself is not sufficient to induce a transition in cell lineage, at least not at 48 hours after induction of *SOX11*. In conclusion, we postulate that forced *SOX11* leads to global changes in chromatin accessibility, specifically of enhancer regions, but additional *SOX11* interactors are needed to drive epigenetic plasticity towards full mesenchymal-adrenergic transitions.

We remove our current ATAC-sequencing data from the manuscript and removed Figure 6a-b and S6a-d as well as associated text in the manuscript and exchanged with our new ATAC-sequencing data in SH-EP after *SOX11* overexpression for 48h as indicated below:

Forced *SOX11* overexpression in SH-EP NB cells impacts genome wide chromatin accessibility

Given that (1) the *SOX11* regulome when overexpressed in SH-EP NB cells largely recapitulates its endogenous transcriptional activity in adrenergic NB cells, (2) *SOX11* regulates a broad epigenetic machinery including SWI/SNF and that (3) forced *SOX11* overexpression in SH-EP cells attenuated the mesenchymal gene signature, we selected this model to study the impact of *SOX11* on genome wide chromatin accessibility. Differential ATAC-seq 48h after *SOX11* overexpression in SH-EP identified 1847 regions with altered chromatin accessibility, *i.e.* closed (n=871) and opened (n=976) (Fig. 6a). Differential ATAC sites were predominantly located at enhancer regions (Fig. 6b) and overlap of *SOX11* binding sites in SH-EP after *SOX11* overexpression with differential ATAC peaks indicates a direct role of *SOX11* in chromatin accessibility. We furthermore see high enrichment in the differential ATAC peaks for a TGA(G/C)TCA motif known to be bound by several transcription factors of the bZIP family including JUN, ATF3, FOSL1, FOSL2 and SWI/SNF component SMARCC1 (Fig. 6c). As mentioned previously, SMARCC1 is a direct *SOX11* target with induced expression upon *SOX11* overexpression and therefore a possible *SOX11* controlled mediator of chromatin accessibility at these sites. In addition, TEAD motifs are highly enriched in the open chromatin regions, which is in line with TEAD2 previously described as a direct regulated target of *SOX11* in neuronal development (Fig. 6c). Additionally, Rajbhandari et al. proposed TEAD4 as an important positive regulator of MYCN and prognostic marker in high-risk NB. In closed chromatin regions upon *SOX11* overexpression, we identified high enrichment for a C/EBP motif (Fig. 6c). Of note, C/EBPB and C/EBPD are master

regulators of the mesenchymal subtype in glioblastoma and C/EBP β is downregulated upon *SOX11* overexpression (adj.pval 6.5e-7, logFC -0.93). Additionally, we found C/EBP motif enrichment in the downregulated genes upon *SOX11* overexpression (Fig. S6c), indicating the downregulation of mesenchymal markers in the SH-EP cells upon *SOX11* overexpression. However, while we do observe positive and negative enrichment of adrenergic and mesenchymal gene signatures upon *SOX11* overexpression respectively (Fig. S4h) we did not observe strong changes in chromatin accessibility at mesenchymal nor adrenergic super-enhancers (data not shown), nor do we see differential chromatin accessibility in common regions bound by the adrenergic master transcription factors (Fig. S6c), suggesting *SOX11* overexpression in itself is not sufficient to induce a full transition of cell lineage, at least not at 48 hours after induction of *SOX11*. Indeed, the core regulatory circuitry members in adrenergic NB are not upregulated after *SOX11* overexpression for 48h (Supplementary Table 2). In conclusion, we postulate that forced *SOX11* overexpression leads to global changes in chromatin accessibility, specifically of enhancer regions, and we assume that additional *SOX11* co-drivers are needed to drive epigenetic plasticity towards full mesenchymal-adrenergic transitions.

Adjusted Figure 6:

Adjusted legend Figure 6: **A.** Heatmap profiles -2 kb and $+2$ kb around the summit of downregulated and upregulated differential ATAC-seq peaks upon *SOX11* overexpression for 48h in SH-EP (3 biological replicates, UT = untreated, DOX = doxycycline mediated induction of *SOX11*). Density profiles are shown representing the average ATAC-seq signal at the presented regions for upregulated regions (green) and downregulated regions (blue). **B.** Genome-wide peak annotation distribution (%) (Homer annotation) for the downregulated and upregulated differential ATAC-seq peaks upon *SOX11* overexpression for 48h in SH-EP. **C.** Motif enrichment for open and closed regions determined by ATAC-seq upon *SOX11* overexpression for 48h in SH-EP cells.

Adjusted Supplemental Figure 6:

Adjusted legend Supplemental Figure 6: **A.** *ATF3*, *FOSL1*, *FOSL2*, *SMARCC1*, *JUN* and *JUNB* mRNA expression (\log_2) upon *SOX11* overexpression for 48h in SH-EP cells. Error bars and diamond shape represent respectively the s.d. and mean of the three biological replicates. Limma Voom for statistical testing. **B.** Top ranked and significant ($FDR < 0.25$) gene set of CEBPB motifs in the downregulated genes upon *SOX11* overexpression in SH-EP. **C.** Heatmap profiles -2 kb and $+2$ kb around the summit of common binding sites of *HAND2*, *PHOX2B* and *GATA3* in adrenergic neuroblastoma cells. Density profiles are shown representing the average ATAC-seq signal upon *SOX11* overexpression for 48h in SH-EP (3 biological replicates, UT = untreated, DOX = doxycycline mediated induction of *SOX11*).

Reviewer 2, Question 5: The high variability in knocking down *SOX11* yet comparable functional impact on the cells suggests there is either an indirect effect that is driving the differences or the slight reduction in *SOX11* observed in CLB-GA cells impacts the key activity (Fig. 3). The authors should identify and report what that key function is. In addition, the colony formation data in Fig. 3c needs quantification otherwise it is not possible to reach a conclusion.

5. Author's response: We agree that for the first two of four shRNAs the knock down efficiency differs between the tested cell lines. However, for shRNA #3 and #4, *SOX11* knockdown is robust in multiple replicates (Rebuttal Fig. 9). Therefore we propose to only keep data of the more robust shRNAs #3 and #4. We adjusted the western blot in Fig. 3A and Fig. S5h accordingly:

Rebuttal Figure 9: A. The western blot data used in the manuscript (Fig. 3A) with reduction of SOX11 protein levels upon SOX11 knockdown in 3 cell lines NGP, CLB-GA and IMR-32 using 2 shRNAs (sh#3 and sh#4). **B.** Reduction of SOX11 protein levels in 2 cell lines NGP and CLB-GA upon SOX11 knockdown using 2 shRNAs (sh#3 and sh#4). **C.** Reduction of SOX11 protein levels in 2 cell lines NGP and CLB-GA upon SOX11 knockdown using 2 shRNAs (sh#3 and sh#4). **D.** Reduction of SOX11 protein levels in 1 cell line NGP upon SOX11 knockdown using 2 shRNAs (sh#3 and sh#4). **E.** Reduction of SOX11 protein levels in 1 cell line NGP upon SOX11 knockdown using 2 shRNAs (sh#3 and sh#4). **F.** Reduction of SOX11 protein levels in 1 cell line IMR-32 upon SOX11 knockdown using 1 shRNAs (sh#4). Protein levels of VCL (vinculin) or β -tubulin (bTub) were used as loading control.

Adjusted Figure 3:

Adjusted legend Figure 3: A. SOX11 protein levels 6 days upon shSOX11 treatment in NGP, CLB-GA and IMR-32 cells with 2 different shRNAs and one non-targeting control (NTC). Vinculin (VCL) is used as loading control.

Adjusted Supplemental Figure 5:

Adjusted legend Supplemental Figure 5: H. SOX11, SMARCC1, c-MYB and MYCN protein levels and loading control ACTB in NGP cells upon knockdown of SOX11 using 2 different shRNAs and a non-targeting control (NTC).

In addition, we added quantification of the colony formation capacity experiments to the manuscript as requested.

Adjusted Figure 3C:

Adjusted legend Figure 3: C. Reduction in colony formation capacity for NGP, CLB-GA and SK-N-AS cells, 14 days upon siRNA SOX11 treatment (dharmafect transfection) as compared to non-targeting control (siNTC). Data were generated in triplicate for each cell line, and quantification was done using ImageJ.

Reviewer 2, Question 6: *The authors wait 48 h after overexpressing SOX11 (Fig. 4b). As this provide ample time for downstream players to impact the system, do the authors observe comparable gene expression changes after 2-4 hours?*

6. Author's response: We agree with the reviewer that the RNA-sequencing experiment 48h upon overexpression of SOX11 will not only provide the direct effects but also indirect effects. Therefore, we performed qPCR and western blot analysis on different timepoints (15 min, 30 min, 1 h, 2 h, 3h, 6h, 9 h, 12h 24 h, 48 h) upon SOX11 overexpression (Rebuttal figure 10 and Fig. S4c). As can be appreciated from the qPCR data and western blot analysis here below, SOX11 mRNA and protein level is already upregulated 15 min and 2 hours respectively upon doxycycline treatment. To evaluate direct downstream targets, we also tested mRNA levels of *c-MYB*, *CBX2*, *MEX3A* and *MEX3B* (top upregulated genes at 48 h upon SOX11 overexpression) and see an upregulation of these genes at the 9h timepoint (Fig. S4c). Therefore, we selected the 9h timepoint upon SOX11 overexpression for additional RNA-sequencing.

Rebuttal Figure 10: A. SOX11 protein levels upon SOX11 overexpression in SH-EP cells over time using different monoclonal expansions (F6, G2, C11). VCL (vinculin) is used as loading control. **B.** SOX11 protein levels of SH-EP untreated and doxycycline treated cells used for RNA sequencing. **C.** Log₂ SOX11 mRNA expression levels upon SOX11 overexpression in SH-EP cells over time. Error bars represent the standard error of three technical replicates.

Using the combination of RNA-sequencing 9h and 48h upon SOX11 upregulation, we were able to distinguish between early and late regulated SOX11 target genes. Interestingly, multiple epigenetic modifiers involved in DNA and histone methylation and chromatin remodelling, including SWI/SNF and PRC1/2 complex members were strongly enriched in the early regulated genes while in late SOX11 regulated genes we observed stronger enrichment of genes involved in axon outgrowth, neural crest cell migration and cytoskeleton. Using overlap with differential gene expression upon SOX11 knockdown in IMR-32, CLB-GA and NGP as well as correlation with SOX11 expression in two independent NB cohorts, we established an early and late SOX11 gene signature (Fig.4c-d). Of further interest, we observe an enrichment of the adrenergic gene sets in the early SOX11 targets possibly suggesting a role of SOX11 in maintenance of cell identity (Fig. S4f).

The integration of the additional transcriptome data after SOX11 knockdown in CLB-GA and NGP NB cells and transcriptome profiling for doxycycline-induced SOX11 expression at an earlier time point (9h after induction) in mesenchymal SH-EP NB cells is presented in chapter “The SOX11 regulated transcriptome is involved in epigenetic control, cytoskeleton and neurodevelopment” and Figure 4 and S4. In line with the new findings on the early SOX11 regulated genes, the manuscript was adjusted as shown below:

To further validate these findings and to filter out transcriptional bystanders, we performed an orthogonal experiment using SOX11 inducible SH-EP cells, which under control conditions do not express SOX11, and obtained transcriptome data at 9h (SOX11 early regulated genes) and 48h (SOX11 late regulated genes) after SOX11 induction (Fig. 4a, Fig. S4a and c, Supplementary Table 2). (...)

Most notably, we already observed an enrichment of the adrenergic NB gene sets in the early SOX11 regulated genes suggesting a possible direct role of SOX11 in maintenance of cell identity (Fig. S4g). (...)

In order to better understand how SOX11 controls these phenotypes through its early regulated genes, we looked into top enriched gene sets in the early regulated genes which included “SWI/SNF complex” and “chromatin remodeling”. This is supported by differential early regulation of several SWI/SNF components such as BAF core component SMARCC1/BAF155 and pBAF specific PBRM1/BAF180. Further, other epigenetic regulators were found including the histone deacetylase HDAC2, the PHF6 NurD component, the H3K27me3 reader and canonical PRC1 complex component CBX2, the chromatin-modifying enzyme lysine-specific demethylase 1 KDM1A/LSD1 and pioneer transcription factor c-MYB, which were all also identified as direct SOX11 targets (see further) (Fig. 4c-d, S4e and g, Fig. 5e). (...)

Adjusted Figure 4:

Adjusted legend Figure 4: A. Overlap of genes perturbed in SH-EP upon SOX11 overexpression for 48h and 9h (adj p value < 0.05) with differential genes upon 9h overexpression being the SOX11 early regulated genes and differential genes upon 48h but not 9h overexpression being the SOX11 late regulated genes (Fisher test p-value <math>< 2.2e-16</math>). **B.** Overlap of genes perturbed in IMR-32, CLB-GA and NGP upon SOX11 knockdown for 48h (adj p value < 0.05) with respectively the SOX11 early and late regulated genes in SH-EP (Fisher test p-value <math>< 2.2e-16</math>). **C.** SOX11 early and late genes signature obtained by the overlap of differentially expressed genes upon SOX11 knockdown in IMR-32, CLB-GA and NGP with the early and late SOX11 regulated genes in SH-EP as well as with genes correlation with SOX11 expression in 2 different NB tumor cohorts (NRC GSE85047, Kocak GSE45547, p-value < 0.05). A color next to each gene represents the involved pathways. Bold and underlined represents genes that are bound by SOX11 in IMR-32, CLB-GA, NGP and SH-EP after SOX11 overexpression for 48h. **D.** Top enriched genesets after doing GSEA analysis (<http://www.qsea-msigdb.org/qsea/index.jsp>, ontology gene sets C5) for SOX11 early and SOX11 late regulated genes. Depicted is the normalized enrichment score (NES, x-axis) and the false discovery rate (FDR, color).

Adjusted Supplemental Figure 4:

Adjusted legend Supplemental Figure 4: C. (right) SOX11 protein levels upon SOX11 overexpression in SH-EP cells over time. VCL (vinculin) is used as loading control. (right) c-MYB, CBX2, MEX3A and MEX3B relative mRNA expression levels upon SOX11 overexpression in SH-EP cells over time. Error bars represent the standard error of three technical replicates. F. Enrichment of the Proneural and Mesenchymal genesets in glioblastoma²⁷, the Adrenergic and Mesenchymal genesets in neuroblastoma⁷ and the mesenchymal vs NB stage 1²⁸ genesets in the differential genes upon SOX11 knockdown in IMR-32, CLB-GA and NGP and SOX11 overexpression for 9h and 48h in SH-EP showing normalized enrichment score (color) and false discovery rate (size).

Reviewer 2, Question 7: Does SOX11 occupy the regulatory elements controlling all the various gene loci as detected by ChIP? Relying on “predicted SOX11 binding sites” (Fig. S4b) is not sufficient. The reported low (12%) use of predicted SOX11 binding sites (Fig. 5) substantiate this concern. Importantly, any ChIP results should be validated by checking that the signal is lost/reduced when SOX11 is knocked down including for the results presented in Fig.5

7. Author’s response: Fig. S4b (now figure S4d in updated manuscript) is an enrichment plot for “predicted SOX11 binding sites” detected in the genes downregulated upon SOX11 knockdown and upregulated upon SOX11 overexpression. It validates our findings that the genes affected by SOX11 knockdown or overexpression can be direct targets (as there is a SOX motif). The predicted binding sites tested here are the genes included in the gene set “SOX11_TARGET_GENES” from msigdb (http://www.gsea-msigdb.org/gsea/msigdb/cards/SOX11_TARGET_GENES). This includes genes containing one or more binding sites for SOX11 in their promoter regions (TSS -1000, +100 bp) as identified by GTRD version 20.06 ChIP-seq harmonization. However, also indirect targets are found to be differentially expressed upon SOX11 knockdown and overexpression and indeed relying on “predicted SOX11 binding sites alone” is insufficient to map direct SOX11 targets.

For this reason, we have strengthened our transcriptome data with new SOX11 binding data using CUT&RUN in adrenergic MYCN amplified (IMR-32, NGP) and MYCN non-amplified cell lines (CLB-GA) as well as a mesenchymal SOX11-negative cell line after SOX11 overexpression (SH-EP SOX11 overexpression 48h), as is also discussed in major question 6 of reviewer 1. In our newly generated SOX11 CUT&RUN datasets we observe significant enrichment of several very similar SOX motifs in the overlapping SOX11 CUT&RUN peaks in the four different NB cell lines (Rebuttal table 1, supplemental table 5). Given that SOX proteins share an HMG domain with more than 80% sequence identity, it is not surprising that the DNA consensus motif that they recognize is also highly similar which has been defined as the heptameric sequence 5’-(A/T)(A/T)CAA(A/T)G-3’^{29,30}. This makes it likely that all these different SOX motifs are actually motifs recognized by SOX11, increasing the total number of targets showing a SOX11 specific binding motif. Of further interest, all other top enriched motifs are also highly similar (Rebuttal table 2, supplemental table 5), showing a consensus

GGA(A/T) motif recognized by the members of the ETS transcription factor family³¹. It is however not uncommon in CHIP-seq datasets, that the top enriched motifs do not resemble the binding motif of the CHIP'ed TFs. This can be due to the high presence of non-target specific motifs repeatedly found across CHIP-seq datasets known as 'zinger' motifs including CTCF-like, JUN-like, ETS-like and THAP11-like motifs. While the biochemical mechanism behind zinger-associated regions is not fully understood yet, Hunt and Wasserman show that zinger motifs are often in proximity of cohesin bound regions³². Given that SOX11 is binding at active transcriptional regulatory regions (Fig. S5c-d) and its proposed role as transcriptional activator, it is thus not surprising that we see overlap with cohesin bound regions, which together with CTCF plays a major role in transcription regulation³³.

Rebuttal table 1: Enrichment of SOX motifs in the homer motif enrichment (known motifs) 200 bp size around peak summit for overlapping SOX11 C&R peaks in IMR-32, CLB-GA, NGP and SH-EP cells after SOX11 overexpression for 48h.

Motif Name	Consensus	P-value	% of Target
Sox4(HMG)/proB-Sox4-ChIP-Seq(GSE50066)/Homer		1,00E-08	8.06%
Sox10(HMG)/SciaticNerve-Sox3-ChIP-Seq(GSE35132)/Homer		1,00E-07	13.83%
Sox3(HMG)/NPC-Sox3-ChIP-Seq(GSE33059)/Homer		1,00E-05	14.18%
Sox2(HMG)/mES-Sox2-ChIP-Seq(GSE11431)/Homer		1,00E-05	7.68%
Sox6(HMG)/Myotubes-Sox6-ChIP-Seq(GSE32627)/Homer		1,00E-04	12.44%
Sox17(HMG)/Endoderm-Sox17-ChIP-Seq(GSE61475)/Homer		1,00E-02	5.43%
Sox15(HMG)/CPA-Sox15-ChIP-Seq(GSE62909)/Homer		1,00E-02	6.22%
Sox9(HMG)/Limb-SOX9-ChIP-Seq(GSE73225)/Homer		1,00E-02	8.95%

Rebuttal table 2: Enrichment of ETS motifs in the homer motif enrichment (known motifs) 200 bp size around peak summit for overlapping SOX11 C&R peaks in IMR-32, CLB-GA, NGP and SH-EP cells after SOX11 overexpression for 48h.

Motif Name	Consensus	P-value	% of Target
Elk4(ETS)/Hela-Elk4-ChIP-Seq(GSE31477)/Homer		1,00E-19	14.13%
Elk1(ETS)/Hela-Elk1-ChIP-Seq(GSE31477)/Homer		1,00E-15	13.65%
Fli1(ETS)/CD8-FLI-ChIP-Seq(GSE20898)/Homer		1,00E-15	18.40%
ELF1(ETS)/Jurkat-ELF1-ChIP-Seq(SRA014231)/Homer		1,00E-13	12.13%
ETV4(ETS)/HepG2-ETV4-ChIP-Seq(ENCODE)/Homer		1,00E-13	19.03%
ETS(ETS)/Promoter/Homer		1,00E-09	7.51%

Additionally, the robust overlap between the SOX11 binding sites in the different cell lines (Fig. 5a, S5a) as well as enrichment of SOX11 binding at the transcription start site of differentially regulated targets (Fig. 5b), make us strongly confident in the validity of the mapped direct SOX11 targets. Furthermore, one has to note that our SOX11 ChIP-sequencing experiments in IMR-32 were performed using an in-house developed SOX11 antibody while SOX11 CUT&RUN was performed using a commercial SOX11 antibody (anti-SOX11, HPA000536). Overlap between the different datasets using two different antibodies and two different techniques, strongly validates the legitimacy of our called SOX11 binding sites (Fig. S5a). Of further note, the antibody used for CUT&RUN (HPA000536) has the label of being a Prestige Antibodies® (powered by Atlas Antibodies), which have undergone stringent validation and characterization using IHC, WB, ICC-IF, or RNA sequencing and compared to bioinformatic information and literature and are characterized via tissue microarrays in over 40 different normal human tissues, over 50 cell lines, and over 20 cancer types. Finally, strong overlap of SOX11 binding sites with RNA-sequencing targets in the same cell lines further validates our CUT&RUN data. Hence, all of the above further makes us strongly confident in the specificity of our SOX11 antibody.

Reviewer 2, Question 8: *The authors make an intriguing suggestion that the potential SOX11-dependent increased expression of CDKN1a (Fig. S4c) is the causative factor for the cell cycle arrest. Does knocking down CDKN1a in these cells restore the cell cycle?*

8. Author's response: As explained for major question 5 of reviewer 1 and question 6 of this reviewer, we performed additional RNA-seq analysis upon SOX11 knockdown in CLB-GA and NGP cells and following 9h SOX11 induction in SH-EP cells, in order to achieve better discrimination between direct targets and transcriptional bystanders. Re-analysis of all data now reveals that CDKN1A (in contrast to the data in IMR-32) is not a major regulated target in CLB-GA (only in the top 40% of upregulated genes and only 0.34 fold change) and is not regulated in NGP cells (rebuttal figure 11).

Rebuttal Figure 11: Log2 CDKN1A mRNA expression levels upon siSOX11 treatment in IMR-32 (left), CLB-GA (middle) and NGP (right) cells. Error bars and gray point represent respectively the 95% confidence interval and mean of the four biological replicates. Statistical analysis with moderated t-test of Limma voom (siSOX11 vs siNTC in IMR-32: $p=6.75e-07$, siSOX11 vs siNTC in CLB-GA: $p=5.10e-06$, siSOX11 vs siNTC in NGP: $p=0.113$).

To further verify these findings, we performed protein blotting for p21 which showed no significant changes in p21 levels 48h and 72h after siRNA SOX11 treatment in CLB-GA and IMR-32 cells (Rebuttal Fig. 12), neither 7 days following shRNA SOX11 transduction in NGP, IMR-32 and CLB-GA cells (Rebuttal Fig. 13-15), except for one shRNA in CLB-GA cells. We conclude that p21 is not a major regulated SOX11 target and decided to take out these data.

To avoid any misinterpretation in the manuscript, we removed the sentence underneath from the manuscript, as well as CDKN1A levels in Fig. S4C:

“In line with the observed cell cycle arrest upon SOX11 knockdown, the CDK inhibitor CDKN1A (also known as p21) ($p=6.8e-07$) was one of the top upregulated genes (Fig. S4c).”

Rebuttal Figure 12: SOX11, p21 and protein loading control (vinculin) protein levels upon siSOX11 treatment in CLB-GA and IMR-32 cells visualized on WB (left) and quantified using ImageJ (right). No induction of p21 protein levels 48h and 72h upon siSOX11 treatment.

Rebuttal Figure 13: SOX11, p21 and protein loading control (vinculin) protein levels 7 days upon shSOX11 treatment in NGP cells visualized on WB (left) and quantified using ImageJ (right). Here, a small reduction of p21 could be observed for 2 shRNA (sh#4 and sh#2).

Rebuttal Figure 14: SOX11, p21 and protein loading control (vinculin) protein levels 7 days upon shSOX11 treatment in IMR-32 cells visualized on WB (left) and quantified using ImageJ (right). No induction of p21 protein levels could be observed.

Rebuttal Figure 15: SOX11, p21 and protein loading control (vinculin) protein levels 7 days upon shSOX11 treatment in CLB-GA cells visualized on WB (left) and quantified using ImageJ (right). A small induction of p21 could be observed for one shRNA (sh#4).

Reviewer 2, Question 9: The authors point out that SOX11 repressed target genes fall within the same gene categories as those previously reported to be controlled by MYCN, but are these the same genes as would be expected if SOX11 and MYCN are working together, or do the loci just happen to fall with common gene categories?

9. Author's response: In our initial RNA-sequencing dataset in IMR-32 after SOX11 knockdown, we indeed saw indications of enhanced ribosome biogenesis, a phenotype that as previously also been reported in MYCN driven NB³⁴. However, we did not see enrichment of translation and ribosome hyperactivity in CLB-GA and NGP cells (Fig. 4e) suggesting this was caused through transcriptional bystander effects or through context specific features of IMR-32 cells, as was also discussed in the answer of major question 5 of reviewer 1. Hence, we don't expect these differentially regulated genes to be the results of collaboration of SOX11 and MYCN.

In light of these new finding, we removed the following information from the manuscript: Finally, SOX11 repressed targets are enriched for gene sets involved in translation initiation, ribosome hyperactivity and mRNA processing, which is indicative of an enhanced ribosome biogenesis response as previously reported in MYCN driven NB(Fig. 4c).

Reviewer 2, Question 10: The authors state that SOX11 DNA motifs correlate with MYCN and MAX sites but do not describe whether the correlation varies with impact on gene expression—is there a correlation of MYCN or MAX motifs with SOX11 repressed or activated genes?

10. Author's response: Given the observed co-binding between MYCN and SOX11 (Fig. 7d), we agree with the reviewer that it would be of interest to check correlation with differential SOX11 regulated targets. To this end, we mapped MYCN ChIP-seq data on the transcription start sites of the early and late SOX11 regulated targets and indeed see correlation both with repressed and activated genes. The same is true for ASCL1 and TWIST1 binding (Fig. S7b, Fisher test p-value < 2.2e-16). Given these new insights, we re-evaluated the relationship between SOX11 and the adrenergic CRC, as is discussed in

the answer to minor question 7 of reviewer 1 as well as the chapter “SOX11 is a core regulatory circuitry master transcription factor in adrenergic NB” in the manuscript and figure 7 and S7.

In the context of the correlation between MYCN binding and SOX11 repressed or activated genes we have changed the manuscript and figures as indicated below:

To investigate further the functional connection between SOX11 and the adrenergic CRC, we compared the binding sites of the major adrenergic CRC members to our own SOX11 multi-omics data. Co-binding of MYCN, ASCL1 and TWIST1 at SOX11 bound enhancers and promoters can be observed as well as HAND2, and to weaker extend GATA3 and HAND2, co-binding at SOX11 bound enhancers (Fig. 7d). More specifically, we observe strong co-binding of MYCN, ASCL1 and TWIST1 at transcription start sites of early and late regulated SOX11 targets (Fig. S7b).

Adjusted Figure 7d:

Adjusted legend Figure 7: D. Heatmap profiles -2 kb and $+2$ kb around the summit of SOX11 CUT&RUN peaks in IMR-32, grouped for promoters or enhancers (homer annotation). On these regions MYCN, HAND2, GATA3, PHOX2B, ASCL1 and TWIST1 ChIP data is mapped and ranked according to the sums of the peak scores across all datasets in the heatmap.

Adjusted Supplemental Figure 7b:

Adjusted legend Supplemental Figure 7: B. Heatmap profiles -2 kb and $+2$ kb around the transcription start site of early and late SOX11 targets in SH-EP, subdivided in upregulated and downregulated genes. On these regions MYCN, HAND2, GATA3, PHOX2B, ASCL1 and TWIST1 ChIP data is mapped and ranked according to the sums of the peak scores across all datasets in the heatmap.

Rebuttal references:

- 1 Zimmerman MW, Liu Y, He S, Durbin AD, Abraham BJ, Easton J *et al.* MYC Drives a Subset of High-Risk Pediatric Neuroblastomas and Is Activated through Mechanisms Including Enhancer Hijacking and Focal Enhancer Amplification. *Cancer Discov* 2018; **8**: 320–335.
- 2 Gartlgruber M, Sharma AK, Quintero A, Dreidax D, Jansky S, Park Y-G *et al.* Super enhancers define regulatory subtypes and cell identity in neuroblastoma. *Nat Cancer* 2021; **2**: 114–128.
- 3 Banerjee, D. *et al.* Lineage specific transcription factor waves reprogram neuroblastoma from self-renewal to differentiation. 2020.07.23.218503 (2020)
- 4 Boeva V, Louis-Brennetot C, Peltier A, Durand S, Pierre-Eugène C, Raynal V *et al.* Heterogeneity of neuroblastoma cell identity defined by transcriptional circuitries. *Nat Genet* 2017; **49**: 1408–1413.
- 5 Afanasyeva, E. A. *et al.* Kalirin-RAC controls nucleokinetic migration in ADRN-type neuroblastoma. *Life Sci. Alliance* **4**(5), e201900332 (2021).
- 6 Barbaric S, Walker J, Schmid A, Svejstrup JQ, Hörz W. Increasing the rate of chromatin remodeling and gene activation—a novel role for the histone acetyltransferase Gcn5. *EMBO J* 2001; **20**: 4944–4951.
- 7 van Groningen T, Koster J, Valentijn LJ, Zwijnenburg DA, Akogul N, Hasselt NE *et al.* Neuroblastoma is composed of two super-enhancer-associated differentiation states. *Nat Genet* 2017; **49**: 1261–1266.
- 8 Shi H, Tao T, Abraham BJ, Durbin AD, Zimmerman MW, Kadoch C *et al.* ARID1A loss in neuroblastoma promotes the adrenergic-to-mesenchymal transition by regulating enhancer-mediated gene expression. *Sci Adv* 2020; **6**: eaaz3440.
- 9 Zimmerman MW, Durbin AD, He S, Oppel F, Shi H, Tao T *et al.* Retinoic acid rewires the adrenergic core regulatory circuitry of childhood neuroblastoma. *Sci Adv* 2021; **7**: eabe0834.
- 10 Congras A, Hoareau-Aveilla C, Caillet N, Tosolini M, Villarese P, Cieslak A *et al.* ALK-transformed mature T lymphocytes restore early thymus progenitor features. *J Clin Invest* 2020; **130**: 6395–6408.
- 11 López-Valero I, Dávila D, González-Martínez J, Salvador-Tormo N, Lorente M, Saiz-Ladera C *et al.* Midkine signaling maintains the self-renewal and tumorigenic capacity of glioma initiating cells. *Theranostics* 2020; **10**: 5120–5136.
- 12 Tsurusaki Y, Koshimizu E, Ohashi H, Phadke S, Kou I, Shiina M *et al.* De novo SOX11 mutations cause Coffin–Siris syndrome. *Nat Commun* 2014; **5**: 4011.
- 13 Zarate YA, Bhoj E, Kaylor J, Li D, Tsurusaki Y, Miyake N *et al.* SMARCE1, a Rare Cause of Coffin–Siris Syndrome: Clinical Description of Three Additional Cases. *Am J Med Genet A* 2016; **170**: 1967–1973.
- 14 Alver BH, Kim KH, Lu P, Wang X, Manchester HE, Wang W *et al.* The SWI/SNF chromatin remodelling complex is required for maintenance of lineage specific enhancers. *Nat Commun* 2017; **8**: 14648.
- 15 Shendy NAM, Zimmerman MW, Abraham BJ, Durbin AD. Intrinsic transcriptional heterogeneity in neuroblastoma guides mechanistic and therapeutic insights. *Cell Rep Med* 2022; **3**: 100632.
- 16 Soufi A, Garcia MF, Jaroszewicz A, Osman N, Pellegrini M, Zaret KS. Pioneer Transcription Factors Target Partial DNA Motifs on Nucleosomes to Initiate Reprogramming. *Cell* 2015; **161**: 555–568.
- 17 Dodonova SO, Zhu F, Dienemann C, Taipale J, Cramer P. Nucleosome-bound SOX2 and SOX11 structures elucidate pioneer factor function. *Nature* 2020; **580**: 669–672.
- 18 Li Y, Jaramillo-Lambert AN, Yang Y, Williams R, Lee NH, Zhu W. And-1 is required for the stability of histone acetyltransferase Gcn5. *Oncogene* 2012; **31**: 643–652.
- 19 Topal S, Vasseur P, Radman-Livaja M, Peterson CL. Distinct transcriptional roles for Histone H3-K56 acetylation during the cell cycle in Yeast. *Nat Commun* 2019; **10**: 4372.
- 20 Geeven G, Teunissen H, de Laat W, de Wit E. peakC: a flexible, non-parametric peak calling package for 4C and Capture-C data. *Nucleic Acids Res* 2018; **46**: e91.

- 21 Gontarz P, Fu S, Xing X, Liu S, Miao B, Bazylianska V *et al.* Comparison of differential accessibility analysis strategies for ATAC-seq data. *Sci Rep* 2020; **10**: 10150.
- 22 Yan F, Powell DR, Curtis DJ, Wong NC. From reads to insight: a hitchhiker's guide to ATAC-seq data analysis. *Genome Biol* 2020; **21**: 22.
- 23 Bhattaram P, Penzo-Méndez A, Sock E, Colmenares C, Kaneko KJ, Vassilev A *et al.* Organogenesis relies on SoxC transcription factors for the survival of neural and mesenchymal progenitors. *Nat Commun* 2010; **1**: 1–12.
- 24 Cooper LAD, Gutman DA, Chisolm C, Appin C, Kong J, Rong Y *et al.* The Tumor Microenvironment Strongly Impacts Master Transcriptional Regulators and Gene Expression Class of Glioblastoma. *Am J Pathol* 2012; **180**: 2108–2119.
- 25 Carro MS, Lim WK, Alvarez MJ, Bollo RJ, Zhao X, Snyder EY *et al.* The transcriptional network for mesenchymal transformation of brain tumours. *Nature* 2010; **463**: 318–325.
26. Kiani, K., Sanford, E. M., Goyal, Y. & Raj, A. Changes in chromatin accessibility are not concordant with transcriptional changes for single-factor perturbations. 2022.02.03.478981 (2022)
- 27 Verhaak RGW, Hoadley KA, Purdom E, Wang V, Qi Y, Wilkerson MD *et al.* Integrated genomic analysis identifies clinically relevant subtypes of glioblastoma characterized by abnormalities in PDGFRA, IDH1, EGFR, and NF1. *Cancer Cell* 2010; **17**: 98–110.
- 28 Rajbhandari P, Lopez G, Capdevila C, Salvatori B, Yu J, Rodriguez-Barrueco R *et al.* Cross-Cohort Analysis Identifies a TEAD4-MYCN Positive Feedback Loop as the Core Regulatory Element of High-Risk Neuroblastoma. *Cancer Discov* 2018; **8**: 582–599.
- 29 Sarkar A, Hochedlinger K. The Sox Family of Transcription Factors: Versatile Regulators of Stem and Progenitor Cell Fate. *Cell Stem Cell* 2013; **12**: 15–30.
- 30 Wegner M. From head to toes: the multiple facets of Sox proteins. *Nucleic Acids Res* 1999; **27**: 1409–1420.
- 31 Oikawa T, Yamada T. Molecular biology of the Ets family of transcription factors. *Gene* 2003; **303**: 11–34.
- 32 Worsley Hunt R, Wasserman WW. Non-targeted transcription factors motifs are a systemic component of ChIP-seq datasets. *Genome Biol* 2014; **15**: 412.
- 33 Zuin J, Dixon JR, van der Reijden MIJA, Ye Z, Kolovos P, Brouwer RWW *et al.* Cohesin and CTCF differentially affect chromatin architecture and gene expression in human cells. *Proc Natl Acad Sci* 2014; **111**: 996–1001.
- 34 Hald ØH, Olsen L, Gallo-Oller G, Elfman LHM, Løkke C, Kogner P *et al.* Inhibitors of ribosome biogenesis repress the growth of MYCN-amplified neuroblastoma. *Oncogene* 2019; **38**: 2800–2813.

Reviewers' Comments:

Reviewer #1:

Remarks to the Author:

Decaesteker et al. have produced an insightful and in-depth review of their initial manuscript. They have addressed every single one of my comments and I have no further comments.

Reviewer #3:

Remarks to the Author:

Overall, the authors have not sufficiently addressed the concerns raised in the initial review. The authors provide a verbose response, yet significant data to support their claims are not included. The authors themselves highlight many of the problems with this manuscript including: 1) "We have one biological replicate" is a very troubling statement to be included in their response especially since statistical analysis for much of the data in original manuscript was requested but not included in the revision. The authors need to validate and confirm their initial findings for any reported work based on only 1 replicate. In addition, the authors have not adequately addressed whether the SOX11 roles they report on are general or specific to the CLB-GA cell line.; 2) Follow up work by the authors confirmed issues with their SOX11 knockdown studies and ATAC-Seq work. Rather than solve the difficulties the authors drop the approach all together and instead provide SOX11 overexpression and ATAC-Seq data. While overexpression of any transcription factor can be used to support knockdown data, one should proceed with heightened caution with overexpression work especially when the transcription factor is part of a conserved family, as is the case with SOX11. As the authors state, "SOX proteins share an HMG domain with more than 80% sequence identity, it is not surprising that the DNA consensus motif that they recognize is also highly similar" and therefore it is probable that overexpression of any SOX factor will result in the occupancy of DNA elements that are not bound by that factor at normal expression levels. In brief, by relying solely on SOX11 overexpression to assess the influence on chromatin accessibility, the authors are likely following off-target sites. The authors need to solve their technical difficulties with their knockdown tactic and provide both the knockdown and overexpression results. The authors state in the rebuttal, "higher sensitivity is required to map subtle changes in differential chromatin accessibility". Typically, this is resolved by deeper sequencing of their samples. Perhaps, the authors should consider doing this and then reevaluate their knockdown work.; 3) The authors suggest SOX11 is an important component of "superenhancers" yet they do not link the various SOX11-superenhancers to the regulated genes. The authors should use simple 3C to show which genes are regulated by their SOX11-superenhancers. In the absence of such information, one cannot conclude whether there is any physiological significance to the potential changes in chromatin accessibility that SOX11 may or may not mediate at enhancer sites.; and 4) In the initial manuscript the authors used a 48h time point to assess the influence of SOX11 on global gene expression. In the response, the authors agree this time point was too late and state "RNA-sequencing experiment 48h upon overexpression of SOX11 will not only provide the direct effects but also indirect effects". In the revised manuscript, the authors use a 9 hour time point. Unfortunately, 9 hours is again too long to identify only direct targets. Furthermore, overexpression of any SOX protein to draw conclusions about any one SOX factor should be cautiously interpreted. Both knockdown and overexpression work should be provided.

REBUTTAL LETTER

Reviewer 1 had no further questions. Reviewer 3 had a few remaining concerns. We have answered each question of reviewer 3, adding relevant sections of the manuscript in blue with new added text underlined and removed text marked by 'strike through'. In the attached updated manuscript file, new text is indicated in blue and removed text is also marked by 'strike through'. References for the rebuttal are listed at the end and if appropriate also included in the adapted manuscript.

Reviewer #3 (Remarks to the Author):

Author's response:

We thank Reviewer #3 for careful assessment of our manuscript and drawing our attention to a number of remaining questions. We have answered each question below and adapted the manuscript accordingly.

Questions:

Reviewer 3, Question 1: *“We have one biological replicate” is a very troubling statement to be included in their response especially since statistical analysis for much of the data in original manuscript was requested but not included in the revision. The authors need to validate and confirm their initial findings for any reported work based on only 1 replicate. In addition, the authors have not adequately addressed whether the SOX11 roles they report on are general or specific to the CLB-GA cell line.*

1. Author's response: In Table 1 we provide an overview of all experiments done indicating the number of cell lines and replicates. We want to stress that for **all *in vitro* and functional experiments the appropriate replicates and statistics were performed.**

In relation to 4C-sequencing, this was indeed performed on one adrenergic neuroblastoma (NB) cell line (CLB-GA) and one mesenchymal NB cell line (SH-EP) with inclusion of statistical analysis using the tool peakC¹ (Figure S2d and Methods section “4C-sequencing”). **We performed additional 4C-sequencing experiments in the adrenergic cell line KELLY and mesenchymal cell line SK-N-AS,** confirming looping between the downstream enhancer loci with the *SOX11* promoter in the adrenergic cell lines while this interaction was not detectable in the mesenchymal cell lines (Fig. S2d).

Furthermore and importantly, publicly available Hi-C data and functional CRISPR data in the neuroblastoma cell line KCNR provide additional evidence for functional activity of the downstream (super)-enhancer region in regulation of *SOX11* expression² (Rebuttal Fig. 1A). Additionally, Hi-C data by Dixon et al. in H1-derived human neuronal progenitor cells³ and Bonev et al.⁴ during mouse neural differentiation show that the insulation of the topologically associated domain in the neighborhood of the *SOX11* region dramatically increases only upon expression of *SOX11* in neuronal progenitor cell. Taken together, **we therefore trust that our novel included 4C-sequencing data and available public data sufficiently support the functional looping between *SOX11* and its downstream (super-)enhancers.**

Rebuttal Figure 1: Mapping of CLB-GA 4C-seq data onto A) Hi-C data in the NB cell line KCNR from Banerjee et al.² and B) Hi-C data in SOX11 expressing H1-derived neuronal progenitor cells from Dixon et al.³ showing similar contact points downstream of SOX11 for both datasets.

We adapted the manuscript as indicated below:

SOX11 is flanked by multiple *cis*-interacting adrenergic specific enhancers

(...) To provide further physical evidence for looping and contact of the cell type-specific enhancers with the promoter of *SOX11*, we performed 4C-seq analysis for the *SOX11* locus in CLB-GA, KELLY (adrenergic MNoA and MNA cell line respectively with multiple *SOX11* downstream enhancers), SH-EP and SK-N-AS (mesenchymal) NB cell lines and observed looping in this highly active region between the downstream enhancer loci with the *SOX11* promoter in the adrenergic cell lines CLB-GA and KELLY while this interaction was not detectable in the mesenchymal cell lines SH-EP and SK-N-AS (Fig. S2d). In support of our findings, interaction of the consensus super-enhancer with the *SOX11* promoter in adrenergic NB cells KCNR was found by Banerjee et al.² using Hi-C analyses. Moreover, targeting of this super-enhancer using CRISPR interference caused attenuated *SOX11* expression. (...)

Adapted Figure S2d:

Adapted legend Figure S2d: 4C-seq analysis of the promoter site and (super-)enhancer region downstream of *SOX11* (chr2, 5.6-7.1Mb, hg19) in the NB cell lines CLB-GA (blue), *KELLY* (green), SH-EP (orange) and *SK-N-AS* (red). The viewpoint is located at the *SOX11* transcription start site (cut out 100 kb). Differential track is show that interactions with downstream enhancers and the *SOX11* promoter were present in adrenergic *SOX11* expressing cell lines (CLB-GA and *KELLY*) and absent in *SOX11*-negative mesenchymal cell lines (SH-EP and *SK-N-AS*). Interaction peaks called by PeakC are shown underneath 4C-seq data. Published and unpublished CHIP tracks for ATAC, H3K27ac, H3K4me1, PHOX2B, HAND2, GATA3 and CTCF in CLB-GA, H3K27ac in *KELLY*, ATAC and H3K27ac in SH-EP and H3K27ac in *SK-N-AS*. Signal represents log likelihood ratio for the CHIP signal as compared to the input signal (RPM normalised). Super-enhancers of CLB-GA are annotated using ROSE (orange bar).

Rebuttal Table 1: Overview of experiments included in the paper, indicating the number of biological and technical replicates.

Technique	Biological replicates / cell lines	Conditions	Technical Replicates
4C-seq	CLB-GA	SOX11 viewpoint	1
	KELLY		
	SK-N-AS		
	SH-EP		
CFA	CLB-GA	siSOX11 14 days	3
	NGP		
	SK-N-AS		
Cell cycle analysis	IMR32	shSOX11 6 days	3
	CLB-GA		
	NGP		

Proliferation assay	IMR-32	shSOX11 4 days	3
	NGP		
Scratch wound assay	SH-EP SOX11 clone1	SOX11 OE 18h	3
	SH-EP SOX11 clone2		
	SH-EP SOX11 clone3		
RNA-seq	IMR-32	siSOX11 48h	4
	CLB-GA		
	NGP		
	SH-EP	SOX11 OE 9h	3
	SH-EP	SOX11 OE 48h	
ChIP-seq/CUT&RUN	IMR-32	SOX11 IP	2
CUT&RUN	CLB-GA	SOX11 IP	1
	NGP		
	SH-EP	SOX11 OE 48h, SOX11 IP	
ATAC-seq	SH-EP	SOX11 OE 48h	3

Of further note, for the identification of SOX11 direct binding sites we want to stress that we have **independent ChIP-sequencing and CUT&RUN data for IMR-32** which are congruent (Fig. S5a), **two CUT&RUN experiments for two additional cell lines (CLB-GA and NGP) and CUT&RUN for SOX11 overexpression in SH-EP cells**. Given the strong congruence for all three adrenergic neuroblastoma cell lines as well as the *SOX11* inducible SH-EP neuroblastoma cells, and further congruence with the transcriptome data of *SOX11* knockdown and overexpression (Fig. 5 and S5), we are strongly confident concerning the validity of our *SOX11* target gene identification presented in this study.

Reviewer 3, Question 2: *Follow up work by the authors confirmed issues with their SOX11 knockdown studies and ATAC-Seq work. Rather than solve the difficulties the authors drop the approach all together and instead provide SOX11 overexpression and ATAC-Seq data. While overexpression of any transcription factor can be used to support knockdown data, one should proceed with heightened caution with overexpression work especially when the transcription factor is part of a conserved family, as is the case with SOX11. As the authors state, "SOX proteins share an HMG domain with more than 80% sequence identity, it is not surprising that the DNA consensus motif that they recognize is also highly similar" and therefore it is probable that overexpression of any SOX factor will result in the occupancy of DNA elements that are not bound by that factor at normal expression levels. In brief, by relying solely on SOX11 overexpression to assess the influence on chromatin accessibility, the authors are likely following off-target sites. The authors need to solve their technical difficulties with their knockdown tactic and provide both the knockdown and overexpression results. The authors state in the rebuttal, "higher sensitivity is required to map subtle changes in differential chromatin accessibility". Typically, this is resolved by deeper sequencing of their samples. Perhaps, the authors should consider doing this and then reevaluate their knockdown work.*

2. Author's response: (See answer question 3) Both question 2 and 3 of the reviewer are concerning our ATAC-sequencing data. For this reason, we decided to combine the answers here below.

Reviewer 3, Question 3: *The authors suggest SOX11 is an important component of "superenhancers" yet they do not link the various SOX11-superenhancers to the regulated genes. The authors should use simple 3C to show which genes are regulated by their SOX11-superenhancers. In the absence of such information, one cannot conclude whether there is any physiological significance to the potential changes in chromatin accessibility that SOX11 may or may not mediate at enhancer sites.*

3. Author's response: We thank this reviewer for further critical evaluation of the ATAC sequencing data.

- 3.1. First, we want to stress that our *SOX11* overexpression model in SH-EP yields a remarkable consistency between direct target genes identified through CUT&RUN mapping of *SOX11* DNA binding sites (after induction) and significantly differentially regulated genes upon (siRNA) knockdown in adrenergic neuroblastoma cells AND *SOX11* overexpression in SH-EP cells. This demonstrates that **forced overexpression of *SOX11* in SH-EP cells can recapitulate the *SOX11*-regulome that we observe in adrenergic NB cell lines** with appropriate *SOX11* expression levels. However, we agree with this reviewer that this not preclude that overexpression does yield, in addition to the bona fide binding sites also some off-target binding sites due to forced overexpression in these mesenchymal cells.
- 3.2. For our ATAC-seq protocols, we performed extensive optimization including OMNI-ATAC and FAST-ATAC approaches, and the current protocol yields robust data for our *SOX11* overexpression system. However, while *SOX11* knockdown in several adrenergic cell lines was highly consistent and our data is very reliable and consistent for monitoring regulation of target genes, no substantial effects were noted on chromatin accessibility upon ATAC-sequencing. This can in partial be explained by incomplete knockdown of *SOX11* upon use of siRNAs. This can in partial be explained by incomplete knockdown of *SOX11* upon use of siRNAs. **We are strongly convinced that partial knockdown approach will not deliver relevant data, while complete CRISPR knockout will result in cell death given the strong *SOX11* dependency for survival of adrenergic neuroblastoma cells** (see also growth arrest and substantial cell death upon transient knockdown within 48-72h, Fig. 3 and S3). Taken together, we propose not to invest further in the knockdown approach.
- 3.3. To answer question 3, we looked deeper into the enhancer landscape regulated upon *SOX11* overexpression using the enhancer-gene link prediction tool PEREGRINE, which incorporates publicly available experimental data from ChIA-PET, eQTL, and Hi-C assays across 78 cell and tissue types including neuroblastoma⁶. While our **ATAC data appear trustable, given that *SOX11* bound sites are enriched for dynamically regulated chromatin states** (Rebuttal Fig. 2a) and 75-80% of differential ATAC regions mapped to known enhancers and bound by *SOX11* (Rebuttal Fig. 2b), **no impact was noted on expression levels for PEREGRINE-predicted enhancer associated genes** (Rebuttal Fig. 2c). This shows discordance between chromatin accessibility and functional consequences and therefore precluding interpretation towards the functional significance of *SOX11* impact on chromatin accessibility. Importantly, **while it is generally assumed that chromatin accessibility is correlated with gene expression at a given locus, recent data have revealed that for single factor perturbations this is not necessarily the case**⁵. This is particularly evident from our deeper analyses of differential ATAC-seq data upon *SOX11* overexpression in SH-EP cells, observing that SH-EP cells already show opened chromatin at promoters of the *bona fide* *SOX11* targets genes (Rebuttal Fig. 2d). This may be partly explained by the fact that for some genes, expression is present but low and induced further by *SOX11*. This observation will require further investigation, which we consider to be beyond the scope of this paper.
- Next, we looked into overlap between differential ATAC-seq peaks and known mesenchymal and adrenergic super-enhancers¹¹, but observed no detectable changes in chromatin accessibility at super-enhancer regions nor do we see differential chromatin accessibility in common regions bound by the adrenergic CRC (data not shown), suggesting *SOX11* overexpression in itself is not sufficient to induce a transition in cell lineage. At present, functional understanding of the core regulatory circuitries (CRC) of transcription factors (TFs) and their combined or individual roles in mesenchymal versus adrenergic neuroblastoma cells is lacking and therefore the **overexpression of one single transcription factor (*SOX11* in this study) in the mesenchymal SH-EP cells is likely**

too simplistic to draw strong conclusions on the epigenetic role of SOX11 in SH-EP cells without accompanying expression of the other CRC TFs in these cells with mesenchymal identity.

Therefore, all together, we propose to take out the ATAC-seq data as at present it does not deliver any substantial additional functional insights while not jeopardizing the main messages of our paper.

Rebuttal Figure 2: A. Heatmap profiles -2 kb and $+2$ kb around the summit of downregulated and upregulated differential ATAC-seq peaks upon SOX11 overexpression for 48h in SH-EP (3 biological replicates, UT = untreated, DOX = doxycycline mediated induction of SOX11). Density profiles are shown representing the average ATAC-seq signal at the presented regions for upregulated regions (green) and downregulated regions (blue) as well as SOX11 CUT&RUN binding in SH-EP cells after SOX11 overexpression for 48h. B. Genome-wide peak annotation distribution (%) (Homer annotation) for the downregulated and upregulated differential ATAC-seq peaks upon SOX11 overexpression for 48h in SH-EP. C. Overlap of genes that are linked with enhancers (based on enhancer-gene links determined by PEREGRINE) that are show differentially reduced (left) or increase (right) chromatin accessibility (as determined by ATAC-seq) with differentially regulated genes as determined by RNA-sequencing. D. Heatmap profiles -2 kb and $+2$ kb around the transcription start site of differential SOX11 targets in SH-EP after SOX11 overexpression for 48h, subdivided in upregulated and downregulated genes. On these regions ATAC-sequencing data in SH-EP cells after SOX11 overexpression for 48h are mapped (3 biological replicates, UT = untreated, DOX = doxycycline mediated induction of SOX11).

To adapt the manuscript, we removed Fig. 6 and S6 and associated text:

Abstract

In addition, epigenetic regulators including the histone deacetylase HDAC2, the H3K27me3 reader and canonical PRC1 complex component CBX2, the chromatin-modifying enzyme lysine-specific demethylase 1 KDM1A/LSD1 and pioneer transcription factor c-MYB are regulated by SOX11. ~~Indeed, forced overexpression of SOX11 in mesenchymal SH-EP neuroblastoma cells induces genome wide chromatin accessibility changes. Finally, we propose SOX11 as a novel *bona fide* master transcription factor of the recently established adrenergic core regulatory circuitry (CRC) in adrenergic high-risk neuroblastoma with a putative function as epigenetic master regulator upstream of the core regulatory circuitry.~~

Introduction

Notably, (1) SOX11 directly regulates 10 SWI/SNF core components and subunit encoding genes, including *SMARCC1*, *SMARCA4* and *ARID1A*, ~~(2) affects global chromatin accessibility,~~ (2) is identified as a *bona fide* early expressed transcription factor of the adrenergic CRC in adrenergic high-risk neuroblastoma and (3) impacts on the adrenergic or mesenchymal transcriptional cell identity but does not induce full phenotypic conversion. We propose SOX11 as epigenetic master regulator upstream of the core regulatory circuitry involved in co-initiation or establishment and/or maintenance of the adrenergic neuroblastoma core regulatory circuit and cell identity.

Forced SOX11 overexpression in SH-EP NB cells impacts genome wide chromatin accessibility

~~Given that (1) the SOX11 regulome when overexpressed in SH-EP NB cells largely recapitulates its endogenous transcriptional activity in adrenergic NB cells, (2) SOX11 regulates a broad epigenetic machinery including SWI/SNF and that (3) forced SOX11 overexpression in SH-EP cells attenuated the mesenchymal gene signature, we selected this model to study the impact of SOX11 on genome wide chromatin accessibility. Differential ATAC-seq 48h after SOX11 overexpression in SH-EP identified 1847 regions with altered chromatin accessibility, *i.e.* closed (n=871) and opened (n=976) (Fig. 6a). Differential ATAC sites were predominantly located at enhancer regions (Fig. 6b) and overlap of SOX11 binding sites in SH-EP after SOX11 overexpression with differential ATAC peaks indicates a direct role of SOX11 in chromatin accessibility. We furthermore see high enrichment in the differential ATAC peaks for a TGA(G/C)TCA motif known to be bound by several transcription factors of the bZIP family including JUN, ATF3, FOSL1, FOSL2 and SWI/SNF component SMARCC1 (Fig. 6c). As mentioned previously, SMARCC1 is a direct SOX11 target with induced expression upon SOX11 overexpression and therefore a possible SOX11 controlled mediator of chromatin accessibility at these sites. In addition, TEAD motifs are highly enriched in the open chromatin regions, which is in line with TEAD2 previously described as a direct regulated target of SOX11 in neuronal development (Fig. 6c). Additionally, Rajbhandari et al. proposed TEAD4 as an important positive regulator of MYCN and prognostic marker in high-risk NB. In closed chromatin regions upon SOX11 overexpression, we identified high enrichment for a C/EBP motif (Fig. 6c). Of note, C/EBPB and C/EBPD are master regulators of the mesenchymal subtype in glioblastoma and C/EBPB is downregulated upon SOX11 overexpression (adj.pval 6.5e-7, logFC 0.93). Additionally, we found C/EBP motif enrichment in the downregulated genes upon SOX11 overexpression (Fig. S6c), indicating the downregulation of mesenchymal markers in the SH-EP cells upon SOX11 overexpression.~~

~~However, while we do observe positive and negative enrichment of adrenergic and mesenchymal gene signatures upon SOX11 overexpression respectively (Fig. S4h) we did not observe strong changes in chromatin accessibility at mesenchymal nor adrenergic super enhancers (data not shown), nor do we see differential chromatin accessibility in common regions bound by the adrenergic master transcription factors (Fig. S6c), suggesting SOX11 overexpression in itself is not sufficient to induce a full transition of cell lineage, at least not at 48 hours after induction of SOX11. Indeed, the core regulatory circuitry members in adrenergic NB are not upregulated after SOX11 overexpression for 48h (Supplementary Table 2). In conclusion, we postulate that forced SOX11 overexpression leads to global changes in chromatin accessibility, specifically of enhancer regions, and we assume that additional~~

~~SOX11 co-drivers are needed to drive epigenetic plasticity towards full mesenchymal-adrenergic transitions.~~

SOX11 is a core regulatory circuitry transcription factor in adrenergic NB

Taken together, our findings support the notion that SOX11 is a canonical CRC member and plays a distinct role, during early sympathoblast development prior to emergence of the adrenergic master regulator PHOX2B and the other CRC members including HAND2 and GATA3. In conclusion, we postulate that SOX11 mediates establishment and maintenance of the adrenergic core regulatory circuitry by modulating the expression of chromatin remodeling complexes and acting as an epigenetic master regulator upstream of the core regulatory circuitry.

Discussion

(...) In addition, several other important epigenetic regulators were noted including chromatin silencing PRC1 complex components and pioneering transcription factor c-MYB. ~~In line with the observed function of SOX11 on chromatin regulatory complexes, forced overexpression of SOX11 in mesenchymal SH-EP neuroblastoma cells induces genome wide chromatin accessibility changes.~~ While these targets require further individual functional validation, the finding of multiple functional targets implicated in a broad range of essential epigenetic regulatory processes is intriguing. (...)

Taken together, these observations, together with the previously established role of SWI/SNF chromatin remodeling in maintenance of lineage-specific, we hypothesize that SOX11 allows NB cells to benefit from enhanced SWI/SNF activity and chromatin remodeling to sustain the establishment and maintenance of the adrenergic core regulatory circuitry of these arrested immature transforming sympathoblasts during tumor initiation.

Methods

Chromatin immunoprecipitation (ChIP) assay and ATAC-seq

~~Chromatin immunoprecipitation (ChIP) and Assay for Transposase Accessible Chromatin (ATAC) sequencing was performed as previously described. (...) For ATAC-seq, 50,000 cells were lysed and fragmented using digitonin and Tn5 transposase. The transposed DNA fragments were amplified and purified using Agencourt AMPure XP beads (Beckman Coulter). ChIP-seq and ATAC-seq libraries were sequenced on the NextSeq 500 platform (Illumina) using the NextSeq 500 High Output kit V2 75 or 150 cycles (Illumina).~~

CUT&RUN and ChIP-seq and ATAC-seq data processing and analysis

~~Prior to mapping to the human reference genome (GRCh37/hg19) with bowtie2, quality of the raw sequencing data of CUT&RUN and ChIP-seq and ATAC-seq was evaluated using FastQC and adapter trimming was done using TrimGalore. (...) for ChIP-seq and ATAC-seq all sequenced reads were mapped. (...) DiffBind was used for differential ATAC peak analysis. (...)~~

Data availability

~~The RNA-sequencing, CUT&RUN and ChIP-sequencing and ATAC-sequencing datasets generated during this study were deposited in the ArrayExpress database at EMBL-EBI (www.ebi.ac.uk/arrayexpress) with accession numbers: E-MTAB-9340, E-MTAB-11883, E-MTAB-11892, E-MTAB-9338, E-MTAB-9464 and E-MTAB-11905 and E-MTAB-11898. (...)~~

Reviewer 3, Question 4. *In the initial manuscript the authors used a 48h time point to assess the influence of SOX11 on global gene expression. In the response, the authors agree this time point was too late and state “RNA-sequencing experiment 48h upon overexpression of SOX11 will not only provide the direct effects but also indirect effects”. In the revised manuscript, the authors use a 9 hour time point. Unfortunately, 9 hours is again too long to identify only direct targets. Furthermore, overexpression of any SOX protein to draw conclusions about any one SOX factor should be cautiously interpreted. Both knockdown and overexpression work should be provided.*

4. Author's response: In order to achieve a reliable and comprehensive identification of SOX11 regulated target genes we **indeed combined knockdown AND overexpression** as suggested by this reviewer. To further avoid inclusion of possible non-physiological regulated targets for SOX11 upon overexpression in SH-EP cells, we additionally included an **extra validation step** through correlation analysis for *SOX11* expression in two independent NB tumor cohorts (GSE85047 and GSE45547) (Fig. 4 and S4).

Upon re-evaluation of early regulated targets, we indeed observe that some SOX11 targets are already differentially expressed earlier than 9h after *SOX11* induction (Rebuttal Fig. 3A). However, we want to point out that based on RNA-sequencing data alone one cannot make any claims concerning direct versus indirect targets as even at earlier timepoints than 9h, indirect effects may still be present. For this reason, in the context of RNA-sequencing data, we refer to genes regulated at 9h as 'early SOX11 regulated genes' and 48h 'late SOX11 regulated genes'. In order to map direct targets of SOX11, we **supplemented RNA-sequencing data with SOX11 binding data using CUT&RUN** in three different adrenergic NB cell lines (IMR-32, CLB-GA, NGP) as well as in SH-EP after *SOX11* overexpression. SOX11 binds both to a subset of early regulated targets (9h) and late regulated targets (48h) (Fig. 4C), indicating that **there are both early and late regulated direct SOX11 targets**. Of further importance, one should also consider that transcription factors can act in a **dose-dependent way** with subsets of genes induced at low levels and others at higher SOX11 levels as has been illustrated to be the case for MYC and MYCN⁷. Notably, in this context, it is important to indicate that at 9h, *SOX11* expression levels are half of those measured at 48h (Rebuttal Fig. 3B).

Taken together, through the combined analysis of RNA-sequencing data subsequent to *SOX11* knockdown AND overexpression, as well as validating based on expression data in primary NB tumors and SOX11 binding data using CUT&RUN, we believe we provided ample validation for the proposed direct targets identified for SOX11.

Rebuttal Figure 3: A. *SOX11*, *CBX2*, *MARCKSL1*, *MEX3A*, *MEX3B* and *c-MYB* relative mRNA expression levels upon *SOX11* overexpression in SH-EP cells over time. Error bars represent the standard error of two technical replicates. B. (left) *SOX11* protein levels upon *SOX11* overexpression in SH-EP cells over time as well as *SOX11* protein levels in IMR-32. Beta-actin is used as loading control. (right) Quantification using ImageJ of the relative *SOX11* protein expression levels in SH-EP after *SOX11* overexpression compared to *SOX11* levels in IMR-32. *SOX11* expression levels are normalized to the loading control.

Rebuttal references:

- 1 Geeven G, Teunissen H, de Laat W, de Wit E. peakC: a flexible, non-parametric peak calling package for 4C and Capture-C data. *Nucleic Acids Res* 2018; **46**: e91.

- 2 Banerjee D, Gryder B, Bagchi S, Liu Z, Chen H-C, Xu M *et al.* Lineage specific transcription factor waves reprogram neuroblastoma from self-renewal to differentiation. 2020; : 2020.07.23.218503.
- 3 Dixon JR, Jung I, Selvaraj S, Shen Y, Antosiewicz-Bourget JE, Lee AY *et al.* Chromatin architecture reorganization during stem cell differentiation. *Nature* 2015; **518**: 331–336.
- 4 Bonev B, Mendelson Cohen N, Szabo Q, Fritsch L, Papadopoulos GL, Lubling Y *et al.* Multiscale 3D Genome Rewiring during Mouse Neural Development. *Cell* 2017; **171**: 557-572.e24.
- 5 Kiani K, Sanford E, Goyal Y, Raj A. Changes in chromatin accessibility are not concordant with transcriptional changes for single-factor perturbation. 2022. doi:10.1101/2022.02.03.478981.
- 6 Mills C, Muruganujan A, Ebert D, Marconett CN, Lewinger JP, Thomas PD *et al.* PEREGRINE: A genome-wide prediction of enhancer to gene relationships supported by experimental evidence. *PLOS ONE* 2020; **15**: e0243791.
- 7 Zeid R, Lawlor MA, Poon E, Reyes JM, Fulciniti M, Lopez MA *et al.* Enhancer invasion shapes MYCN-dependent transcriptional amplification in neuroblastoma. *Nat Genet* 2018; **50**: 515–523.

Reviewers' Comments:

Reviewer #3:

Remarks to the Author:

The authors have responded very well to the previous concerns. The manuscript now stands as an important contribution to the field.